# Neurocognitive characterization of behaviour and mental illness through time-varying brain network analysis

Xiao Chang [1,2,3,4,5,33] ✉, Tianye Jia [1,2,3,4,5,33], Zening Fu [6], Shitong Xiang [1,2], Yunman Xia [1,2,3], Chao Xie[1,2,7], Jiyan Zou[1,2], Miao Cao [1,2], Jie Zhang[1,2], Tobias Banaschewski [8], Arun L. W. Bokde [9], Sylvane Desrivières [10], Herta Flor [11,12], Antoine Grigis[13], Hugh Garavan [14], Penny Gowland [15], Andreas Heinz [16], Rüdiger Brühl [17], Jean-Luc Martinot [18,19], Marie-Laure Paillère Martinot[18,20], Eric Artiges [18,19], Frauke Nees [8,11,21], Dimitri Papadopoulos Orfanos [13], Vincent Frouin [13], Luise Poustka[22], Sarah Hohmann [8], Christian Baeuchl [23], Michael N. Smolka [24], Nilakshi Vaidya [25], Henrik Walter [26], Robert Whelan [27], Jianfeng Feng [1,2,4,5,28,29], Vince D. Calhoun [6] & Gunter Schumann [3,25,30,31,32] ✉, On behalf of the IMAGEN Consortium* & environMENTAL Consortium*

Human cognitive processing involves dynamic interactions across brain regions, evolving over time. Traditional neuroimaging analysis often overlooks this temporal aspect, limiting insights into how functional network connectivity (FNC) supports ongoing cognition and behaviour. Using sliding window analysis, we captured FNC changes during tasks, reflecting network reconfiguration in cognitive processes. We further determined behavioural relevance of time-varying FNC by relating network measurements with task performances and psychopathology. We found that several whole-brain FNC patterns, or states, persist across resting and task-based fMRI, with state occurrences fluctuating with the most prominent task stimuli. Regional FNC distinguishes specific task conditions, and time-varying FNC explains more variance in psychopathology symptoms compared to static connectivity. These findings highlight that cognitive tasks reshape regional and whole-brain connectivity. By considering the different FNC states, time-varying connectivity provides a more comprehensive representation of brain interactions and thus may represent a better neural proxy for cognition and behaviour.

Reinforcement-related processes, including reward processing, cognitive inhibition and emotional processing are composed of several fundamental, intertwined aspects of human cognition[1]. Dysfunction of reinforcement-related cognition has been linked to a range of mental health problems during adolescence, including internalizing and externalizing symptoms, many of which transition into mental disorders during adulthood[2,3].

Reinforcement-related cognitive processes engage interactions between brain networks, often measured as functional network connectivity (FNC) derived from functional magnetic resonance imaging

A full list of affiliations appears at the end of the paper. *Lists of authors and their affiliations appear at the end of the paper. ✉e-mail: xchang@fudan.edu.cn; gunter.schumann1961@gmail.com

(fMRI) during execution of cognitive tasks. For example, facial emotional processing involves a network centered on the orbitofrontal cortex, amygdala and the superior temporal sulcus[4,5]. Reward processing is linked to inhibitory regulation between the medial prefrontal cortex and the striatum[6,7]. Meanwhile, regional brain interactions are also part of the large-scale, whole-brain network organization[8,9]. On the whole-brain level, studies found that brain network organization is highly similar ($r > 0.9$) among resting-state and multiple tasks fMRI sessions[10–12]. How brain network reorganizes to support ongoing cognitive tasks is yet an unresolved question.

While brain functional network connectivity is subject to dynamic modulations[13], traditional FNC analyses assume a constant connectivity strength (static connectivity) within a session or a task condition. As a consequence, static FNC analyses, including the FNC analyses cited above, do not provide the sensitivity necessary to register moment-to-moment FNC fluctuations, which are thought to reflect ongoing cognitive activity[14,15]. In the absence of this information, the exact relation between cognitive task demands and FNC remains unclear.

Time-varying FNC analyses applied to resting-state fMRI[16,17] have demonstrated the existence of distinct connectivity patterns[18]. These whole-brain connectivity patterns (i.e. FNC states) are thought to be related to both intrinsic thoughts, arousal, neuromodulation as well as to non-neural sources such as head movement and random fluctuations[19,20]. In resting-state analyses, it can be challenging to determine cognitive correlates of FNC fluctuations, whereas task-based fMRI can provide a benchmark to test the association between FNC changes and ongoing cognitive processes[14,15]. Initial time-varying analyses applied to task-based fMRI have observed that whole-brain network changes with the onset and offset of task blocks, and that their occurrences may differ between tasks[13,21]. However, the more detailed cognitive and behavioural relevance of FNC states across tasks conditions remains to be elucidated.

Symptoms of reinforcement-related cognitive dysfunction are at the core of many psychiatric disorders. For example, dysfunctions in reward processing, cognitive inhibition and emotional processing affects patients with depression as well as patients with alcohol use disorder, among many others[22,23]. Yet, correlations between functional connectivity and individual symptoms of cognitive dysfunction, or indeed psychiatric diagnoses, are neither high nor specific[24]. Previous studies have shown that time-varying FNC is predictive of upcoming behavioural response and correlates with task performance[15,21,25]. Thus, time-varying FNC may serve as a more precised neural characterization of cognitive dysfunction and may increase both strength and specificity of the correlations with behavioural symptoms[26].

In this study, we applied in the IMAGEN cohort, a large European sample of young adults, a time-varying FNC analysis framework to the resting-state (REST) and three reinforcement-related tasks fMRI sessions. These fMRI tasks, namely the emotional faces task (EFT)[27], monetary incentive delay task (MID)[28] and stop-signal task (SST)[23], interrogate emotional processing, reward processing and cognitive inhibition, respectively. We replicated and extended our findings in STRATIFY, an independent dataset of patients with major depression and alcohol dependence who have been characterised in a similar way to IMAGEN. Our aims were to 1) understand the cognitive relevance of time-varying FNC by relating the whole-brain and regional FNC with ongoing task stimuli and performances; 2) determine task specificity of the whole-brain and regional FNC changes across multiple reinforcement-related tasks; 3) investigate the utility of time-varying connectivity in explaining mental health symptoms in the general population and in patients. Our overarching goal is to improve understanding of the neural basis of cognitive processing by elucidating brain network reconfiguration in a time-resolved manner and to provide a better neural correlate for psychopathology using time-varying analysis.

## Results

### Summary of analysis strategy

We analyzed resting-state and three reinforcement-related task-fMRI of 1,417 participants of the IMAGEN cohort[1] at age 19 years (Table S1–S2). (i) Time-varying functional network connectivity (FNC) of each scanning session was derived using sliding window analysis[18] (Fig. 1a) on time-series of 61 components from group independent component analysis (ICA)[29] (Fig. S2 and Table S3). In this study, we used the term FNC to refer to the correlation between ICA components' time-courses[30]; (ii) We examined the similarity of FNC matrix derived from a sliding window (windowed FNC) between time points for each scanning session (Fig. 1b); (iii) To extract reoccurring patterns of whole-brain FNC (FNC state), we performed $k$-means clustering on windowed FNC matrices. (iv) Then we systematically examined coupling between task conditions with whole-brain FNC state and regional FNC to determine the relevance of FNC fluctuation in relation to ongoing cognitive process (Fig. 1c); (iv) We investigated the utility of time-varying FNC in explaining reinforcement-related psychopathology and cognition (Fig. 1d). Behavioural variance explained was compared using the same set of FNC from time-varying FNC and static FNC. (v) In the independent STRATIFY dataset ($n = 439$, Table S13), we validated the generalizability of time-varying FNC in explaining clinical symptoms (Validation analysis). Throughout the manuscript, age, sex, recruitment sites and head motion were regressed in between-subject analyses. Unless specified, two-sided $p$-values were reported, and multiple testing was corrected using the false discovery rate (FDR) at $p < 0.05$.

### Resting-state and task-fMRI sessions contain reoccurring FNC states

We first examined the similarity of windowed FNC matrices between time points and found that whole-brain FNC correlations were constantly changing with scanning time (Fig. 1b). Interestingly, FNC correlations of the block-designed tasks (EFT and MID) showed a periodical temporal structure, suggesting that their whole-brain network is changing within ongoing tasks.

To extract recurring network patterns, we used k-means clustering on windowed FNC matrices. We predefined four clusters using the elbow criterion[31] (Fig. S3). The four different network states identified in this way were present during both, task-fMRI and resting state (REST) sessions (Fig. 2a). Connectivity profiles of the same state in different sessions have a higher correlation (average $r = 0.86 \pm 0.17$) than correlations between different states (average $r = 0.67 \pm 0.10$, two-sample $t$-test: $t_{118} = 7.17$, $p = 6.93 \times 10^{-11}$, 95% CI = [0.13, 0.24], Coden'd = 1.63). The only exception is the MID state 3, which does not resemble state 3 in other sessions. We hypothesize that occurrence of state 3 is related to reduced vigilance, and as MID is the only task with immediate incentives, it is easier for participants to maintain high levels of vigilance throughout the test. SST and EFT showed the four similar connectivity patterns as the REST.

The connectivity profiles indicate that FNC states have different degrees of integration and segregation between functional domains, which can be quantified by modularity and participation coefficient from graph metrics[32] (Fig. 2b and Table S4). The Louvain modularity algorithm was applied to detect the optimal community structure of network[33] (Supplementary Information). FNC state 1 has a significantly higher participation coefficient (positive FNC: $F_{(3,12)} = 22.13$, $p = 3.52 \times 10^{-5}$, $\eta^2 = 0.85$; negative FNC: $F_{(3,12)} = 34.38$, $p = 3.58 \times 10^{-6}$, $\eta^2 = 0.90$) and a lower modularity ($F_{(3,12)} = 37.81$, $p = 2.16 \times 10^{-6}$, $\eta^2 = 0.90$) than the other states (Fig. 2c).

The connectogram plot showed the strongest 100 connectivity of each state (Fig. S4). FNC state 1 is the least modular state, with mainly weakly positive connectivity within functional domains. Both state 2 and state 3 exhibited segregation (strong negative connectivity) between primary regions (visual and sensorimotor networks) and

### a) Time-varying functional network connectivity (FNC) from resting-state and task-based fMRI

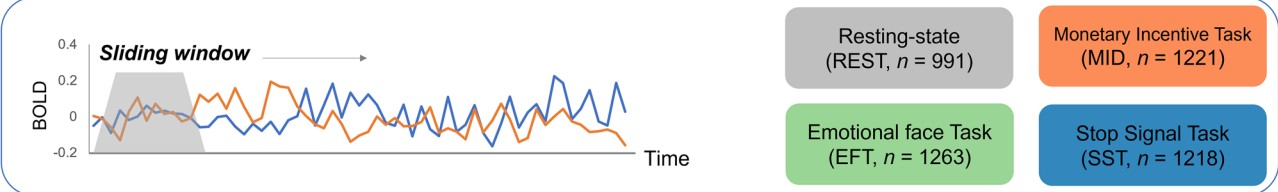

### b) Task modulation on whole-brain FNC state occurrence and regional connectivity

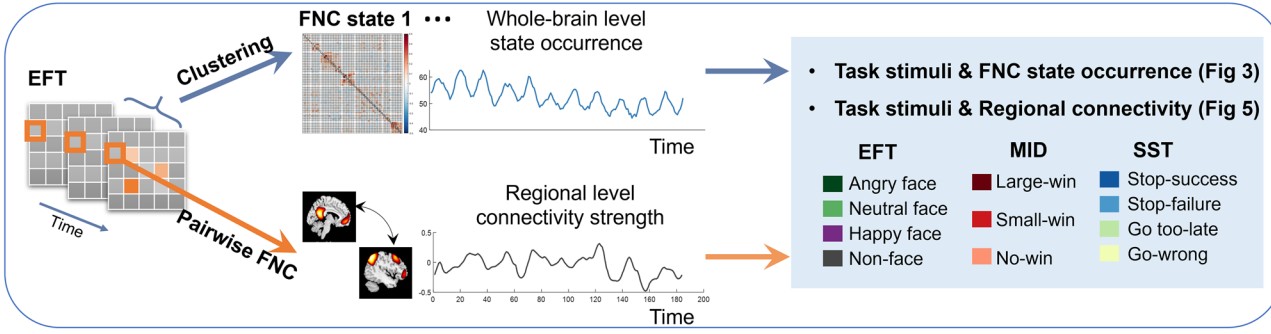

### c) Associations between time-varying or static FNC and behaviours

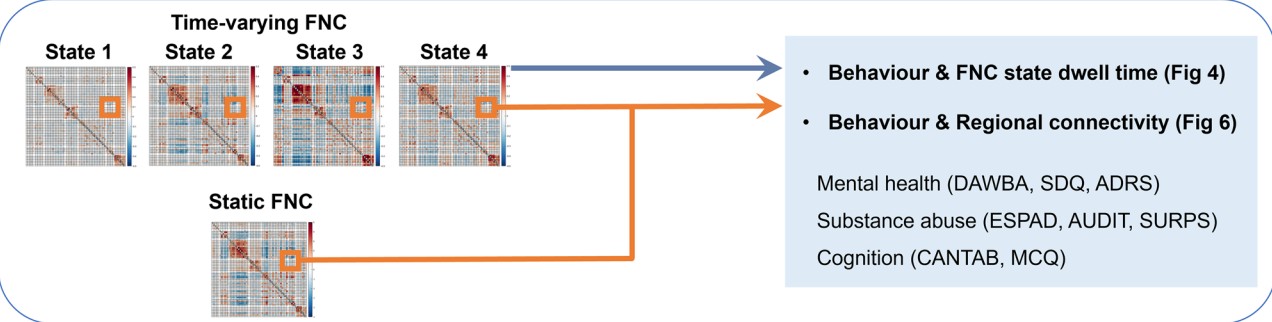

### d) Validation in external STRATIFY cohort

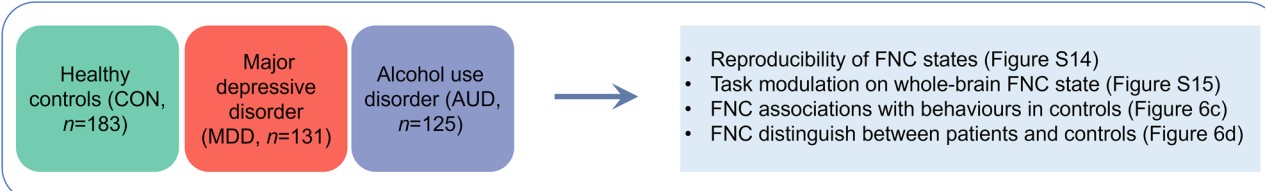

**Fig. 1 | Summary of analysis strategy. a** Resting-state (REST) and three reinforcement-related task-fMRI–emotional faces task (EFT)[27], monetary incentive delay task (MID)[28] and stop-signal task (SST)[23]– from the IMAGEN cohort were analyzed for this study. After pre-processing and quality control, each fMRI session was parcellated into 61 components using the group independent component analysis (ICA). Time-varying functional network connectivity (FNC) between each pair of the 61 components was estimated using the sliding window analysis. The window length is 17.6 s for the REST, EFT and SST sessions, and is 8.8 s for the MID to accommodate the block duration of MID (~10 s). **b** On the whole-brain level, k-means clustering analysis was applied to windowed FNC matrices to derive reoc-curring FNC states from each scanning session. FNC state occurrences (i.e., the proportion of participants classified into a given state) were correlated with ongoing task stimuli to investigate how task conditions modulate whole-brain FNC. On the regional level, connectivity strength between each pair of ICA components was assessed. Correlation between regional connectivity strength and task stimuli was calculated to investigate task modulation on regional connectivity. **c** To determine the behavioural relevance of whole-brain FNC state and regional FNC, we performed sparse partial least squares (sPLS) between FNC states dwell time (the average length of a subject having a certain state) and mental health symp-toms and cognitions. Additionally, we compared behavioural variance explained by regional time-varying FNC versus static FNC using linear regression. **d** The results were validated in an independent clinical dataset, including the repro-ducibility of FNC states and their associations with task modulation and behavioural outcomes.

higher-order regions (default mode, cognitive control networks), with state 3 exhibiting higher modularity and segregation between primary regions and subcortical and cerebellum. The connectivity profile of state 4 is similar to state 1 but with stronger connectivity within functional domains and a lower participation level.

We demonstrated the robustness of our results by confirming reproducibility of the FNC states and dwell time of each state using different window lengths (from 8.8 s to 70.4 s, Figs. S5−6, Table S5) and different clustering numbers (from 2 to 5 clusters, Figs. S7-8, Table S6).

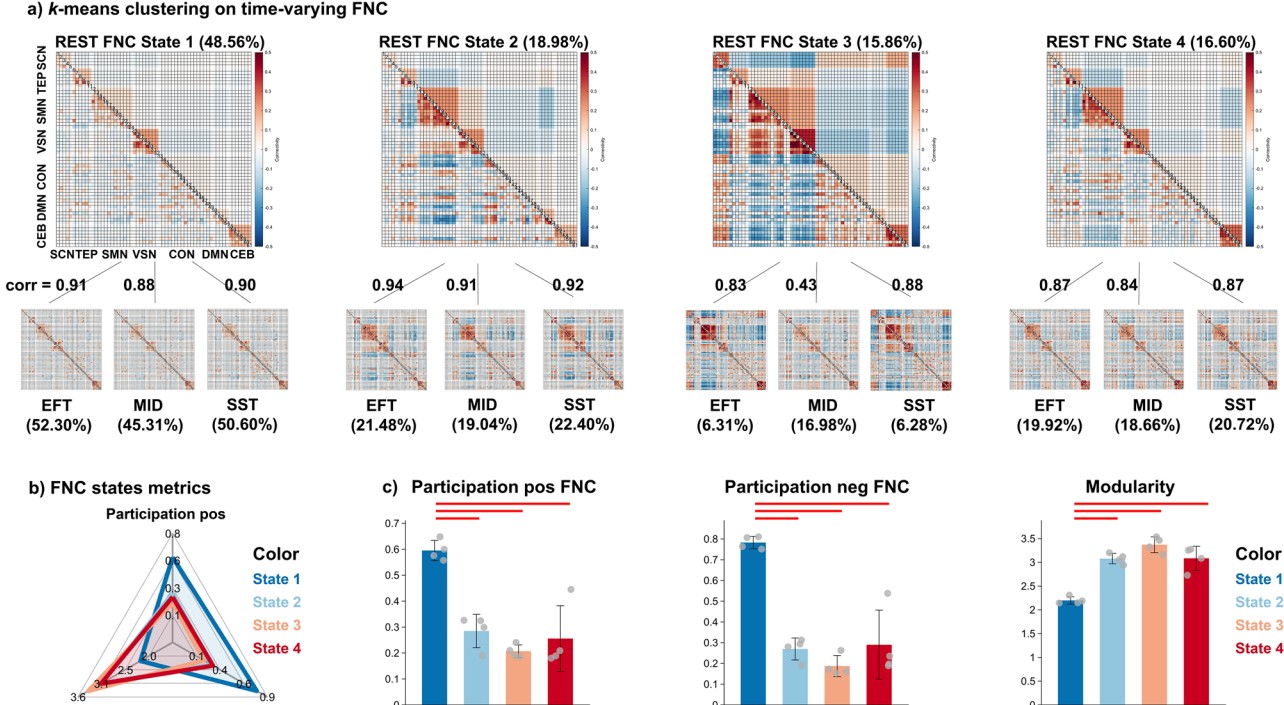

**Fig. 2 | Resting-state and task-fMRI sessions contain reoccurring FNC states.**
**a** Four representative FNC states (cluster centroids) were derived from the *k-means* clustering applied to sliding-window FNC matrices from the resting-state (upper row) and three task-fMRI sessions (lower row). Each FNC state represents a reoccurring functional connectivity pattern between 61 ICA components, which are categorised into seven functional domains (Fig. S2). The percentage of each state within a scanning session is shown in parentheses. (Note: Due to decimal approximation, the percentages for the four FNC states in the EFT and MID sessions sum to 100.01% and 99.99%, respectively). Connectivity profiles of the same state across resting-state and task-fMRI sessions exhibit a higher correlation (average $r = 0.86 \pm 0.17$) compared to correlations between different states (average

$r = 0.67 \pm 0.10$, two-sample *t*-test: $t_{118} = 7.17$, $p = 6.93 \times 10^{-11}$, 95% CI = [0.13, 0.24], Coden'd = 1.63). **b** The four FNC states differ in network modularity and participation coefficients, which quantify the level of segregation and integration between functional modules. The modularity and participation coefficients are averaged across resting-state and task-fMRI sessions. **c** One-way ANOVA and post-hoc *t*-test comparisons between FNC states on modularity and participation coefficients. Red lines indicate significant differences between FNC states after FDR correction (Table S4). Abbreviations for the seven functional domains: subcortical (SCN), temporal (TEP), sensorimotor (SMN), visual (VSN), cognitive control (CON), default mode (DMN) and cerebellar (CEB) networks.

## FNC states are differentially modulated by ongoing task events

Having identified four stable FNC states, we examined how task events modulate their occurrences by calculating the Pearson correlation between FNC state occurrence and task conditions for each task session. For the EFT and MID, where stimuli onset and duration are the same for all participants, the state occurrences were calculated as the proportion of subjects having a given state at a particular time point at the group level. Pearson's correlation was performed between state occurrences and task conditions at the group level, and compared with a null model to determine their significance (Supplementary Information). We also compared the correlation coefficients between conditions (a contrast of interest) using Pearson and Filon's *z*-statistic[34]. For the SST, as stimuli onsets varied across participants, we calculated the partial correlations between state occurrences and task stimuli (controlling for other types of stimuli) at the individual level. Correlation coefficients were compared to 0 using a one-sample *t*-test to determine their significance.

In the EFT task (Fig. 3a), state 1 and state 4 showed positive correlations with angry faces (all $r > = 0.27$, $p < = 0.0002$, Table S7) and negative correlations with non-face stimuli ($r < = -0.55$, $p < 0.0001$). Comparing between conditions, state 1 and state 4 exhibited significantly higher correlations for faces stimuli (angry, happy and neutral faces) than non-face stimuli (all Pearson and Filon's $z > = 6.79$, $p < 0.0001$, Table S8). In contrast, FNC state 2 showed strong positive correlation with non-faces stimuli ($r = 0.85$, $p < 0.0001$), and negative correlations with all types of faces stimuli ($r < = -0.26$, $p < 0.0001$). Occurrence of state 3 linearly increased with scanning time and exhibited positive correlations with non-face stimuli ($r = 0.20$, $p = 0.007$).

In the MID task (Fig. 3b), state 1 had a higher correlation with large-win than small-win trials ($z = 2.35$, $p = 0.02$, Table S8), whereas state 2 showed the inverse pattern again ($z = -2.79$, $p = 0.005$), and negative correlations with large-win trials ($r = -0.20$, $p = 0.008$, Table S7).

In the SST task (Fig. 3c), stimuli onsets differ across subjects. We therefore calculated partial correlations between state occurrence and task stimuli at the individual subject level and compared correlations between conditions using one-sample *t*-test. We analysed stop signals (stop-success and stop-failure) and go errors (go too-late and go-wrong). FNC state 1 showed a positive correlation with stop success ($t = 2.60$, $p = 0.01$, Table S7), whereas FNC state 2 exhibited a negative correlation ($t = -4.60$, $p = 4.63 \times 10^{-6}$). For go-related errors, FNC state 3 showed a strong positive correlation with go too-late error ($t = 15.86$, $p = 2.88 \times 10^{-43}$), while other states exhibited negative correlations ($t < -4.01$, $p < 6.72 \times 10^{-5}$). State 1 and 4 showed positive correlations with go wrong errors ($t > 4.32$, $p < 1.72 \times 10^{-5}$) and state 2 showed a negative correlation ($t = -10.33$, $p = 5.45 \times 10^{-24}$).

Considering all three tasks, FNC state 1 positively correlated with the most prominent task stimuli (EFT faces stimuli, MID large-win and SST stop signals), whereas state 2 negatively correlated with these conditions, suggesting that state 1 and 2 represent task engagement and disengagement, respectively. State 3 occurrence in the EFT and SST linearly increased with scanning time, suggesting an indicator of attention lapses. State 4 only positively correlated with angry faces in the EFT, which may reflect passive emotional processing. This hypothesis was supported by correlation between FNC state dwell time and task performances (Figs. S9-10). We found that SST go too-late

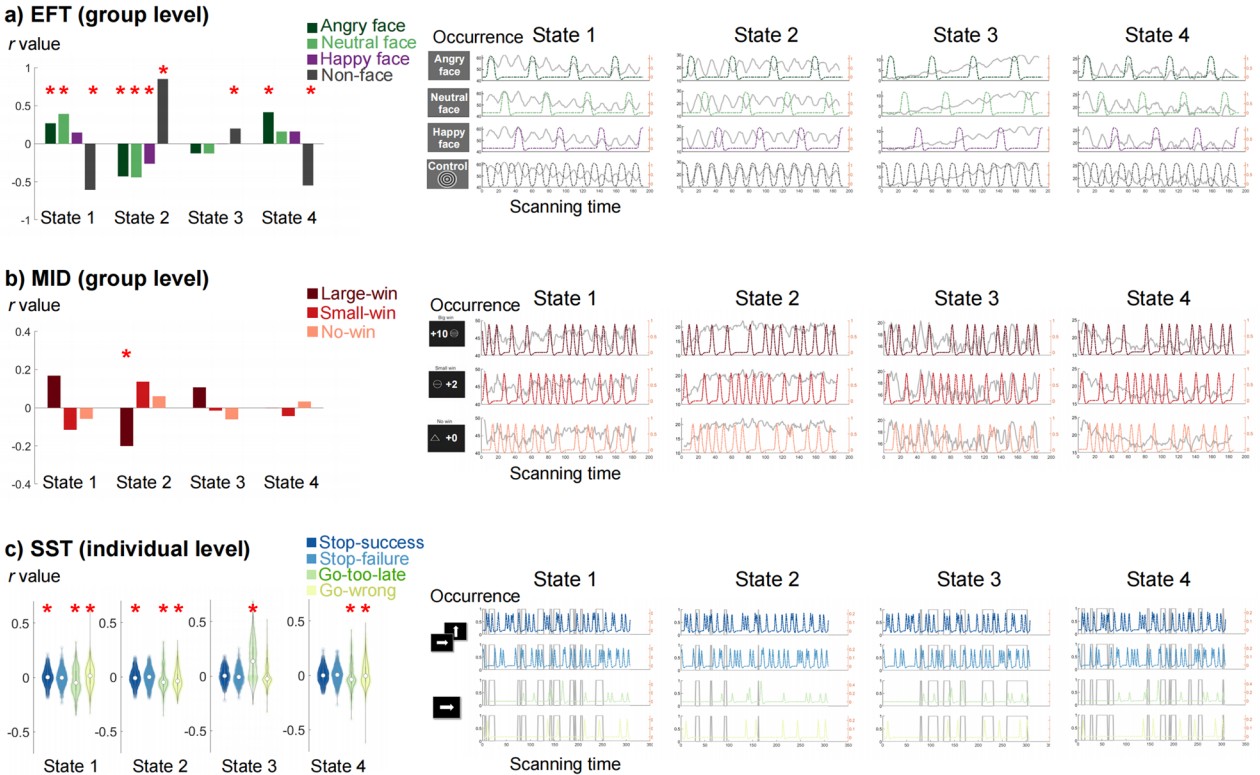

**Fig. 3 | Correlation between FNC state occurrence and task conditions for the EFT, MID and SST sessions.** Correlation coefficients ($r$ value) were shown on the left and compared between conditions. Occurrence of FNC states (grey line) and task conditions (coloured lines) as a function of scanning time were plotted on the right. For the **a** EFT and **b** MID, stimuli onset and duration were the same for all participants, thus we calculated the Pearson correlation between state occurrences and task stimuli at the group level. The significance of correlation coefficients was compared with a null model by randomly shuffling the order of the FNC state label. In the EFT, real human facial images depicting happy, neutral, and angry

expressions were presented as stimuli[27]. **c** For the SST, as stimuli onsets differ across participants, we calculated the partial correlation between state occurrence and task stimuli (controlling for other types of stimuli) on the individual level. A random subject's state occurrence and stimuli onset were shown. Correlation coefficients were compared using one-sample $t$-test to determine their significances (Table S7). Asterisks indicate FDR-corrected statistical significance when comparing correlations between task stimuli and state occurrence with the null model (EFT and MID) or with zero (SST).

error positively correlated with state 3 (Spearman's $rho = 0.42$, $p = 9.03 \times 10^{-24}$) and negatively associated with state 1 ($rho = -0.10$, $p = 9.29 \times 10^{-4}$), suggesting that FNC state 1 facilitates and state 3 interferes with ongoing task performances. Note that FNC state fluctuations were not driven by choice of window length (Fig. S11).

**Whole-brain FNC states correlate with reinforcement-related behaviours**

Next, we examined behavioural relevance of FNC states by carrying out sparse Partial Least Square (sPLS) analysis[35] and hold-out validation on the FNC state dwell time and 29 behavioural items, including mental health assessments (DAWBA, SDQ, ADRS), substance uses (ESPAD, AUDIT, SURPS) and reinforcement-related cognitions (MCQ, CANTAB) (Fig. 4a and Table S9). FNC states 1 and 4 were negatively associated with psychopathology and substance use behaviours, whereas states 2 and 3 showed positive associations (Fig. 4b and Table S10). Specifically, FNC state 1 in the EFT and SST was negatively correlated with internalizing symptoms (specific phobia and peer problems) and externalizing symptoms (smoking, binge drinking and risk-taking). FNC state 2 in the MID and SST was positively correlated with internalizing symptoms, including phobia, depression, eating disorders, as well as externalizing problems, including obsessive-compulsive disorder and ADHD symptoms. FNC state 3 in the REST, EFT and SST sessions was positively correlated with substance use such as smoking, drinking, and related personality including impulsivity and sensation seeking. FNC state 4 in the EFT, MID and SST were negatively correlated with internalizing symptoms, including phobia, depression, anxiety and

positively correlated with impulsivity, sensation seeking and delay aversion, risk-taking from CANTAB gambling task.

**Regional FNC involves in task-specific cognitive processes**

In addition to describing ongoing task modulation on a whole-brain level (Fig. 3), we further investigated task effects on regional FNC between two ICA components (pair-wise FNC). Unlike the whole-brain FNC state, regional FNC could distinguish task conditions related to emotional processing, reward sensitivity or motor inhibition. These task conditions included angry versus neutral faces in EFT; large-win versus no-win in MID and stop-success versus stop-failure in SST. Among all 1830 ($61 \times 60/2$) pair-wise FNC, 559 FNC were significant for EFT contrast; 333 and 303 FNC were significant for MID and SST contrasts, respectively (one-sample $t$-statistic) (Fig. 5a-c).

To determine the specificity of these regional FNC, we plotted 20 connections with the highest $t$-statistics in one task, and compared their $t$-statistics with those in the other two tasks. FNC associated with emotional processing (with the highest $t$-statistics of angry versus neutral faces) do not distinguish task conditions in the MID or SST (Fig. 5a). A similar task specificity was observed for the MID and SST (Fig. 5b, c). Anatomic locations of the 20 task-specific FNC are shown in the connectogram plot (Fig. 5d). The EFT task-specific connections are mainly between the bilateral superior temporal regions and the default mode, prefrontal areas, and visual cortices. For the MID, connections of the default mode regions with frontoparietal regions, visual and sensorimotor areas showed higher correlations with the large-win condition compared to the no-win condition. For the SST, connections of the

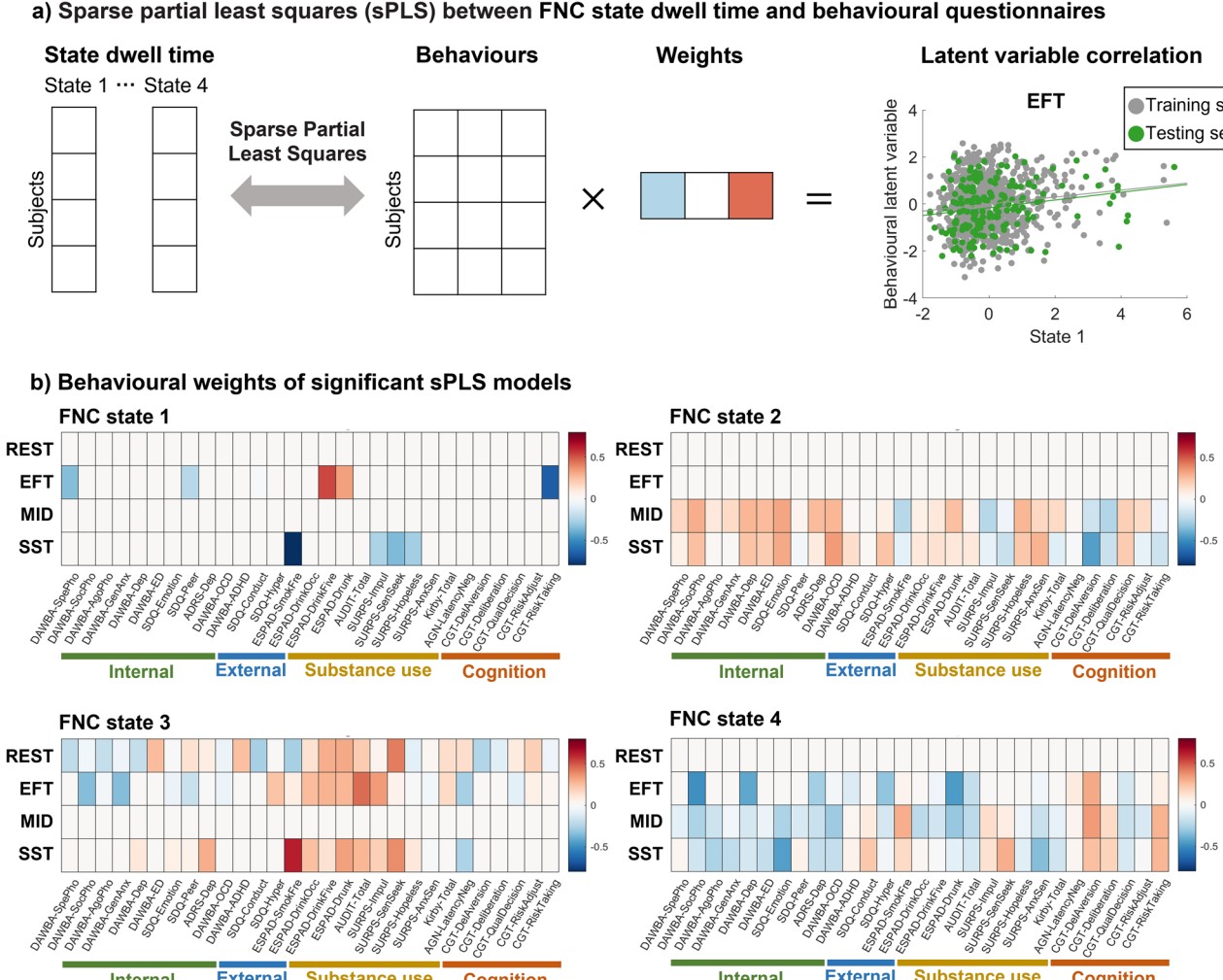

**Fig. 4 | Sparse partial least squares (sPLS) analysis on each FNC state dwell time and behaviour questionnaires for resting-state and task-fMRI sessions. a** A multiple hold-out framework was applied to the sPLS analysis using the CCA/PLS Toolkit[35]. The statistical significance of association in the test set was determined by 5000 iterative permutation procedures ($p < 0.05$). An example correlation between latent variables in the training and test set was plotted on the right. **b** For significant sPLS models ($p < 0.05$), behavioural weights were plotted with a heatmap. Behavioural items were grouped into categories: internalizing symptoms, externalizing symptoms, substance use, and cognitive performances. The correlation value and significance of all sPLS models are listed in Table S10. Behavioural questionnaire abbreviations: Development and Well-Being Assessment (DAWBA); Strengths and Difficulties Questionnaire (SDQ); Adolescent Depression Rating Scale (ADRS); European School Survey Project on Alcohol and Other Drugs (ESPAD); Alcohol Use Disorders Identification Test (AUDIT); Substance Use Risk Profile Scale (SURPS); Monetary-Choice Questionnaire (MCQ); Affective Go-Nogo task (AGN) (CANTAB, www.cambridgecognition.com); Cambridge Gambling Task (CGT) (CANTAB).

sensorimotor areas with default mode and temporal regions, as well as connections between the putamen and frontoparietal regions, showed higher correlations with the stop-success condition compared to the stop-failure condition. Detailed FNC labels and $t$-statistics are listed in Table S11. Taken together, our results suggest that whole-brain and regional FNC may reflect different aspects of cognitive processing: while whole-brain FNC states are broadly modulated by the most salient stimuli (EFT faces stimuli, MID large-win and SST stop signals), regional FNC are related to more subtle task processes and show task specificity.

### Time-varying FNC explains more behavioural variance than static FNC

Finally, we tested whether measuring task-specific regional FNC in different FNC states increases the explanation of variance of reinforcement-related behaviours in the IMAGEN cohort. We averaged the top 20 task-specific connections (Fig. 5) from each time-varying FNC state and static FNC, and then regressed the 29 behavioural items (Table S9) on time-varying FNC and static FNC. We focused on 23 behaviours that can be fitted using both time-varying FNC and static

FNC (full model, model significance uncorrected $p_{\text{one-tail}} < 0.05$, Table S12). While variance explained of individual behaviours by single neuroimaging measures is generally low, we found that time-varying FNC explained a higher proportion of variance ($R^2$) in 22 out of 23 behaviours (adjusted $R^2$ 19/23 behaviours) compared to the static FNC regressiobn model (Fig. 6a). Other model fitting parameters including adjusted $R^2$ and log-likelihood (which balance model accuracy and model complexity) also suggested a better fitting using time-varying FNC for behavioural regression (Fig. S13b and Table S12).

We show one behavioural fitting from each task using time-varying FNC and static FNC as an illustration (Fig. 6b). In the EFT, variance explained of peer problem from Strengths and Difficulties Questionnaire (SDQ) by time-varying FNC was 0.71% compared to 0.07% by static FNC. Correlation between the fitted and observed behavioural score using a time-varying FNC regression model was $r = 0.08$, $p = 0.003$, 95% CI = [0.03, 0.14] and $r = 0.03$, $p = 0.36$, 95% CI = [−0.03, 0.08] in a static FNC model. In the MID, variance explained of emotional problems from SDQ by time-varying FNC was 0.86% compared to 0.64% by static FNC. Correlation between the fitted and

## a) EFT angry faces versus neutral faces effect of regional FNC

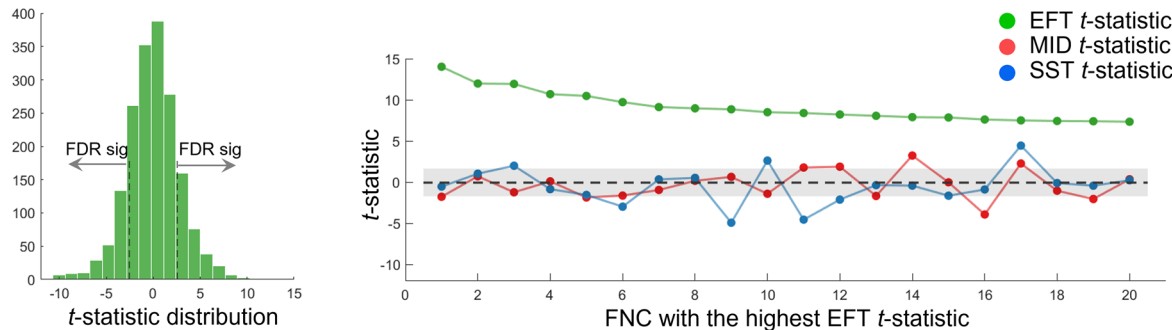

## b) MID large-win versus no-win effect of regional FNC

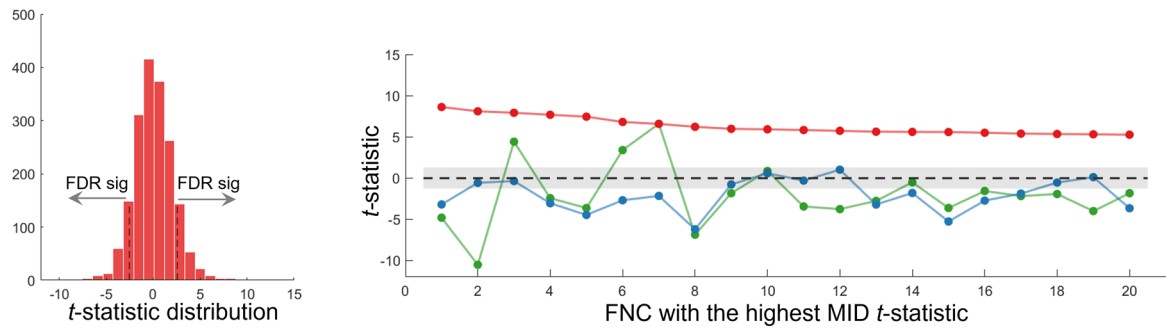

## c) SST stop-success versus stop-failure effect of regional FNC

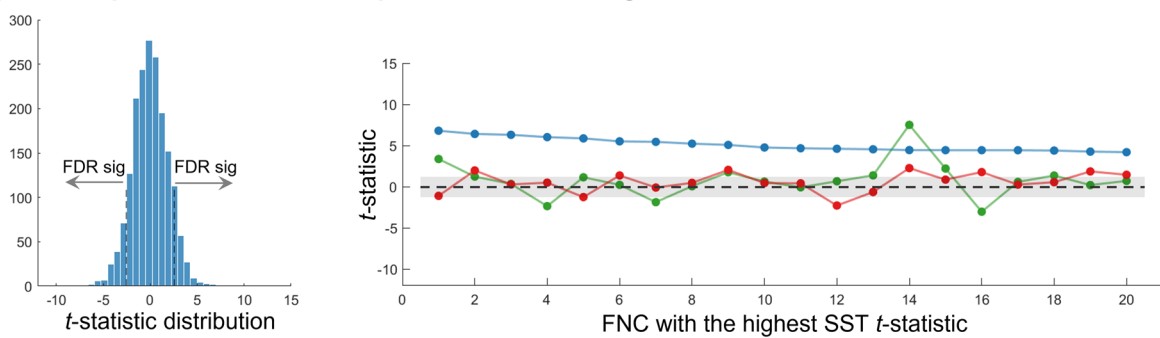

## d) Anatomical locations of the 20 task specific connectivity

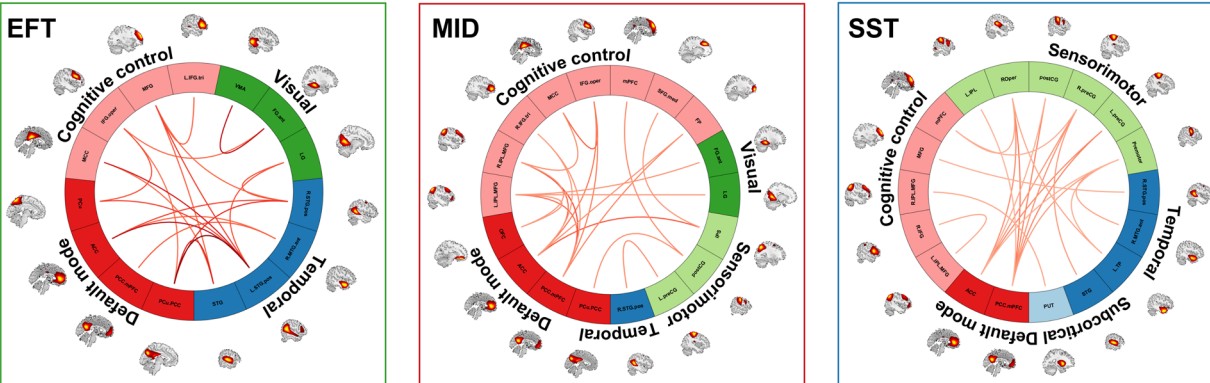

**Fig. 5 | Regional FNC distinguishes contrasts that are related to emotional processing, reward sensitivity and motor inhibition. a** For the EFT, one-sample *t*-tests were performed between FNC correlations with angry faces versus neutral faces. The distribution of *t*-statistic of all 1830 (61×60/2) pair-wise FNC (i.e., connectivity between two ICA components) was shown on the left. On the right shows the 20 task-specific FNC with the highest EFT *t*-statistics, which do not distinguish the MID and SST contrasts; similarly, **b** FNC with the highest *t*-statistics of large-win versus no win condition in the MID do not distinguish the EFT and SST conditions; and **c** FNC with the highest *t*-statistics of stop-success versus stop-failure condition in the SST do not distinguish the EFT and MID conditions. **d** The connectogram plot shows anatomical locations of the 20 task-specific FNC of EFT, MID and SST. Their functional domain names were labelled according to Fig. S2 and Table S3.

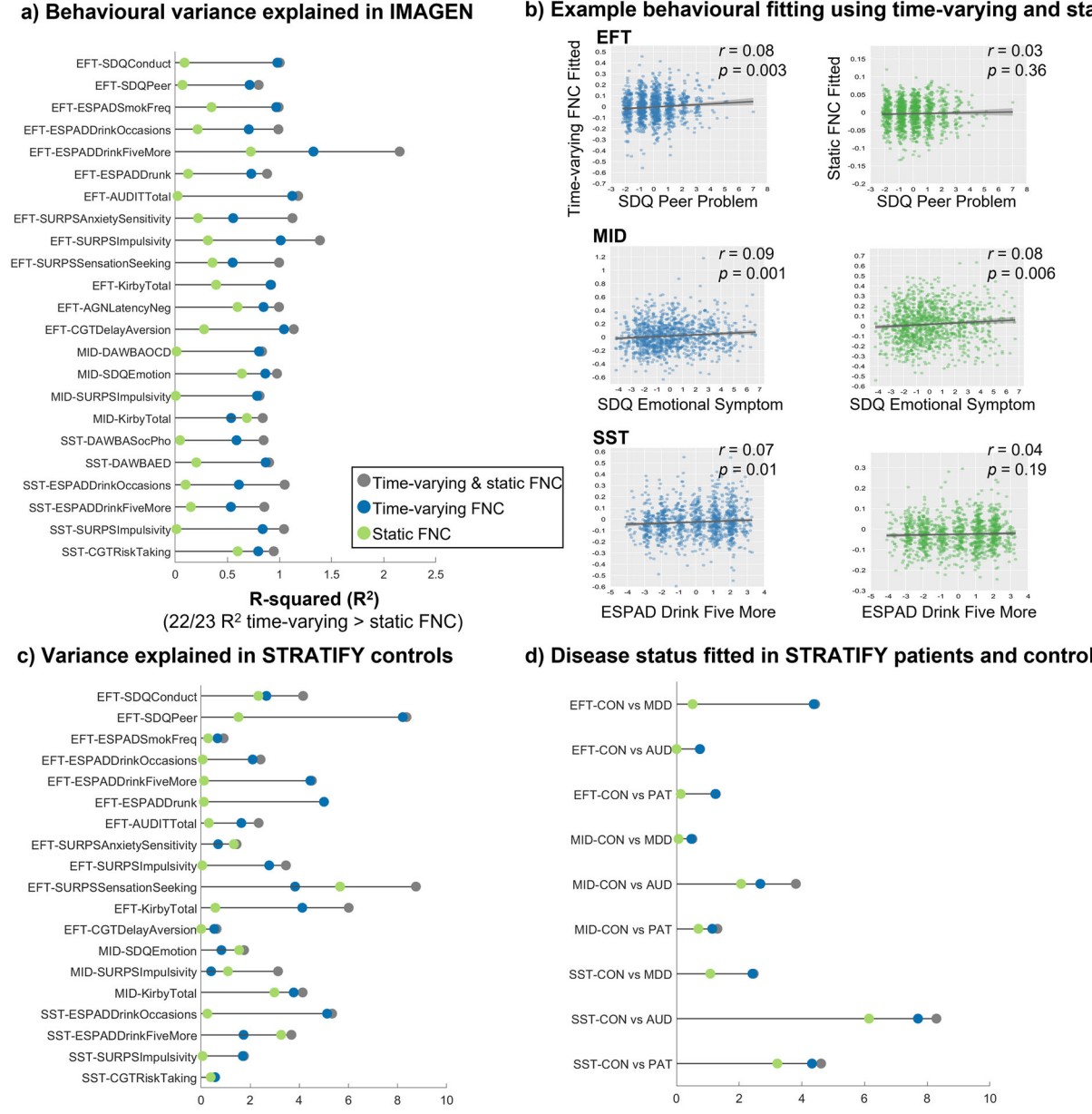

**Fig. 6 | Behavioural variance explained by time-varying FNC and static FNC from the EFT, MID and SST sessions.** The same set of 20 task-specific connectivity identified from the previous section was taken from time-varying FNC states and static FNC. **a** In the IMAGEN cohort, linear regression of 29 behaviours (Table S9) were regressed as a function of time-varying FNC (blue), static FNC (green), or both (gray). Model significance is obtained by comparing the full model with null models (only constant term). Among the 23 significantly fitted behaviours (model significance uncorrected $p_{one-tail} < 0.05$, Table S12), 22 behaviours showed a higher $R^2$ using time-varying FNC than static FNC regression model. **b** As an example, we showed one behavioural fitting using time-varying FNC and static FNC from each

task. **c** The same model fitting and comparison procedure were applied in the STRATIFY healthy controls. For the behavioural items that are available in STRATIFY (Table S9), time-varying FNC has a higher $R^2$ using time-varying FNC in 14 out of 19 behavioural items. **d** Further, the disease status of STRATIFY patients and controls were regressed as a function of time-varying FNC and static FNC using logistic regression. All models showed a numeric higher $R^2$ using time-varying FNC than static FNC regression model. Abbreviations: CON healthy controls, AUD alcohol use disorder, MDD major depressive disorder, PAT all patients. Covariates of age, sex, recruitment sites and head motion were regressed before analysis.

observed score using time-varying FNC was $r = 0.09$, $p = 0.001$, 95% CI = [0.04, 0.15] and $r = 0.08$, $p = 0.006$, 95% CI = [0.02, 0.14] using static FNC. In the SST, variance explained of European School Survey Project on Alcohol and Other Drugs (ESPAD) binge drink score by time-varying FNC was 0.53% compared to 0.15% by static FNC. Correlation between the fitted and observed score is significant for time-varying FNC ($r = 0.07$, $p = 0.01$, 95% CI = [0.02, 0.13]) but not static FNC ($r = 0.04$, $p = 0.19$, 95% CI = [-0.02, 0.10]).

## Validation in an independent dataset
We validated our results in the independent STRATIFY dataset which includes patients with alcohol use disorder (AUD, $n = 125$), major depressive disorder (MDD, $n = 131$) and healthy controls (CON, $n = 183$, Table S13)[36,37]. Patients with MDD were included if they (i) had a Patient Health Questionnaire-9 (PHQ-9) score ≥ 15, and (ii) met the diagnostic criteria for a current major depressive episode. AUD patients were included if they (i) scored ≥ 15 on the Alcohol Use Disorders

Identification Test (AUDIT), and (ii) met the diagnostic criteria for alcohol dependence or harmful alcohol use (Supplementary Information). The MRI acquisition protocol of STRATIFY was identical to the IMAGEN cohort and is described in detail in: https://stratify-project.org/documentation/. Using the sliding window and k-means clustering analysis (Supplementary Information), we derived four FNC states from each session of the STRATIFY cohort (REST, EFT, MID and SST), which showed a high-level of correspondence with the FNC states derived from the IMAGEN resting-state (within-state correlation ranges from 0.80 to 0.96, except for MID state 3, Fig. S14). Overall, the FNC states were reproducible in the independent STRATIFY sample.

Our replication of the correlation between occurrences of FNC state and ongoing task stimuli (Fig. S15) yielded results similar to the IMAGEN dataset (Fig. 3 and Table S8). Specifically, in the EFT, states 1 and 4 positively correlated with faces stimuli, while state 2 positively correlated with non-faces stimuli. MID state 2 negatively correlated with large-win condition, but this was observed only in MDD patients. SST state 3 positively correlated, and state 2 negatively correlated with go too-late errors across all three groups.

In replicating the correlation between FNC state dwell time and task performances (Fig. S10), we confirmed significant associations between SST go too-late errors with dwell time in state 1 ($rho = -0.26$, $p = 0.01$) and state 3 ($rho = 0.31$, $p = 0.003$) within the STRATIFY CON group (Table S14). Among individuals with MDD and AUD, SST go too-late errors were positively correlated with dwell time in state 3 ($rho = 0.28, 0.34$; $p = 0.02, 0.004$ respectively). Group comparisons of task performance showed that MDD patients exhibited lower accuracy and longer reaction time (RT) during the MID task compared to CON, while AUD patients showed only longer RT relative to CON (Fig. S16). For SST task performances, AUD patients had longer RT in response to stop signals, and MDD patients exhibited longer RT for go signals.

As in IMAGEN, task-specific regional FNC (Fig. 5) derived from time-varying FNC was superior to static FNC in explaining behavioural variance in the STRATIFY sample (Fig. 6c-d). While the sample size of STRATIFY being smaller than IMAGEN increases the risk for $R^2$ inflation[38] (see example in Fig. S17), it does not affect the relation of $R^2$ between time-varying and static FNC measurements. Regression analysis in STRATIFY controls revealed that time-varying FNC has a numeric higher $R^2$ using time-varying FNC in 14 out of 19 behavioural items, whereas static FNC only showed a higher $R^2$ in 5 behavioural items (Fig. 6c). For example, in the EFT, variance explained of peer problem from SDQ by time-varying FNC was 8.21% compared to 1.52% by static FNC. In the MID, variance explained of reward delay discounting from MCQ by time-varying FNC was 3.77% compared to 2.98% by static FNC. In the SST, variance explained of ESPAD binge drinking by time-varying FNC was 5.13% compared to 0.25% by static FNC.

### Testing the clinical relevance of time-varying FNC

To investigate the clinical relevance of our model, we used logistic regression to test the performance of time-varying FNC and static FNC in distinguishing disease status (CON vs. MDD, CON vs. AUD, CON vs all patients (PAT)). We found that time-varying connectivity in the EFT explained 4.38% of the variance of MDD, compared to 0.51% in static FNC, which was mainly driven by the time-varying connectivity of state 3 that was significantly different between CON and MDD patients ($t = 3.53$, $p = 0.0005$, Fig. S18). In the SST, there was a high explanation of variance of AUD by time-varying FNC of 7.70% vs. 6.14% by static FNC, consistent with the association with the binge drinking score observed in IMAGEN and STRATIFY controls. Post-hoc analysis showed that time-varying connectivity of states 1, 2, and 4 ($t = 3.74, 3.18, 3.38$, $p = 2.23 \times 10^{-4}$, 0.002, $8.29 \times 10^{-4}$, respectively), as well as in static FNC ($t = 4.32$, $p = 2.21 \times 10^{-5}$), are significantly different between CON and AUD patients (Fig. S18). Time-varying FNC compared to static FNC in the MID explained 0.47% vs. 0.06% in MDD, 2.67% vs. 2.06% in AUD and

1.14% vs. 0.70% in all patients compared to controls. The clinical utility of time-varying FNC was particularly evident in the EFT and SST, where the explained variance was higher in patients with MDD and AUD, respectively, consistent with the superior performance of time-varying FNC in explaining behavioural symptoms in population-based samples.

### Sensitivity analysis

To assess whether the results were influenced by potential site effects (Table S2), we first extracted the four FNC states separately for each of the eight recruitment sites (Fig. S19). We then calculated the spatial correlation of the FNC states and their occurrences between each individual site and the combined data from all sites combined. The results showed that the FNC states derived from individual sites were highly similar to those from the all sites, and their occurrences exhibited moderate to large consistency across sites. Furthermore, we replicated the main analyses within each recruitment site, including: (i) the correlation between FNC state occurrence and task conditions (Fig. S20); (ii) the correlation between FNC state dwell time and task performance (Fig. S21); and (iii) the behavioral variance explained by time-varying and static FNC (Fig. S22). The main findings were reproducible, supporting the robustness of our results across recruitment sites.

To minimize the influence of head motion on our results, we excluded subjects with excessive head motion (mean framewise displacement (FD) > 0.2 mm; see Table S1). Additionally, in the estimation of time-varying FNC, we included the six realignment parameters as covariates in the sliding window analysis to reduce the impact of micro-movements on FNC estimation (Materials and Methods). We evaluated the correlation between FD and state occurrences during each scanning session (Fig. S23), which showed consistently low correlations (median range: −0.046 to 0.078). Furthermore, we included head motion as a covariate in the correlation analyses between state occurrences and task events (Fig. S24), and found that the results from the EFT and SST sessions remained robust. In all between-subject analyses, covariates including age, sex, recruitment site, and head motion (mean FD) were regressed out.

## Discussion

By applying time-varying connectivity analyses to two large, behaviourally well-characterized datasets, we found two sets of complementary brain network features. The first set, a whole-brain network is characterized by the transition between different FNC states according to ongoing task demands, which potentially reflect fluctuation of state of mind such as attention, engagement, motivation and emotion processing. The second set of network features is regional and task-specific. By combining information from these two sets of networks, we found that time-varying FNC compared to the commonly used static FNC explains a higher amount of variance of reinforcement-related behaviours in the general population and is much more sensitive in detecting disease status in patients with depression and alcohol use disorder. These findings offer a more detailed understanding of how dynamic changes in brain connectivity are linked to behaviour and psychopathology, beyond what is captured by static neuroimaging analyses.

We discovered that FNC states are directly linked to basic cognitive processes. While previous studies showed that whole-brain networks are switching between an segregated and integrated state[13] depending on factors such as arousal and task complexity[14,15], the cognitive relevance of FNC states remained unclear. We identified FNC states related to task engagement and motivation (states 1 and 4), which co-occur with faces stimuli and negatively correlate with control stimuli in the EFT. FNC state 1 is negatively correlated with SST go-too late error performances. We also characterized an FNC state representing disengagement (state 2) that is negatively correlated with faces stimuli in the EFT and large-win in the MID. Lastly, we identified an FNC

state linked to vigilance (state 3). Occurrence of FNC state 3 in the EFT and SST linearly increases with scanning time and is related to more go too-late errors in the SST. Our conclusion is supported by the fact that in the only task with immediate feedback, the MID task, FNC state 3 does not occur, suggesting that its task design induces participants to maintain a high level of vigilance. These results were replicated in the independent STRATIFY sample.

We found that FNC states with similar modularity (like FNC state 2 and 4) can still have different associations with task processing. Thus, it is necessary to consider the existence of distinct connectivity patterns, rather than just the level of network integration. FNC state 2 is characterized with more negative connectivity between sensorimotor regions and cognitive control, default mode networks (Fig. S8). The level of disassociation between primary and higher-order regions could explain their different correlations with task stimuli and behaviours[39].

We then demonstrated the behavioural relevance of FNC states in relation to reinforcement-related psychopathology and cognition. Interestingly, FNC state 1 and 4, which show positive association with conditions of interest during tasks, are negatively associated with psychopathology and substance use, whereas FNC state 2 and 3 show positive associations with symptoms (Fig. 4). Behavioural correlates of state 2 include internalizing and externalizing problems from the DAWBA and hopelessness and anxiety sensitivity from the SURPS (Fig. 4b). Behavioural correlates of state 3 include substance abuse and SURPS impulsivity and sensation seeking (Fig. 4c). These correlations demonstrate how basic cognitive processes in the form of FNC may underlie psychopathology and mental health. Their relation to the behavioural domains interrogated in specific neuroimaging tasks is, however, only an indirect one. To identify task-specific connectivity, we measured pairwise connectivity between regional areas in all three reinforcement-related tasks.

Each task evoked a different set of regional FNC which showed task-specific modulation effects. The task-specific FNC are partly among task-induced activation regions, but also involve other brain areas, such as the default mode regions in all three tasks (Fig. 5 and Table S11). For example, previous studies indicated that superior temporal gyrus is an important node in social cognition and has rich connections with visual and emotional processing brain regions[40,41]. Our results show that FNC between superior temporal gyrus and various frontal, occipital regions and default mode areas are specifically coupled with angry versus neutral faces. For the MID, we found connectivity between core regions of reward processing, medial prefrontal cortex[6,42] and frontoparietal, superior temporal regions to be associated with large-win versus no-win conditions. For the SST, connectivity between sensorimotor areas and default mode network, and connectivity between the inferior frontal gyri with other brain regions are associated with stop-success versus stop-failure condition, aligned with the involvement of these regions in the motor inhibition[23,43].

With the state-based regional FNC, we could model the task-specific processing in different mental states, thus better correlating with a cognition or a behaviour of interest, compared to static connectivity (Fig. 6). We showed that time-varying FNC accounts for a numeric larger $R^2$ than static FNC in both IMAGEN and STRATIFY datasets. This result is generalized to regression of disease status in the STRATIFY controls and patients. We hypothesize that the superior performance of time-varying FNC is due to integration of different mental states that are represented by whole-brain FNC states. Global mental states such as motivation and vigilance will influence the association between regional FNC and behaviour[26]. By using regional FNC from different whole-brain FNC states, we improve the behavioural association in both population cohort and clinical samples. Note that the better performance of time-varying FNC is not solely driven by including more regressors (4 time-varying FNC vs 1 static FNC), as adjusted $R^2$ and log-likelihood which balance model accuracy

and model complexity also suggested a better fitting using time-varying FNC (Fig. S13 and Table S12).

Several points need to be considered when interpreting the current results. First, fMRI-based time-varying FNC analysis is not suitable for detecting instantaneous changes induced by event-related design tasks, due to the limited temporal resolution of commonly used fMRI scanning protocols (1–3 s) and the delay of the hemodynamic response. It is well suited to detect FNC changes in block-design tasks or mixed blocked/event-related designs[44]. We need to consider stimulus presentation strategies and the type of questions we could ask using this technique. Other methodical efforts to improve temporal resolution include applying scanning protocol with a short repetition time[45] and using statistical methods that may detect more transit FNC changes[16,17,46]. Second, the underlying biological mechanism of reoccurring FNC states is not fully resolved. Static functional brain organization is, to a large extent, shaped by structural brain connectivity, but also influenced by co-activation patterns across brain regions[12,47]. Some studies showed that dynamic functional brain networks may be subject to neuromodulation[48,49]. Yet, the precise neurobiological mechanism underlying state emergence and transition is still elusive. Third, we used a commonly applied group ICA method to retrieve brain network components[29] (https://trendscenter.org/software/gift/), with the assumption that the spatial extent of components does not change in a scanning session. The assumption is reasonable, as previous studies showed that ICA components have a good correspondence between different sessions and different cohorts[12,50,51]. However, for studies particularly interested in precise localization of brain activation and connectivity in cognitive processing may consider other approaches allowing both spatial and temporal changes of a network. Some of the efforts that aim to capture spatiotemporal dynamics of brain regional activity can be found here[52,53]. Fourth, resting-state fMRI were preprocessed using the FSL and ANTs, while task-based fMRI were preprocessed with the SPM (Supplementary Information). All preprocessing scripts and documentation are openly available at: https://github.com/imagen2/imagen_processing/. Despite the use of different pipelines, we found that the four FNC states were highly similar between resting-state and task-based fMRI sessions (Fig. 2a), suggesting that the identified FNC states are robust to preprocessing variations.

Taken together, our findings improve analyses of brain network connectivity by taking into account two dimensions: The first dimension represents whole-brain network switches between several intrinsic network organizations that support flexible cognitive demands. The second dimension measures regional time-varying FNC that reflect different aspects of reinforcement-related behaviours. By combing these two dimensions, we improve our understanding of spatiotemporal dynamics of brain functional network and provide better neural correlates of participants' mental states and psychopathology, compared to one-dimensional static fMRI network analyses.

## Methods

### Participants

The dataset is part of a population-based neuroimaging-genetics study: the IMAGEN cohort[1], aiming to identify neurobiological correlates of reinforcement related behaviours in adolescents followed from 14 to 22 years old. For this study, we analyzed neuroimaging and behavioural data from 1417 participants at mean age of $19.09 \pm 0.76$ years old, including 733 female subjects (51.73%). Participants were recruited from eight research centers in the United Kingdom, German, France and Ireland. Ethics approval was obtained at each site by local research ethics committee. Written consent was obtained from each participant. After a multi-step quality control pipeline (Table S1), we analyzed data from 991 subjects with resting-state fMRI, 1263 participants with EFT, 1221 with MID and 1218 subjects with SST (Table S2).

## Neuropsychological assessments

We included behaviour questionnaires which measure mental disorder symptoms, substance use and neuropsychological functions of adolescents. Symptom severity were assessed using the Development and Well-Being Assessment (DAWBA)[54] and Strengths and Difficulties Questionnaire (SDQ)[55], which covers common emotional, behavioural and hyperactivity symptoms in adolescents. We used self-rated questions (symptom counts) of the DAWBA and subscale scores of the SDQ. Depression symptom was additionally assessed with the Adolescent Depression Rating Scale (ADRS)[56]. Smoking and drinking behaviours were measured using the European School Survey Project on Alcohol and Other Drugs (ESPAD)[57] and Alcohol Use Disorders Identification Test (AUDIT)[58]. Substance Use Risk Profile Scale (SURPS) measures personality risk for substance abuse, including four dimensions: hopelessness, anxiety sensitivity, impulsivity and sensation seeking[59]. Neuropsychological functions such as reward sensitivity, affective processing and cognitive control were assessed with the following questionnaires: Monetary-Choice Questionnaire (MCQ) was administered to assess delay discounting i.e. the tendency to choose sooner but smaller over later but larger rewards[60]. Affective Go-Nogo task (AGN) and Cambridge Gambling Task (CGT) were completed with the computerized Cambridge Neuropsychological Test Automated Battery (CANTAB, www.cambridgecognition.com), which assess the perceptual bias in facial emotion perception and risk-taking behaviour outside a learning context. To avoid multi-collinearity, we removed behavioural items with a correlation coefficient larger than 0.8. We further removed near-zero variance predicator, which has a unique value less than 5%, i.e. more than 95% of participants having the same value[61]. In total, 29 behavioural items assessing reinforcement-related psychopathology and functions were included in the analysis and listed in the Table S9. Distributions of all participants' scores were plotted on Fig. S12.

## MRI acquisition and preprocessing

Neuroimaging data was collected on 3 T scanners at eight study centers with different manufacturers (Siemens: 4 sites, Philips: 2 sites, General Electric: 1 site, and Bruker: 1 site). Standardized hardware for visual and auditory stimulus presentation (Nordic Neurolabs, Norway) was used at all sites. Scanning parameters were harmonized across sites prior to study. For each participant, we collected a high-resolution T1-weighted MRI (T1) scan using the Magnetization Prepared Rapid Acquisition Gradient Echo (MPRAGE) sequence. Blood-oxygen-level-dependent (BOLD) functional images were acquired with gradient-echo, echo-planar imaging (EPI) sequence during resting-state and three cognitive tasks: emotional faces task (EFT)[27], monetary-incentive delay task (MID)[28] and stop-signal task (SST)[23]. Detailed task protocol was provided in the Supplementary Information. Scanning parameters of T1-weigthed images are: repetition time (TR) = 2300 ms; echo time (TE) = 2.8 ms; flip angle (FA) = 8°; isotropic voxel size 1.1 mm; 256 × 256 × 160 matrix; sagittal slice plane. Acquisition parameters of functional images are: TR = 2,200 ms, TE = 30 ms, FA = 75°; 64 × 64 × 40 matrix, voxel size = 3.4 × 3.4 × 2.4 mm, slice gap = 1 mm. In this study, resting-state fMRI data was only used to provide a set of spatial templates from the group independent component analysis for three task sessions.

Resting-state fMRI data were pre-processed with FMRIBs Software Library (FSL version 5.0.9) and Advanced Normalization Tools (ANTs version 1.9.2). Non-brain tissue was removed (FSL BET) and images were corrected for head motion (FSL MCFLIRT), then spatially smoothed using a 4 mm FWHM Gaussian kernel. In addition, artifact components were removed for each data set using an automatic classification algorithm (ICA-AROMA v0.3). The resulting cleaned data set was detrended and normalized to MNI standard space using the custom EPI template (ANTs). Pre-processed data were resliced to 3 mm isotropic voxels. Quality control was carried out in the same manner as

task-based fMRI data, excluding subjects with incomplete demographic or scanning information, excessive head motion (mean framewise displacement > 0.2 mm); or with unsuccessful normalization. Finally, resting-state datasets from 991 subjects entered the group independent component analysis.

Task-based fMRI scans were pre-processed using Statistical Parametric Mapping (SPM12, http://www.fil.ion.ucl.ac.uk/spm/). Pre-processing steps include: non-brain tissue removal, slice-timing correction, head movement realignment using a rigid body transformation, and images were non-linearly warped on the MNI space using a customized EPI template. Normalized images were smoothed using 5 mm FWHM Gaussian kernel. Pre-processed data were resliced to 3 mm isotropic voxels.

To ensure quality of data for further analysis, we excluded: 1) subjects with incomplete demographic information or task onset files simultaneously recorded during scanning; 2) subjects with excessive head movement (mean framewise displacement > 0.2 mm); 3) subjects with unsuccessful normalization (using the criterion of a correlation between individual mask and group mask <0.85)[62]. In total, we analyzed 1263 participants with EFT, 1221 subject with MID and 1218 subjects with SST neuroimaging data (Tables S1–S2).

## Group independent component analysis and postprocessing

Group independent component analysis (ICA) was applied on pre-processed functional images using the GIFT software (https://trendscenter.org/software/gift/). We selected a relative high model order of 100 independent components to achieve a functional parcellation of brain corresponding to known anatomical and functional segmentations[12,18,62,63]. To ensure we have a set of comparable independent components (ICs) from ICA among the three task-based fMRI sessions, we first applied ICA to the resting-state dataset, and used aggregated group spatial maps from the resting-state as a reference to guide ICA for task fMRI data[64]. Specifically, for the resting-state dataset, we first applied a two-step data reduction to reduce the computational demand: 1) individual subject data were decomposed into 110 components using standard principal component analysis (PCA); 2) individual data were concatenated into one group, and further reduced to 100 components using the expectation maximization (EM) algorithm. Then Infomax ICA algorithm[65] was applied to the reduced group data to obtain aggregated group spatial maps from all subjects. Infomax ICA was repeated 10 times to avoid local minima in ICASSO. Next, aggregated group spatial maps were used as a reference to guide subject-level ICA for each task session. Above procedure results in 100 ICs of each task session corresponding to the resting-state spatial maps for every subject.

We selected a set of ICs which exhibit peak activations in gray matter, low spatial overlap with known vascular, ventricular, motion, and susceptibility artifacts, and corresponding time-courses dominated by low-frequency fluctuations[18]. Component selection was confirmed by two researchers (XC and ZNF, Table S3). Time-courses of the selected components underwent post-processing to 1) remove linear, quadratic, and cubic trends; 2) regress 6 head motion parameters estimated during realignment; 3) de-spike detected outliers; 4) low-pass filter with a cut-off frequency of 0.18 Hz. The low-pass filter was chosen to remove physiological sources (heartbeat, respiration) and motor response in tasks which are higher than 0.18 Hz[21,66].

## Time-varying functional network connectivity (FNC) estimation

Time-varying FNC was estimated between each pair of the selected components in a segment of scanning period (time window), instead of using the entire scanning session. The tapered time window was created by convolving a rectangle with a Gaussian kernel (sigma = 3 TR) to obtain a time-specific, whole-brain connectivity profile. Window length is 8TR or 17.6 s for the REST, EFT and SST sessions, and 4TR or 8.8 s for the MID to accommodate the block duration of MID (~ 10 s).

Then the time window was moved forward 1 TR to get the next whole-brain connectivity matrix and repeated until the end of a scanning session. To estimate full covariance matrix using relatively a short time segment, we used L1 penalized graphical LASSO method to estimate regularized precision matrix (inverse covariance matrix) and calculated the covariation matrix from the precision matrix[67]. This results in a series of windowed FNC matrices for every subject of each data session. The number of windows equals to the scanning time points (unit of TR) minus a window length. For every subject, the scanning period consists of 197 EFT scans and 191 MID scans. The stop-signal task was designed to automatically adjust to subjects' performance (to ensure every subject has, on average, 50% successful rate to respond to stop signals). Therefore, the scanning length of stop-signal task varies from 310 to 350 scans across subjects. We included the first 310 scans to ensure the number of time windows are comparable between subjects. As previous studies suggest that task-induced brain activation might inflate estimation of connectivity by imposing co-occurring changes in BOLD signal due to task stimuli presentation[68,69], we regressed task-evoked activation from BOLD signal by including task design matrix and the six realign parameters as covariates prior to the sliding window analysis. To show the robustness of time-varying FNC estimation[19,70], we repeated the time-varying FNC analysis using different window lengths of 8.8 s (4TR), 17.6 s (8TR), 35.2 s (16TR) and 70.4 s (32TR) (Figs. S5, 6), and calculated spatial correlation (Pearson's correlation) between FNC states with different sliding window lengths (Table S5).

## FNC states derived from clustering on time-varying FNC matrices

To identify reoccurring whole-brain FNC states, we applied the $k$-means clustering analysis to windowed FNC matrices for each session. Similar to the identification of EEG microstate[71,72], we first identified the windows which have local maxima in connectivity variance across component pairs (subject exemplars)[18]. Clustering analysis was applied to the set of subject exemplars with 150 repetitions. The resultant group centroid was used as the initial centroid for clustering analysis on all windowed FNC matrices. Using the elbow criterion[31], the optimal cluster numbers were four for three sessions (REST, EFT and MID), and five for the SST (Fig. S3). We set four as the predefined clustering number in the main text. To evaluate the influence of clustering numbers on FNC states estimation, we repeated the clustering analysis using $k$ varying from 2 to 5 (Figs. S7-8) and calculated spatial correlations (Pearson' correlation) between FNC states with different clustering numbers (Table S6).

## Network graph metrics comparison of FNC states

We quantitatively compared level of network integration and segregation of the derived FNC states using the Brain Connectivity Toolbox (BCT)[73]. A commonly used network segregation measure is modularity, which quantifies the degree to which a network can be divided into several subnetworks with maximal within-group connectivity and minimal between-group connectivity. The Louvain modularity algorithm was applied to detect the optimal community structure of the network and calculate modularity[33]. In addition, we calculated the participation coefficient, which assesses the diversity of connections between subnetworks of nodes (i.e. regions). The participation coefficient was calculated for positive and negative FNC separately and averaged across all regions. As network metrics can be influenced by basic characteristics, such as connection weights and distributions, we compared the modularity and participation coefficient of each FNC state with metrics calculated from 100 randomized networks. The randomized networks preserved the degree distribution of positive and negative FNC while rewiring the connections at random. Normalized modularity and participation coefficient were calculated by dividing empirical values by randomized metrics.

## Whole-brain FNC state occurrences correlated with task events and performances

We further investigated whether the FNC states co-occur with ongoing task events (i.e. task stimuli and subjects' performance). For EFT and MID tasks, experimental designs (stimuli sequence and onset time) are the same across all participants, and thus we calculate Pearson's correlation between occurrences of task events and FNC state at the group level. The occurrence of a task event was generated using stimuli onsets and duration convolved with canonical hemodynamic response function (HRF). The occurrence of a state at the group level was calculated as the proportion of subjects having the given state at a particular time point. Correlation coefficients were compared between conditions using Pearson and Filon's z-statistic from the R package 'cocor'[34] (Fig. 3 and Table S8). For the SST, as stimuli onsets differ across subjects, we calculated the partial correlations between state occurrences and a given task stimulus, while controlling for other types of stimuli at the individual level. Correlation coefficients were compared to 0 using a one-sample $t$-test to determine their significances. Multiple testing effect was controlled using the false discovery rate (FDR, $q < 0.05$) for number of task events × 4 FNC states.

Next, we examined correlation between FNC states' dwell time with task performances in the MID and SST across participants (EFT does not require responses). Spearman's rank correlation was performed as task performances do not follow normal distribution (Fig. S9). MID performances include accuracy (ACC) and response time (RT) of large-win, small-win and no-win trials. SST performances include go response time, stop response time, percentage of go-wrong error, go too-late error and stop-failure error. Stop response time was calculated as mean go signal response time minus mean stop signal delay per subject (Fig. S10).

## Sparse partial least squares (sPLS) between FNC state dwell time and reinforcement-related behaviours

To test if FNC state dwell time (the average length of a subject having a certain state)[30] is related to individual differences in reinforcement-related behaviours (Fig. S12 and Table S9), we applied a multiple hold-out framework for sPLS analysis using the CCA/PLS Toolkit[35]. Data were split into a training and test set (20%), and the training set was further divided into a training and validation set (20%) to select the best hyperparameter (L1 penalty). The statistical significance of the test set association was determined by 5000 times permutation inference ($p < 0.05$). The random data split procedure was repeated three times to improve model robustness, and the split with the highest test set association was reported in the Results (Fig. 4 and Table S10). Covariates of age, sex, recruitment sites and head motion (mean framewise displacement) were regressed before analysis.

## Correlation between pair-wise FNC and task events

To examine whether pair-wise FNC correlated with task events and show task-specific effect, we calculated the Pearson correlations between strength of pair-wise FNC and occurrence of task conditions within a task session for every subject. For the EFT, we compared correlation coefficients of FNC with angry faces versus neutral faces condition using one-sample $t$-test. Resultant $t$-statistics reflects how strongly the tested FNC fluctuate with angry versus neutral faces conditions. For the MID, we compared correlations between large-win and no-win conditions to examine whether the tested FNC showing reward sensitivity effect. For the SST, correlations are compared between stop-success and stop-failure conditions to test motor inhibition effect (Fig. 5). Next, we performed a specificity analysis to examine whether connections showing the highest $t$-statistics in one task also have high $t$-statistics in the other two tasks. For example, we selected the 20 FNC with the largest $t$-statistic of angry vs neutral faces conditions from EFT, and calculated their $t$-statistic in the MID or SST conditions. If the $t$-statistic is high, it suggests that the top 20 FNC also

distinguish MID and SST conditions. Otherwise, it suggests that the top 20 FNC are specifically modulated by EFT conditions. This analysis was performed for all three tasks (Table S11).

## Behavioural variance explained by time-varying FNC and static FNC

To examine the variance explained for reinforcement-related behaviours (Table S9) by regional time-varying FNC and static FNC, we extracted the 20 task-specific FNC (Fig. 5d) from the four FNC states of each participant (Fig. 2a shows the cluster centroid of each FNC state). In parallel, we derived the same 20 task-specific regional FNC from static connectivity analysis (i.e., to calculate a connectivity strength using the time course of an entire scanning session, rather than from a segment of time window). To reduce the number of behavioural fitting parameters, we averaged the connectivity strength of the 20 task-specific regional FNC. Therefore, we obtain four state-based time-varying FNC, and one static FNC for every participant. Linear regression models were fitted for every behaviour with the time-varying FNC, static FNC or both (full model). We focused on behaviours that can be fitted using both time-varying FNC and static FNC (full model, model significance $p_{\text{one-tail}} < 0.05$, uncorrected). Among these behaviours, we compared variance explained ($R^2$), adjusted $R^2$ and log-likelihood to assess model fitting performance (Table S12). A higher $R^2$ and log-likelihood indicate a better fitting. We showed one behavioural fitting from each task using time-varying FNC and static FNC as an illustration (Fig. 6).

## STRATIFY cohort

The STRATIFY cohort[36,37] recruited patients (ages 19-25) with alcohol use disorder (AUD), major depression disorder (MDD) and healthy controls (CON) from three recruitment sites in Berlin, London and Southampton, of which Berlin and London are also recruitment sites for IMAGEN. For the current study, we included participants with AUD ($n = 125$), MDD ($n = 131$), and CON ($n = 183$).

Patients with MDD were included if they (i) exhibited current and acute depressive symptoms, with a Patient Health Questionnaire-9 (PHQ-9) score ≥ 15, and (ii) met the diagnostic criteria for a current major depressive episode, confirmed by scoring "1" on at least one of the following M.I.N.I. modules: MINI_A_MDEC, MINI_A_MDEMFC, or MINI_A_MDER.

Patients with AUD were included if they (i) scored ≥ 15 on the Alcohol Use Disorders Identification Test (AUDIT), and (ii) met the diagnostic criteria for alcohol dependence or harmful alcohol use, confirmed by scoring "1" on the MINI_J_AAC or MINI_J_ADC modules.

Healthy controls were excluded if they met criteria for any of the above diagnoses. Demographic information is provided in Table S13. MRI acquisition parameters and task designs in STRATIFY were harmonized with the IMAGEN cohort to maximize comparability between datasets. Additional details on study protocols are available at: https://stratify-project.org/.

## Replication of FNC states in the STRATIFY cohort

Resting-state and the three task fMRI data (EFT, MID, SST) were preprocessed with fMRIPrep 20.2.3[74]. We then applied guide ICA for task fMRI data using group spatial maps from the IMAGEN resting-state as a reference. Time-varying FNC and four FNC states were derived in the same procedure described above. We calculated spatial correlation between STRATIFY FNC states and those obtained from IMAGEN resting-state using Pearson's correlation.

Next, we examined cognitive relevance of the FNC states in the STRATIFY by correlating FNC state occurrences with task stimuli and task performances as in the main analysis. Pearson's correlation was performed between occurrences of FNC state and task stimuli occurrence of the EFT and MID session. Spearman's correlation was performed between dwell time of FNC state and performances of the MID

and SST across participants. Multiple comparison effects were controlled using FDR correction (number of tested tasks events × number of FNC states).

Finally, we tested whether task-specific regional FNC derived from time-varying FNC can better fit behaviours in STRATIFY controls and distinguish disease status (CON vs. MDD or CON vs. AUD) than static FNC. In STRATIFY controls, 19 behavioural items were available as IMAGEN cohort (Table S9). We compared the model fitting parameters $R^2$, adjusted $R^2$ and log-likelihood in the time-varying FNC and static FNC regression model. In STRATIFY controls and patients, we performed a logistic regression using time-varying FNC and static FNC on disease status (0 controls, 1 patients).

## Reporting summary

Further information on research design is available in the Nature Portfolio Reporting Summary linked to this article.

## Data availability

Data from the IMAGEN and STRATIFY projects are accessible upon request through: https://imagen-project.org. Source data are provided with this paper.

## Code availability

The core code used to run the analyses reported in this study can be found at: https://github.com/xchang007/tvFNC_IMAGEN.git.

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

## Acknowledgements

This work received support from the following sources: National Key R&D Program of China (2023YFE0199700 to XC, 2019YFA0709502 to JF), National Natural Science Foundation of China (82102138 to XC, W2541022 to GS, 82150710554 to GS, T2122005 to TJ), National Science and Technology Major Project (2025ZD0215100 to GS), 111 Project (B18015 to JF), Shanghai Municipal Science and Technology Major Project (2018SHZDZX01 to JF), Shanghai Sailing Program (21YF1402400 to XC), the Shanghai Pujiang Project (18PJ1400900 to TJ), the European Union-funded FP6 Integrated Project IMAGEN (Reinforcement-related behaviour in normal brain function and psychopathology) (LSHM-CT-2007-037286 to GS), the Horizon 2020 funded ERC Advanced Grant 'STRATIFY' (Brain network based stratification of reinforcement-related disorders) (695313 to GS), Horizon Europe 'environMENTAL', grant no: 101057429 to GS, UK Research and Innovation (UKRI) Horizon Europe funding guarantee (10041392 and 10038599) to SD, Human Brain Project (HBP SGA 2, 785907, and HBP SGA 3, 945539) to GS. The German Center for Mental Health (DZPG), the Bundesministerium für Bildung und Forschung (BMBF grants 01GS08152; 01EV0711; Forschungsnetz AERIAL 01EE1406A, 01EE1406B; Forschungsnetz IMAC-Mind 01GL1745B), the Deutsche Forschungsgemeinschaft (DFG project numbers 458317126 [COPE] to GS, 186318919 [FOR 1617], 178833530 [SFB 940], 386691645 [NE 1383/14-1], 402170461 [TRR 265], 454245598 [IRTG 2773]), the Medical Research Foundation and Medical Research Council (grants MR/R00465X/1 and MR/S020306/1) to SD, the National Institutes of Health (NIH) funded ENIGMA-grants 5U54EB020403-05, 1R56AG058854-01 and U54 EB020403 as well as NIH R01DA049238, the National Institutes of Health, Science Foundation Ireland (16/ERCD/3797) to ALWB and RW. Further support was provided by grants from: the Eranet Neuron (Grant ANR-18-NEUR00002-01– ADORe to JLM); Agence Nationale de la Recherche (Grant ANR-12-SAMA-0004 -GeBra to JLM); Assistance-Publique Hôpitaux-de-Paris and INSERM (interface grant to JLM); Paris Descartes University (Grant collaborative-project-2010 to JLM); Paris Sud University (Grant IDEX-2012 to JLM); Fondation de l'Avenir (Grant AP-RM-17-013 to JLM); Fondation de France (Grant 00081242 to JLM); Fédération pour la Recherche sur le Cerveau, and Fondation pour la Recherche Médicale (Grants DPA20140629802 and ADOLIMIS DPP20151033945 to JLM); the Ile-de-France Region (Action 16700103 -grant to QIM– VEAVE, n 23002745–23002747 to JLM).

## Author contributions

Conceptualization: G.S., V.D.C. Methodology: X.C., T.J., Z.F., S.X. Investigation: X.C., T.J., Z.F., S.X. Data acquisition: T.B., A.L.W.B., S.D., H.F., A.G., H.G., P.G., A.H., R.B., J.L.M., M.L.P.M., E.A., F.N., D.P.O., V.F., L.P., S.H., C.B., M.N.S., N.V., H.W., R.W., G.S.; Data were acquired and provided by the IMAGEN Consortium and environMENTAL Consortium Supervision: G.S., V.D.C., J.F. Writing—original draft: X.C., G.S. Writing—review and editing: G.S., T.J., Y.X., C.X., J.Y.Z., M.C., J.Z., J.F.

## Funding

## Competing interests

Authors declare that they have no competing interests. Dr Banaschewski served in an advisory or consultancy role for AGB Pharma, eye level, Infectopharm, Medice, Neurim Pharmaceuticals, Oberberg GmbH and Takeda. He received conference support or speaker's fee by Janssen-Cilag, Medice and Takeda. He received royalties from Hogrefe, Kohl-hammer, CIP Medien, Oxford University Press; the present work is unrelated to these relationships. Dr Barker has received honoraria from General Electric Healthcare for teaching on scanner programming courses. Dr Poustka served in an advisory or consultancy role for Roche and Viforpharm and received speaker's fee by Shire. She received royalties from Hogrefe, Kohlhammer and Schattauer. Dr Bokde has received honoraria from Elsevier, Inc. for editorial work. The present work is unrelated to the above grants and relationships.

## Additional information

[1]Institute of Science and Technology for Brain-Inspired Intelligence (ISTBI), Fudan University, Shanghai, China. [2]Key Laboratory of Computational Neuroscience and Brain-Inspired Intelligence, Fudan University, Ministry of Education, Shanghai, China. [3]Centre for Population Neuroscience and Stratified Medicine (PONS), ISTBI, Fudan University, Shanghai, China. [4]Zhangjiang Fudan International Innovation Center, Shanghai, China. [5]MOE Frontiers Center for Brain Science, Fudan University, Shanghai, China. [6]Tri-institutional Center for Translational Research in Neuroimaging and Data Science (TReNDS), Georgia State University, Georgia Institute of Technology, Emory University, Atlanta, GA, USA. [7]Department of Psychological and Cognitive Sciences, Tsinghua University, Beijing, China. [8]Department of Child and Adolescent Psychiatry and Psychotherapy, Central Institute of Mental Health, Medical Faculty Mannheim, Heidelberg University, Mannheim, Germany. [9]Discipline of Psychiatry, School of Medicine and Trinity College Institute of Neuroscience, Trinity College Dublin, Dublin, Ireland. [10]Social, Genetic, Developmental Psychiatry (SGDP) Centre, Institute of Psychiatry, Psychology & Neuroscience, King's College London, London, UK. [11]Institute of Cognitive and Clinical Neuroscience, Central Institute of Mental Health, Medical Faculty Mannheim, Heidelberg University, Mannheim, Germany. [12]Department of Psychology, School of Social Sciences, University of Mannheim, Mannheim, Germany. [13]NeuroSpin, CEA, Université Paris-Saclay, Gif-Sur-Yvette, France. [14]Departments of Psychiatry and Psychology, University of Vermont, Burlington, VT, USA. [15]Sir Peter Mansfield Imaging Centre School of Physics and Astronomy, University of Nottingham, University Park, Nottingham, United Kingdom. [16]Department of Psychiatry and Psychotherapy, University of Tübingen, and German Center for Mental Health (DZPG), site Tübingen, Germany. [17]Physikalisch-Technische Bundesanstalt (PTB), Braunschweig and Berlin, Berlin, Germany. [18]Institut National de la Santé et de la Recherche Médicale, INSERM U 1299 "Trajectoires développementales & psychiatrie", University Paris-Saclay, Ecole Normale Supérieure Paris-Saclay, CNRS, Gif-sur-Yvette, France. [19]EPS Barthélémy Durand, Etampes, France. [20]AP-HP. Sorbonne University, Department of Child and Adolescent Psychiatry, Pitié-Salpêtrière Hospital, Paris, France. [21]Institute of Medical Psychology and Medical Sociology, University Medical Center Schleswig-Holstein, Kiel University, Kiel, Germany. [22]Department of Child and Adolescent Psychiatry, Center for Psychosocial Medicine, University Hospital Heidelberg, Heidelberg, Germany. [23]Department of Systems Neuroscience, University Medical Center Hamburg-Eppendorf, Hamburg, Germany. [24]Department of Psychiatry and Psychotherapy, Technische Universität Dresden, Dresden, Germany. [25]Centre for Population Neuroscience and Stratified Medicine (PONS), Department of Psychiatry and Neuroscience, Charité-Universitätsmedizin Berlin, Berlin, Germany. [26]Department of Psychiatry and Psychotherapy CCM, Charité-Universitätsmedizin Berlin, Berlin, Germany. [27]School of Psychology and Global Brain Health Institute, Trinity College Dublin, Dublin, Ireland. [28]Shanghai Center for Mathematical Sciences, Shanghai, China. [29]Department of Computer Science, University of Warwick, Coventry, UK. [30]National Center for Neurologic Disorders, Huashan Hospital, Fudan University, Shanghai, China. [31]German Centre for Mental Health (DZPG), Brandenburg, Germany. [32]Department of Psychiatry, Cambridge University, Cambridge, UK. [33]These authors contributed equally: Xiao Chang, Tianye Jia. ✉e-mail: xchang@fudan.edu.cn; gunter.schumann1961@gmail.com

## the IMAGEN Consortium

**Tobias Banaschewski** [8], **Arun L. W. Bokde** [9], **Sylvane Desrivières** [10], **Herta Flor** [11,12], **Antoine Grigis** [13], **Hugh Garavan** [14], **Penny Gowland** [15], **Andreas Heinz** [16], **Rüdiger Brühl** [17], **Jean-Luc Martinot** [18,19], **Marie-Laure Paillère Martinot** [18,20], **Eric Artiges** [18,19], **Frauke Nees** [8,11,21], **Dimitri Papadopoulos Orfanos** [13], **Vincent Frouin** [13], **Luise Poustka** [23], **Sarah Hohmann** [8], **Christian Baeuchl** [23], **Michael N. Smolka** [23], **Nilakshi Vaidya** [24], **Henrik Walter** [25], **Robert Whelan** [26], **Jianfeng Feng** [1,2,4,5,27,28], **Vince D. Calhoun** [6] & **Gunter Schumann** [3,24,29,30,31]

## EnvironMENTAL Consortium

**Tobias Banaschewski** [8], **Sylvane Desrivières** [10], **Andreas Heinz** [16], **Frauke Nees** [8,11,21], **Nilakshi Vaidya** [24], **Henrik Walter** [25], **Vince D. Calhoun** [6] & **Gunter Schumann** [3,24,29,30,31]

