## [Transparent Peer Review file · Nature Communications]

Neurocognitive characterization of behaviour and mental illness through time-varying brain network analysis

Corresponding Author: Dr Xiao Chang

Parts of this Peer Review File have been redacted as indicated to maintain patient confidentiality.

Version 0:

Reviewer comments:

Reviewer #1

(Remarks to the Author)

The manuscript by Chang et al. investigates time-varying functional network connectivity (FNC) during resting-state and three reinforcement-related tasks across 1,417 participants. The authors identify four consistent FNC states, showing that these states are sensitive to task performance and psychopathology. Overall, the manuscript presents valuable findings with solid validation in an independent dataset, providing novel insights into how brain networks adapt to support cognitive tasks and suggesting potential biomarkers for assessing individual psychopathology risk. However, I have several suggestions that could further strengthen the manuscript:

1. Since the imaging data were collected from multiple sites, it would be important to investigate whether site effects influenced the FNC states and subsequent results. If significant site effects are found, harmonization techniques should be applied to address this issue.
2. Could the authors provide more explanation as to why modularity and participation coefficient are used to represent brain network segregation and integration? These two metrics are typically negatively correlated, which may fail to capture an optimized network with both segregation and integration. Alternatively, using metrics like Cp/Lp or local/global efficiency may better characterize this issue.
3. The MID task showed a low correlation with the clustering centroid of state 3 for resting-state and other task, suggesting a unique connectivity pattern in the MID. However, it may not be appropriate to force the MID state into state 3. Did the authors consider conducting clustering on the pooled matrix across all tasks (REST, EFT, MID, and SST)? This approach could better identify both common and task-specific FNC states.
4. In the independent dataset, the authors observe similar patterns in the four time-varying FNC states and their associations with task performance, which is consistent with the main results. Given that this dataset includes participants with clinical conditions, it would be valuable to investigate whether these individuals exhibit altered FNC values, dynamic characteristics, or behavioral relevance compared to the primary findings. This additional analysis could provide deeper insight into the relationship between time-varying FNC and mental health.
5. The manuscript discusses the classification capability of time-varying FNC between patients and healthy controls. However, it would be helpful if the authors could clarify which specific FNC states exhibit stronger predictive power for distinguishing between these groups.
6. In the section "Regional FNC Involved in Task-Specific Cognitive Processes," I recommend that the authors expand on the specific regions associated with each task, as well as those shared across the three tasks. This would provide a clearer understanding of the task-specific brain regions involved in each cognitive process.
7. Including detailed clinical information about the participants from the independent STRIFY cohort would be valuable. This should include diagnostic criteria, symptom severity, and task performance scores, along with behavioral questionnaire data. Such information would enable a more comprehensive comparison with the main findings.
8. The authors mention different preprocessing approaches for resting-state and task-state fMRI data. It would be helpful to explain whether these preprocessing steps might have influenced the results and whether the approach was consistent across tasks.
9. Time-varying FNC is known to be affected by head motion. A more thorough discussion of head motion in this study would be beneficial. This could include applying "scrubbing" to further mitigate motion artifacts or analyzing the correlation between head motion and FNC.
10. To improve consistency and readability, I recommend formatting all tables in the manuscript as three-line tables.

11. In Figure 2, could the authors clarify why the percentages for the four FNC states do not sum to 100% in the resting-state fMRI and EFT conditions?

12. In Figure 6, it would be helpful to explain what the gray dots represent.

(Remarks on code availability)

The code provided is well-written and reproducible, allowing for the analysis reported in the manuscript to be conducted. However, the README file lacks sufficient instructions and clarity in its presentation.

Reviewer #2

(Remarks to the Author)

The paper was well written, interesting, and lays out a rigorous set of analyses with an impressive replication. My biggest point of feedback is the methods are very dense which lead me to some confusion between, "states" that were found across tasks were being used as predictors or and the regional FNCs (which aren't yet clear to me how they are calculated) and other ways functional connectivity used is such as calculating graph metrics. This seems to be a crux of the paper so I have made several suggestions where the authors could clarify the methods, as well as other points of clarification.

The introduction is clear and well written. I do wonder about why the authors characterize the three tasks as "reinforcement tasks" could the authors explain that?

Since the sample is bordering on youth/adult age range and clarification if these were all adults in this sample would be helpful

Can the authors provide rationale for smoothing the task data (5 mm) more than the rest data (4 mm)?

Since the authors are using ICA for network definition, can they clarify what the nodes are for the graph metrics calculations of modularity and PC? Were these run voxelwise? Were the networks used to calculate PC the networks that were derived from the Louvain algorithm?

When the authors are describing the correlation between pairwise FNCs and task events (Figure 5) it's not clear; a. What pairwise FNC refers to, are these pairwise correlations between each ICA component making up each state? Or are these pairwise correlations between networks identified using the Louvain algorithm? Also, how were the regional network components given the labels that appear in Figure 5? Looks like some of this is covered in figure S2 but I missed it in the methods.

Further, it isn't clear which states any of the regional FNCs are from? Or are they not from specific states but averaged along the whole task?

I found the section describing the time-varying vs. static FNC prediction a little confusing, it's possible 'state' is being used to mean both unique FNC pattern and task in this section as in "The 20 task specific FNC was averaged to derive four state-based time-varying FNC and one static FNC for every participant." Are these 4 different FNC's one for each task? I am also confused by what metric is being used to predict reinforcement related behaviors, is it the average FC of that FNC? How does this capture the time-varying aspect?

The discussion felt a little long and repetitive. It would help the reader if the authors slimmed down the results summarizing.

In the discussion when the authors state - "These discoveries represent a major refinement in mechanistic characterizations of behaviour and psychopathology compared to standard neuroimaging analyses." I'm not sure what the mechanism is they are referring to or how it has been refined, I would suggest either laying out the mechanism or avoiding such language.

Some of the figures such as Figure 3 and 4 and in many of the supplementary figures, the individual plots are very tiny and not legible. Please increase font and plot sizes or consider breaking them into several figures.

(Remarks on code availability)

I reviewed the github with the code, I did not attempt to use any of the code myself. It appears as though the code is well commented. There are several changes that could help with code usability The code does not have a README nor a guide to how to use the scripts. The scripts have all paths hardcoded and it's not clear what the structure of the author's data and scripts are so including that in the README would be helpful too.

Reviewer #3

(Remarks to the Author)

The manuscript by Chang and colleagues characterizes the time-varying functional connectivity in three different cognitive tasks and investigates the relationship between the dynamic configurations during the task with behavioral and cognitive measures that indicate neuropsychiatric profile of the individuals. The study is exploratory in nature, which is fine given that

the study uses a large dataset and they use multiple reinforcement-related tasks with clinically-relevant behavioral measures. The main methodological approaches regarding the derivation of time-varying functional connectivity is sound and consistent. However, there several major issues in the manuscript that should be addressed. In brief, a central point of the manuscript is the discrete FNC states, given the exploratory nature of their study, it is essential to characterize those states, which is not sufficiently done in the manuscript. Another general issue about the manuscript is that the relationship between multiple analysis approaches was not very established. Finally there are some concerns regarding the some of the results that could be addressed by more rigorous approaches.

Major comment 1: The manuscript puts little emphasis on characterizing and defining what each state might indicate. For example, how each state is related to the global fluctuations? Do linearly increasing trend in state 3 is also exhibited in the resting-state? What are the proportions of the networks (and how they are similar/different to resting-state networks) at each state? What is the level of anti-correlations for each state and across which networks? Does high number states lead to very frequent jumps between states rather than more continuous dwell times? Do some FNCs coincide with responses, or epochs in which head movement is more probable? The elbow criterion is a straight forward approach to decide the number of states. However, a systematic investigation of these states would affect the decision as well. So far I see the dwell times of each state do not differ much between resting-state and across task scans, which raises doubts on their implication on task. Furthermore, figure 3 suggest that there are very subtle differences between states 2, 3 and 4, which is very difficult to figure out. Although there are a lot of material in supplementary text, they help understanding different number of states or sliding window lengths, but they do not help understanding what each FNC state might indicate.

Major comment 2: In Figure 2, the values in radial plot at panel b does not resemble those in the bar plots at panel c. For example, the average participation coefficients of state 2 and state 4 is very similar in panel c, but in the radial plot state 2 and 3 exhibit similar participation neg, whereas state 4 has a much higher values close to state 1. Similarly, modularity of state 4 appears to be around 3 in panel c, but it is around 2.5 in panel b. Please double check or explain why these differences appear in the figure.

Major comment 3: I am confused due to the continuous time courses of the state occurrences for EFT and SST tasks in Figure 3. As they used k-means clustering to identify time-varying FNC states, it is expected that for each centroid the occurrences would be in discrete times. I could not find any explanation to understand how that occurrences were calculated. I am guessing that the numbers indicate total occurrences across subjects (or probability of occurrences, which is not clear neither in the text nor in the figure).

Major comment 4: The authors claim differences in correlations between FNC states and blocks that indicates certain features of the task. However, they do not show whether these correlations actually mean something significant at first place. Although for EFT task it is pretty clear that there is an association between detected states and task blocks, for the other two task is the correlation is very low and the evidence for a significant effect is not convincing. For example, although the authors claim a significant difference between large-win and small-win correlations of state 1 and 2, the correlations for MID task are very low $r < 0.2$. Indeed, a visual inspection of the task block suggest that MID task blocks appears to have a temporal structure, i.e. "no-win" trials are concentrated in the beginning of the session, "small-win" trials are concentrated at the middle and "large-win" trials are concentrated at the end of the session. Whether or not this is imposed by chance, it might be imposing a low correlation, also given that the authors seem to report Pearson correlation in this analysis that is sensitive to outliers. The authors should have strictly control for any redundant correlations before claiming any association between FNC states and task blocks. This is even more clear in SST task. The all distributions presented in Figure 3c suggest that the average correlation between FNC states and task blocks might not be different than 0, maybe with an exception of state 3 vs. "go-too-late". Before comparing the correlations across task blocks, the authors should test whether these distributions are significantly different that 0, preferably against a null model in which the correlations are calculated by random states with matching frequencies but time information is scrambled.

Major comment 5: The authors claim that state 3 might be related to "attention lapses". Related to the previous comment that the authors put insufficient efforts to characterize each of these states, the authors do not present any evidence that might suggest such speculation plausible. In resting state studies similar states can be observed and they are typically seem to be related to changes in arousal or sleep. State 3 appears to be related to activation or deactivation of sensory networks. As I suggested before please check whether this state exhibit similar temporal structure in resting-state as well and carefully support any claims with evidence and/or reference to previous literature.

Major comment 6: The results that are presented in Figure 4 is not penetrable and it is not easy to match them to those described in the text. Possibly, there are some latent variables representing externalizing and internalizing symptoms that are correlated with each other. On the righthand side the regression plots suggest that there is a weak but significant effects. However, it is not possible to understand what is being plotted and that what the latent variable indicate. Also the weights for those behavioral abbreviations are difficult to make sense of, it is not possible to spot externalizing and internalizing symptoms. Moreover, the authors do not elaborate how these results make sense in light of other results. As far as I see there is no good correspondence between the results presented in Figure 2 and PLS results. For example despite strong correlations with task blocks with EFT, this task does not appear in PLS analysis. Also, this analysis shows that FNC state 3 also appears to be relevant in resting-state. This also supports the suspicion that the component what is thought to be attention lapse might be sleep/arousal-level related.

Major comment 7: I am not sure if I understood very well the analysis regarding task contrasts presented in figure 5. As far as I understood, this analysis shows that other features of FNCs explain task related activity better than discrete states that are extracted by k-means clustering. Indeed, all these results points to a that trying to detect discrete whole-brain states might

not be possible and instead there might be overlapping configurations of networks. So far I know, this is a common problem in many time-varying functional connectivity studies. Therefore it is not possible to expect the authors to address this issues completely. However, it is important to consider its implications, which might change the interpretation of the results. Moreover, from the distributions in figure 5, one would expect there would be negative connectivity patterns that are as strong as the positive ones. Why the authors did not include negative connections?

Minor comment: Please add a legend to Figure 2 indicating the abbreviations used for each network as in supplementary figure 4.

Minor comment: In figure 3, the time-courses of the first state suggest that indeed this state is almost perfectly in line with presentation of faces regardless of the emotional content. It is clear that if they merge all face trials, the correlations with state 1 would drastically increase.

Minor comment: In the field of time-varying fMRI, multiple terminologies appears, which is inconvenient. This issue is not addressed well in the paper. One important distinction is that many papers that the authors discussed in the introduction deals with the functional connectivity that defines a cortical area through contiguous parcels based on anatomical and/or functional architecture. As far as I understood, the authors especially chose the term functional network connectivity (FNC) to emphasize the fact that they measure the interactions between functional networks from ICA instead of cortical areas. It is important to clearly distinguish these differences throughout the manuscript.

(Remarks on code availability)

Reviewer #4

(Remarks to the Author)

(Remarks on code availability)

Version 1:

Reviewer comments:

Reviewer #1

(Remarks to the Author)

The authors have addressed several of the previous comments and revised the manuscript accordingly, resulting in improved clarity and presentation. However, a number of issues remain that would benefit from further revision to enhance the overall rigor and coherence of the study:

1. To address potential site effects, the authors calculated spatial correlations between FNC states derived from each site and those from all other sites combined, reporting similar patterns that suggest general consistency across sites. While this provides some supportive evidence, it does not directly address whether the subsequent statistical analyses were affected by site effects. A more robust and widely accepted approach would be to apply harmonization techniques, such as ComBat, to explicitly account for site-related variability in the FNC data. It would also be valuable to test whether the main findings are reproducible across individual sites. Additionally, although the authors plotted state occurrence per site and noted a high similarity with the overall results, including quantitative similarity measures would strengthen this conclusion.
2. The STRATIFY sample includes participants with clinical diagnoses; however, the current analyses were performed on the full sample. To assess whether the findings are consistent across different diagnostic groups, it is recommended that the authors report FNC state results and their associations with task stimuli and behavioral performance separately for each group.
3. The diagnostic criteria for clinical group inclusion in the STRATIFY cohort should be described more precisely. For instance, it is unclear whether MDD was diagnosed using DSM-IV criteria or another standardized approach. The current description—"patients with MDD were included if they exhibited current and acute depressive symptoms, with a PHQ-9 score of 15 or above"—does not constitute a formal diagnosis and would benefit from additional clarification.
4. On page 10, the authors compare task performance across MDD, AUD, and control groups, stating, for example, "For MID task performances, MDD patients exhibited lower accuracy and longer reaction time (RT) compared to CON, ...". While informative, this content appears in a section focused on correlations between FNC state occurrence and ongoing task stimuli, which disrupts the logical flow. It is recommended that this paragraph be relocated to a section more appropriate for reporting behavioral performance results.
5. There appears to be an inconsistency in the description of preprocessing procedures. The response letter states that both resting-state and task-based fMRI data underwent the same standard preprocessing, whereas the Supplementary Materials describe different pipelines for each. This discrepancy should be resolved to ensure clarity and consistency. Additionally, a brief note on whether any preprocessing differences could have influenced the results would be helpful.
6. The authors are encouraged to consider implementing a scrubbing procedure to further reduce motion-related artifacts.

This method preserves the number of time points while mitigating motion effects. At least, evaluating the correlation between head motion and FNC measures would help confirm that motion has been sufficiently controlled for.

7. In Figure 2, the percentages for the four FNC states in the EFT and MID conditions still do not sum to 100%. The authors are encouraged to carefully review the manuscript and all supplementary figures to ensure accuracy and consistency throughout.

(Remarks on code availability)

I reviewed some of the code without executing it. The code appears clear and well-organized, and the accompanying README file is well-structured and easy to follow.

Reviewer #2

(Remarks to the Author)

The authors have thoroughly addressed all my comments and concerns. The changes have greatly improved the clarity of the manuscript. I believe this paper is now suited for publication in Nature Communications. Great job - I look forward to seeing it out.

(Remarks on code availability)

The code has been improved with the revision. There is now a clear README with step by step instructions and information about external tools that were used. The authors also seemed to have made the code more flexible and easy to use with the data they provided.

Reviewer #3

(Remarks to the Author)

I thank the authors for addressing all issues raised in the previous review thoroughly and to the point. All the responses were detailed and clear. I have no other comments.

(Remarks on code availability)

I did not attempt to run the codes. As far as I could see the codes reproduce the statistical analyses provided in the results given the main metrics pre-calculated using another library and stored in data folder. The codes look well organised. It would be useful to specify which Matlab toolboxes are required to run the code in the readme document (e.g. statistics..etc).

Reviewer #4

(Remarks to the Author)

(Remarks on code availability)

Version 2:

Reviewer comments:

Reviewer #1

(Remarks to the Author)

The authors have well addressed all my concerns. Nice work!

(Remarks on code availability)

Reviewer #4

(Remarks to the Author)

(Remarks on code availability)

Point-by-point response

Reviewer #1 (Remarks to the Author):

The manuscript by Chang et al. investigates time-varying functional network connectivity (FNC) during resting-state and three reinforcement-related tasks across 1,417 participants. The authors identify four consistent FNC states, showing that these states are sensitive to task performance and psychopathology. Overall, the manuscript presents valuable findings with solid validation in an independent dataset, providing novel insights into how brain networks adapt to support cognitive tasks and suggesting potential biomarkers for assessing individual psychopathology risk. However, I have several suggestions that could further strengthen the manuscript:

Response: We thank the reviewer for the thorough assessment and the constructive suggestions to improve our manuscript. We provide a point-by-point response to each question below. Corresponding changes to the revised version of the Manuscript and Supplementary Materials are **marked in red**.

1. Since the imaging data were collected from multiple sites, it would be important to investigate whether site effects influenced the FNC states and subsequent results. If significant site effects are found, harmonization techniques should be applied to address this issue.

Response: We thank the Reviewer for highlighting this important issue. To minimize potential site effects, harmonization procedures and quality control measures were implemented across all sites prior to MRI data acquisition (as detailed in ¹). Briefly, scanning parameters that were compatible with all sites were used to ensure images had comparable contrast and signal-to-noise ratio. Additionally, the following quality control procedures were carried out before data acquisition: (1) a phantom scan was conducted to adjust for geometric distortions and signal uniformity across different hardware configurations; (2) healthy volunteers were scanned at each site to assess variability in both structural and functional MR images.

For this study, we added a sensitivity analysis to assess the influence of site effects on the FNC states' estimations. Specifically, we examined the spatial correlations between the FNC states derived from each of the eight recruitment sites and those from all other sites combined

(Fig. S18). The results indicated that FNC states derived from individual sites were highly similar to those derived from other sites ($r \geq 0.91, p < 10^{-5}$). In addition, we plotted each state's occurrence for individual sites, which are highly similar to the states' occurrence derived from all sites combined.

We have now mentioned the sensitivity analysis in the manuscript and included detailed results in the Supplementary Materials (Fig. S18):

Manuscript page 12 Sensitivity analysis

To assess whether the derived FNC states were influenced by site effects (Table S2), we examined the spatial correlation between the FNC states derived from each of the eight recruitment sites and those from all other sites combined (Fig. S18). The results showed that FNC states derived from individual sites were highly similar to those derived from other sites ($r \geq 0.91, p < 10^{-5}$). In addition, we plotted each state's occurrence for individual sites, which are highly similar to the states' occurrence derived from all sites combined.

Fig. S18 Influence of recruitment site effect on FNC state. a) Spatial correlation between FNC states derived from each of the eight recruitment sites and those from all other sites combined. Red numbers indicate the lowest spatial correlation for each scanning session. b)

FNC states occurrences derived from each individual site (in different colours) are similar to those derived from all sites combined (in black). The x-axis represents the scanning time of each session. The y-axis is the occurrence for each FNC state, calculated as the proportion of subjects being categorized into a certain state at each time point.

2. *Could the authors provide more explanation as to why modularity and participation coefficient are used to represent brain network segregation and integration? These two metrics are typically negatively correlated, which may fail to capture an optimized network with both segregation and integration. Alternatively, using metrics like Cp/Lp or local/global efficiency may better characterize this issue.*

Response: We chose modularity and participation coefficient because these two measurements are based on the community participation of a network. Previous studies and our own results (Figure 2) showed that modularity alters within a task-fMRI session (e.g., EFT, SST, and MID) and is relevant for the current cognitive processes ^{2,3}. Thus, we compared modularity and participation coefficients between the derived FNC states.

Note that, as modularity and participation coefficients could be influenced by basic characteristics such as the degree distribution, we compared these two coefficients of each FNC state based on normalization metrics calculated from 100 randomized networks. The randomized networks were designed to preserve the degree distribution of positive and negative FNC, while rewiring the connections at random. Normalized modularity and participation coefficient were then calculated by dividing their empirical values by the average of randomized metrics.

While clustering coefficient (Cp), path length (Lp), and efficiency are also commonly used network measurements, they do not reflect the modularity structure of the network and hence are less relevant to cognitive processing. Therefore, we did not report these measurements in the manuscript.

3. *The MID task showed a low correlation with the clustering centroid of state 3 for resting-state and other task, suggesting a unique connectivity pattern in the MID. However, it may not be appropriate to force the MID state into state 3. Did the authors consider conducting*

clustering on the pooled matrix across all tasks (REST, EFT, MID, and SST)? This approach could better identify both common and task-specific FNC states.

Response: We thank the Reviewer for raising this insightful question. To clarify, the four FNC states were extracted separately from each task fMRI and resting-state session. For example, the MID scanning session was divided into 182 overlapping windows for every subject. Then k -means clustering was performed on 182 windowed FNC matrices \times 1221 subjects = 222,222 FNC matrices. The clustering centroids were derived specifically from the MID FNC matrices and were not constrained by predefined FNC states from other sessions. Notably, we observed that the four FNC states from each session showed similar connectivity patterns, with the exception of MID state 3 (Figure 2a).

We have tried to apply the clustering algorithm across all sessions (REST, EFT, MID, and SST), as suggested by the Reviewer. However, this approach presented a substantial computational challenge due to the large number of FNC matrices involved: 222,222 FNC matrices for MID, 232,392 matrices for EFT, 365,400 matrices for SST, and 149,641 matrices for REST. Combining all these matrices to compute pairwise distances for clustering would lead to an out-of-memory error. Therefore, we opted to perform k -means clustering separately for each scanning session to manage the computational demands more effectively.

4. *In the independent dataset, the authors observe similar patterns in the four time-varying FNC states and their associations with task performance, which is consistent with the main results. Given that this dataset includes participants with clinical conditions, it would be valuable to investigate whether these individuals exhibit altered FNC values, dynamic characteristics, or behavioral relevance compared to the primary findings. This additional analysis could provide deeper insight into the relationship between time-varying FNC and mental health.*

Response: We thank the Reviewer for this suggestion. As the Reviewer suggested, we compared the dwell time of FNC states between STRATIFY major depressive disorder (MDD), alcohol use disorder (AUD) and healthy controls (CON). Only REST state 4 exhibited a higher dwell time in CON as compared to MDD (two-sample t -test, $t = 3.48$, $p = 5.57 \times 10^{-4}$) and AUD patients ($t = 3.99$, $p = 8.11 \times 10^{-5}$) after FDR correction (**Author Response Figure 1**).

In addition, we compared symptom severity and task performance between STRATIFY patients and controls in response to Point 7 below. We also compared FNC strength derived from time-varying and static connectivity across groups, to supplement the results of *Testing the clinical relevance of time-varying FNC* (in response to Point 5 below).

Author Response Figure 1 FNC dwell time comparison between patient STRATIFY major depressive disorder (MDD), alcohol use disorder (AUD), and healthy controls (CON). Asterisks indicate significant differences after FDR correction for multiple comparisons.

5. *The manuscript discusses the classification capability of time-varying FNC between patients and healthy controls. However, it would be helpful if the authors could clarify which specific FNC states exhibit stronger predictive power for distinguishing between these groups.*

Response: On page 12 of the Manuscript, *Testing the clinical relevance of time-varying FNC*,

we found that the connectivity derived from time-varying FNC states explains a higher variance than that derived from static FNC when distinguishing patients from controls (Figure 6d).

To further examine which FNC state exhibits stronger predictive power, we conducted a post-hoc analysis to compare the connectivity of each FNC state and static FNC between CON, MDD, and AUD (Fig S17).

For the EFT task, Figure 6d shows that the connectivity strength of time-varying FNC provides better discrimination between CON and MDD patients (R^2 : time-varying FNC = 3.00%, static FNC = 0.34%). Correspondingly, in Figure S17, we observed that the connectivity strength of state 3 was significantly different between CON and MDD patients (two-sample t -test, $t = 3.08$, $p = 0.002$).

For the SST task, time-varying FNC exhibited stronger predictive power in distinguishing CON from AUD patients (R^2 : time-varying = 7.26%, static FNC = 5.56%). In Figure S17, we found significant differences in the connectivity strength of states 1, 2 and 4 ($t = 3.87$, 3.21 , 3.86 , $p = 1.31 \times 10^{-4}$, 0.001 , 1.38×10^{-4} , respectively), as well as in static FNC ($t = 4.42$, $p = 1.33 \times 10^{-5}$), between CON and AUD patients.

We have incorporated this post-hoc analysis into the Manuscript and Supplementary Materials (Fig S18):

Manuscript page 11, Testing the clinical relevance of time-varying FNC

To investigate the clinical relevance of our model, we used logistic regression to test the performance of time-varying FNC and static FNC in distinguishing disease status (CON vs. MDD, CON vs. AUD, CON vs all patients (PAT)). We found that time-varying connectivity in the EFT explained 3.00% of the variance of MDD, compared to 0.34% in static FNC, **which was mainly driven by the time-varying connectivity of state 3 that was significantly different between CON and MDD patients ($t = 3.08$, $p = 0.002$, Fig S17)**. Of note, the coefficients for the regressors in these disorders were in opposite directions, as the combined patient sample vs. controls showed no notable explanation of variance in either condition (Figure 6d and Table S15). In the SST, there was a high explanation of variance of AUD by time-varying FNC of 7.26% vs. 5.56% by static FNC, consistent with the association with the binge drinking score observed in IMAGEN and STRATIFY controls. **Post-hoc analysis showed that time-varying connectivity of states 1, 2, and 4 ($t = 3.87$, 3.21 , 3.86 , $p = 1.31 \times 10^{-4}$, 0.001 , 1.38×10^{-4} ,**

respectively), as well as in static FNC ($t = 4.42, p = 1.33 \times 10^{-5}$), are significantly different between CON and AUD patients (Fig S17).

Fig. S17 Post-hoc group comparison of connectivity strength of time-varying FNC and static FNC between CON, MDD, and AUD patients. Two-sample t -tests were performed between groups to compare the connectivity strength of each state in time-varying FNC and static FNC across groups. Red lines indicate significant differences after FDR correction for multiple comparisons.

6. In the section "Regional FNC Involved in Task-Specific Cognitive Processes," I recommend that the authors expand on the specific regions associated with each task, as well as those shared across the three tasks. This would provide a clearer understanding of the task-specific brain regions involved in each cognitive process.

Response: We thank the Reviewer for this suggestion. In the revised manuscript, we have added the description of the task-specific connections. Detailed FNC labels and t -statistics are listed in Table S11.

In Figure 5, we found that connections associated with emotional processing in the EFT (showing the highest t -statistics of angry versus neutral faces) do not distinguish task conditions in the MID or SST (Figure 5a). A similar task-specificity effect was observed for the MID and SST (Figure 5b-c). Among the top 20 task-specific connections in each task, only the connectivity between the middle frontal gyrus (MFG) and right posterior middle temporal gyrus (R.STG.pos) exhibited significant between-condition differences in both EFT ($t = 7.51$) and SST ($t = 4.45$).

Manuscript page 9, Regional FNC involves in task-specific cognitive processes

To determine the specificity of these regional FNC, we plotted 20 connections with the highest t -statistics in one task, and compared their t -statistics with those in the other two tasks. FNC associated with emotional processing (with the highest t -statistics of angry versus neutral faces) do not distinguish task conditions in the MID or SST (Figure 5a). A similar task specificity was observed for the MID and SST (Figure 5b-c). Anatomic locations of the 20 task-specific FNC are shown in the connectogram plot (Figure 5d). **The EFT task-specific connections are mainly between the bilateral superior temporal regions and the default mode, prefrontal areas, and visual cortices. For the MID, connections of the default mode regions with frontoparietal regions, visual and sensorimotor areas showed higher correlations with the large-win condition compared to the no-win condition. For the SST, connections of the sensorimotor areas with default mode and temporal regions, as well as connections between the putamen and frontoparietal regions, showed higher correlations with the stop-success condition compared to the stop-failure condition.** Detailed FNC labels and t -statistics are listed in Table S11. Taken together, our results suggest that whole-brain and regional FNC may reflect different aspects of cognitive processing: while whole-brain FNC states are broadly modulated by the most salient stimuli (EFT faces stimuli, MID large-win and SST stop signals), regional FNC are related to more subtle task processes and show task specificity.

7. Including detailed clinical information about the participants from the independent STRTIFY cohort would be valuable. This should include diagnostic criteria, symptom severity, and task performance scores, along with behavioral questionnaire data. Such information

would enable a more comprehensive comparison with the main findings.

Response: The STRATIFY cohort recruited individuals with moderate to severe depressive symptoms or alcohol abuse symptoms. Patients with MDD were included if they exhibited current and acute depressive symptoms, with a Patient Health Questionnaire-9 (PHQ-9) score of 15 or above. Patients with AUD were included if they scored 15 or higher on the Alcohol Use Disorders Identification Test (AUDIT). The symptom severity scores for both patients and controls are shown in **Author Response Figure 2** below.

Author Response Figure 2. Depression and alcohol abuse symptom severity of STRATIFY CON, MDD, and AUD patients. Red lines indicate significant group differences after FDR correction for multiple testing.

We also compared MID and SST task performances between STRATIFY CON, MDD, and AUD patients (Note: the EFT is a passive viewing task that does not require participant responses). As shown in **Figure S15**, MID accuracy (percentage of correct trials) is lower in MDD patients compared to CON across all trial types: large-win, small-win, and no-win. Reaction time (RT) for all MID trial types was significantly longer in both MDD and AUD patients compared to CON. For SST performance, the stop signal RT is significantly longer in AUD patients than in controls. Additionally, the go signal reaction time is significantly longer in both AUD and MDD patients compared to CON. The SST was designed to automatically adjust to ensure every subject has, on average, a 50% success rate in responding to stop signals.

Thus, no significant group differences were observed in the percentage of correct responses to stop signals or errors related to go signals (e.g., late responses or incorrect responses).

The task performance comparisons were consistent with previous knowledge of cognitive impairments in AUD and MDD patients. In general, patients exhibited longer response times, with MDD patients showing poorer performance on MID accuracy, while AUD patients demonstrated more pronounced deficits in inhibitory control, as evidenced by longer stop signal reaction times.

We have added the symptom assessment and task performances of STRATIFY participants to the revised Manuscript and Supplementary Materials:

Supplementary Materials Page 7, Replication of findings in the independent STRATIFY cohort STRATIFY cohort

The STRATIFY cohort ^{4,5} recruited patients (ages 19-25) with alcohol use disorder, major depression and healthy controls from three recruitment sites in Berlin, London and Southampton, of which Berlin and London are also recruitment sites for IMAGEN. We included alcohol use disorder (AUD, $n = 150$), major depressive disorder (MDD, $n = 168$) and healthy controls (CON, $n = 210$) from STRATIFY. **Patients with MDD were included if they exhibited current and acute depressive symptoms, with a Patient Health Questionnaire-9 (PHQ-9) score of 15 or above. Patients with AUD were included if they scored 15 or higher on the Alcohol Use Disorders Identification Test (AUDIT).** Demographic information is provided in Table S13.

Manuscript Page 10, Validation in an independent dataset

Our replication of the correlation between occurrences of FNC state and ongoing task stimuli (Table S14) yielded results similar to IMAGEN (Figure 3 and Table S8): We found that states 1 and 4 were positively correlated with conditions of interests, while state 3 in the SST session correlated with SST go too-late errors (one-sample t -test, $t_{265}=9.84$, $p=1.09 \times 10^{-19}$, 95% CI=[0.07, 0.11]). **For MID task performances, MDD patients exhibited lower accuracy and longer reaction time (RT) compared to CON, whereas AUD patients showed only longer RT relative to CON (Fig S15). For SST task performances, AUD patients had longer RT for stop**

signals, and both AUD and MDD patients exhibited longer RT for go signals (Fig S15). We replicated the significant correlations between SST go too-late error with dwell time of state 1 and state 3 in STRATIFY CON (Table S15).

Fig. S15 MID and SST task performances of STRATIFY CON, MDD and AUD patients. Red lines indicate significant group differences after FDR correction.

8. *The authors mention different preprocessing approaches for resting-state and task-state fMRI data. It would be helpful to explain whether these preprocessing steps might have influenced the results and whether the approach was consistent across tasks.*

Response: The IMAGEN data were centrally preprocessed using standardised pipelines, with scripts and documentation openly available at: https://github.com/imagen2/imagen_processing/. Both resting-state and task-based fMRI data underwent standard preprocessing steps, including brain extraction, motion correction, normalisation, spatial smoothing, and resliced to 3mm isotropic voxels. The same customised EPI template was used for normalisation across all sessions to ensure a consistent transformation from individual space to MNI standard space.

To assess whether preprocessing differences influenced data quality, we compared the group masks generated for resting-state and task-based fMRI sessions. Specifically, we computed a group mask by averaging individual subject masks and applying a 70% threshold for each session. The resulting group mask from resting-state data exhibited high correlations with those from EFT, MID, and SST sessions ($r = 0.9695, 0.9696, 0.9696$), suggesting

comparable data quality across sessions.

Furthermore, as demonstrated in Figure 2a and Figure S14, the four distinct FNC states remain highly consistent across sessions and datasets, i.e., IMAGEN and STRATIFY, despite variations in preprocessing software. This suggests that the identified FNC states are robust to preprocessing variations. For analyses investigating FNC states in relation to task stimuli, we used task-based fMRI data and did not directly compare resting-state and task-based results. Thus, our findings are unlikely to be affected by differences in preprocessing pipelines.

9. *Time-varying FNC is known to be affected by head motion. A more thorough discussion of head motion in this study would be beneficial. This could include applying "scrubbing" to further mitigate motion artifacts or analyzing the correlation between head motion and FNC.*

Response: We agree with the Reviewer that head motion can be a significant confound in fMRI studies, and we have taken several steps to mitigate its potential influence on our results: 1) we first excluded subjects with excessive head motion (mean framewise displacement (FD) > 0.2 mm), to minimise the influence of excessive motion on our results (Table S1); 2) when calculating time-varying FNC, we regressed the six realign parameters as covariates in the sliding window analysis to minimise micro head movements on FNC estimation (Supplementary Material and Methods); 3) further, in between-subject analyses, we regressed covariates including age, sex, recruitment sites and head motion (mean FD). The mean FD values for participants in both resting-state and task-based fMRI sessions are shown in **Author Response Figure 3** below.

We did not apply the "scrubbing" procedure, which is commonly used in static functional connectivity analysis, because in our study, the "scrubbing" procedure may lead to a varying number of brain scans in a sliding window when estimating time-varying FNC.

Author Response Figure 3. Mean framewise displacement (FD) of resting-state and task-based fMRI sessions of included participants.

10. *To improve consistency and readability, I recommend formatting all tables in the manuscript as three-line tables.*

Response: We thank the Reviewer for this suggestion. We have formatted all tables as three-line tables. The revised tables include Table S7-S9 and Table S12-S16 in the Supplementary Materials.

11. *In Figure 2, could the authors clarify why the percentages for the four FNC states do not sum to 100% in the resting-state fMRI and EFT conditions?*

Response: We thank the Reviewer for the careful observation. The discrepancy arises due to decimal approximation, where the percentages of the four states in the resting-state and the EFT sessions did not add up to 100%. To clarify this and prevent any misunderstanding, we have retained two decimal places in the revised manuscript, so that the percentages of state occurrences all add up to 100%.

Figure 2 Resting-state and task-fMRI sessions contain recurring FNC states. **a)** Four representative FNC states (cluster centroids) were derived from the *k-means* clustering applied to sliding-window FNC matrices from the resting-state (upper row) and three task-fMRI sessions (lower row). Each FNC state represents a recurring functional connectivity pattern between 61 ICA components, which are categorised into seven functional domains (Figure S2). **The percentage of each state within a scanning session is shown in parentheses.** Connectivity profiles of the same state across resting-state and task-fMRI sessions exhibit a higher correlation (average $r=0.86\pm0.17$) compared to correlations between different states (average $r=0.67\pm0.10$, two-sample *t*-test: $t_{118}=7.17$, $p=6.93\times10^{-11}$, 95% CI=[0.13, 0.24], Cohen's $d=1.63$). **b)** The four FNC states differ in network modularity and participation coefficients, which quantify the level of segregation and integration between functional modules. The modularity and participation coefficients are averaged across resting-state and task-fMRI sessions. **c)** One-way ANOVA and post-hoc *t*-test comparisons between FNC states on modularity and participation coefficients. Red lines indicate significant differences between FNC states after FDR correction (Table S4).

Abbreviations for the seven functional domains: subcortical (SCN), temporal (TEP), sensorimotor (SMN), visual (VSN), cognitive control (CON), default mode (DMN) and cerebellar (CEB) networks.

12. In Figure 6, it would be helpful to explain what the gray dots represent.

Response: We thank the Reviewer for raising this point. In Figure 6, the gray dots represent behavioural variance explained by both time-varying FNC and static FNC using a linear regression model. We added the color legend to Figure 6 to explicitly indicate this.

Figure 6 Behavioural variance explained by time-varying FNC and static FNC from the EFT, MID and SST sessions. The same set of 20 task-specific connectivity identified from the previous section was taken from time-varying FNC states and static FNC. **a)** In the IMAGEN cohort, linear regression of 29 behaviours (Table S9) were regressed as a function of time-varying FNC (blue), static FNC (green), or **both (gray)**. Model significance is obtained by comparing the full model with null models (only constant term). Among the 23 significantly

fitted behaviours (model significance uncorrected $p_{\text{one-tail}} < 0.05$, Table S12), 22 behaviours showed a higher R^2 using time-varying FNC than static FNC regression model. **b)** As an example, we showed one behavioural fitting using time-varying FNC and static FNC from each task. **c)** The same model fitting and comparison procedure were applied in the STRATIFY healthy controls. For the behavioural items that are available in STRATIFY (Table S9), time-varying FNC has a higher R^2 using time-varying FNC in 18 out of 19 behavioural items. **d)** Further, the disease status of STRATIFY patients and controls was regressed as a function of time-varying FNC and static FNC using logistic regression. All models showed a numeric higher R^2 using time-varying FNC than static FNC regression model.

Abbreviations: CON: healthy controls; AUD: alcohol use disorder; MDD: major depressive disorder; PAT: all patients. Covariates of age, sex, recruitment sites and head motion were regressed before analysis.

Reviewer #1 (Remarks on code availability):

The code provided is well-written and reproducible, allowing for the analysis reported in the manuscript to be conducted. However, the README file lacks sufficient instructions and clarity in its presentation.

Response: We thank the Reviewer for the valuable suggestion. We have added a comprehensive README file that includes step-by-step instructions for using the code and data. Additionally, we have provided example data to demonstrate the main analyses performed in this project.

The revised code and data are available at the following repository:
https://github.com/xchang007/tvFNC_IMAGEN.git

A screenshot of the updated README file is attached below for reference.

Product Solutions Resources Open Source Enterprise Pricing
Search Sign in Sign up

xchang007 / tvFNC_IMAGEN Public
Notifications Fork Star

Code Issues Pull requests Actions Projects Security Insights

main
1 Branch
0 Tags

Code

xchang007 Update README.md	ld5290 · 14 minutes ago	23 Commits
code Add files via upload	28 minutes ago	
data Add files via upload	30 minutes ago	
README.md Update README.md	14 minutes ago	

Menu
⋮

tvFNC_IMAGEN

This repository contains code and data for the project "Neurocognitive characterization of behaviour and mental illness through time-varying brain network analysis"

code

- #### 1. rest & task ICA

GIFT toolbox (<https://trendscenter.org/software/gift/>)
Calhoun, V. D., Adali, T., Pearlson, G. D. & Pekar, J. J. Hum. Brain Mapp. 14, 140–151 (2001)
- #### 2. time-varying FNC

GIFT toolbox – Temporal dFNC
- #### 3. time-varying FNC corr and FNC network metrics

code\S_FNC_states_network.m
data\R_FNC_states_network.mat
- #### 4. FNC states corr task stimuli

code\S_FNC_states_corr_task_EFT.m
data\R_FNC_states_corr_task_EFT.mat
example data EFT
- #### 5. sPLS between state dwell time and behaviours

sPLS analysis between FNC states' dwell time and 29 behavioural items using CCA/PLS Toolkit
Mihalik, A. et al. Biol. Psychiatry Cogn. Neurosci. Neuroimaging 7, 1055–1067 (2022).
code\S_run_sPLS.m
data\R_run_sPLS.mat
- #### 6. FNC task specificity

code\S_corr_SPM_regional_FNC.m
data\R_corr_SPM_regional_FNC.mat
- #### 7. compare_state_based_Nfc_mdI

code\S_compare_state_based_Nfc_mdI.m
data\R_compare_state_based_Nfc_mdI.mat
example data EFT

No description, website, or topics provided.

Readme

Activity

1 star

1 watching

0 forks

Report repository

Releases

No releases published

Packages

No packages published

Languages

MATLAB 100.0%

© 2025 GitHub, Inc. Terms Privacy Security Status Docs Contact Manage cookies Do not share my personal information

Reviewer #2 (Remarks to the Author):

The paper was well written, interesting, and lays out a rigorous set of analyses with an impressive replication. My biggest point of feedback is the methods are very dense which lead me to some confusion between, “states” that were found across tasks were being used as predictors or and the regional FNCs (which aren’t yet clear to me how they are calculated) and other ways functional connectivity used is such as calculating graph metrics. This seems to be a crux of the paper so I have made several suggestions where the authors could clarify the methods, as well as other points of clarification.

Response: We thank the reviewer for the positive assessment and constructive suggestions to improve our manuscript. We have revised Figure 1 to provide a clearer overview of the analysis strategies. Besides, we made clarifications to several places as suggested by the Reviewer. Please find the point-by-point response to each question below. Corresponding changes to the revised version of the Manuscript and Supplementary Materials are **marked in red**.

1. *The introduction is clear and well written. I do wonder about why the authors characterize the three tasks as “reinforcement tasks” could the authors explain that?*

Response: We thank the Reviewer for raising this question. The term "reinforcement tasks" commonly refers to the mental and neural mechanisms that individuals utilize information from rewards and punishments feedback to guide learning, decision-making, and adaptive behaviour^{6,7}. These processes involve evaluating outcomes and modifying actions, which are related to emotion regulation⁸ and cognitive controls⁹. Therefore, the IMAGEN cohort employed the emotional faces task (EFT), monetary incentive delay task (MID) and stop-signal task (SST), to assess these reinforcement-related cognitive processes¹. Dysfunction of these reinforcement-related processes is closely linked to a range of mental health problems during adolescence, and therefore is of particular interest in this study.

2. *Since the sample is bordering on youth/adult age range and clarification if these were all adults in this sample would be helpful*

Response: The IMAGEN study is a population-based, longitudinal cohort that recruited 2,000 adolescents aged 14 years from eight research centers across the United Kingdom, Germany,

France, and Ireland. Participants were followed up twice, at ages 19 and 23. In this study, we analyzed neuroimaging and behavioural data from 1,417 participants, with a mean age of 19.09 ± 0.76 years. The youngest participant was 17.68 years old. The age distribution of participants is shown in **Author Response Figure 4**.

Author Response Figure 4. Age distribution of recruited IMAGEN participants used in this study

3. *Can the authors provide rationale for smoothing the task data (5 mm) more than the rest data (4 mm)?*

Response: The choice of kernel size was based on common practices in fMRI studies, where 4-8mm FWHM Gaussian kernel is commonly used for balancing the trade-off between signal-to-noise ratio and the preservation of spatial specificity¹⁰. The IMAGEN data were centrally preprocessed using standardized pipelines, with scripts and documentation openly available at: https://github.com/imagen2/imagen_processing/. We did not re-run preprocessing for this study.

Nevertheless, we did apply a multi-step quality control pipeline for the resting-state and task-fMRI data to ensure the quality of data for further analysis (Table S1). The slightly different smoothing kernel is unlikely to change the nature of our results, as demonstrated in Figure 2a and Figure S14, the four distinct FNC states remain highly consistent across sessions and datasets, including IMAGEN and STRATIFY, despite variations in preprocessing software. This suggests that the identified FNC states are robust to preprocessing variations. For analyses investigating FNC states in relation to task stimuli, we used task-based fMRI data and did not directly compare resting-state and task-based results. Thus, our findings are unlikely to be

affected by differences in choice of smoothing kernel.

4. *Since the authors are using ICA for network definition, can they clarify what the nodes are for the graph metrics calculations of modularity and PC? Were these run voxelwise? Were the networks used to calculate PC the networks that were derived from the Louvain algorithm?*

Response: We thank the Reviewer for the clarification. We used the 61 selected ICA components as nodes to calculate the graph metrics (modularity and participation coefficient), based on functional network connectivity (FNC) between each pair of ICA components. The ICA components were used as nodes consistently throughout the study. Graph metrics were calculated using the Brain Connectivity Toolbox (BCT) ¹¹. The Louvain modularity algorithm was applied to detect the optimal community structure of network and calculate modularity and participation coefficient ¹².

We clarified the node definition and module detection algorithm in the revised Manuscript and provided detailed description in the Supplementary Materials and Methods:

Manuscript page 5, Summary of analysis strategy

We analyzed resting-state and three reinforcement-related task-fMRI of 1,417 participants of the IMAGEN cohort ¹ at age 19 years (Table S1-S2). (i) Time-varying functional network connectivity (FNC) of each scanning session was derived using sliding window analysis ¹³ (Figure 1a) on time-series of 61 components from group independent component analysis (ICA) ¹⁴ (Figure S2 and Table S3). **In this study, we used the term FNC to refer to the correlation between ICA components' time-courses** ¹⁵;

Manuscript page 6, Resting-state and task-fMRI sessions contain reoccurring FNC states

The connectivity profiles indicate that FNC states have different degrees of integration and segregation between functional domains, which can be quantified by modularity and participation coefficient from graph metrics ¹⁶ (Figure 2b and Table S4). **The Louvain modularity algorithm was applied to detect the optimal community structure of network** ¹² (Supplementary Materials and Methods).

Supplementary Materials and Methods, page 4, Network graph metrics comparison of FNC states

We quantitatively compared the level of network integration and segregation of the derived FNC states using the Brain Connectivity Toolbox (BCT) ²⁶. A commonly used network segregation measure is modularity, which quantifies the degree to which a network can be divided into several subnetworks with maximal within-group connectivity and minimal between-group connectivity. The Louvain modularity algorithm was applied to detect the optimal community structure of the network and calculate modularity ²⁷. In addition, we calculated the participation coefficient, which assesses the diversity of connections between subnetworks of nodes (i.e. regions). The participation coefficient was calculated for positive and negative FNC separately and averaged across all regions. As network metrics can be influenced by basic characteristics, such as connection weights and distributions, we compared the modularity and participation coefficient of each FNC state with metrics calculated from 100 randomized networks. The randomized networks preserved the degree distribution of positive and negative FNC while rewiring the connections at random. Normalized modularity and participation coefficient were calculated by dividing empirical values by randomized metrics.

5. *When the authors are describing the correlation between pairwise FNCs and task events (Figure 5) it's not clear; a. What pairwise FNC refers to, are these pairwise correlations between each ICA component making up each state? Or are these pairwise correlations between networks identified using the Louvain algorithm? Also, how were the regional network components given the labels that appear in Figure 5? Looks like some of this is covered in figure S2 but I missed it in the methods.*

Response: We thank the Reviewer for the careful reading. To clarify, pairwise FNC refers to the correlations between each pair of the 61 ICA components. These 61 ICA components were categorized into seven functional domains: subcortical (SCN), temporal (TEP), sensorimotor (SMN), visual (VSN), cognitive control (CON), default mode (DMN), and cerebellar (CEB) networks. The labels for these domains are shown in Figure 5 and Figure S2. The anatomical locations and peak coordinates for the 61 ICA components are provided in Table S3.

We have added this information in the Manuscript and Figure 5 legend:

Manuscript page 9, Regional FNC involves in task-specific cognitive processes

In addition to describing ongoing task modulation on a whole-brain level (Figure 3), we further investigated task effects on regional FNC between two ICA components (pair-wise FNC).

Figure 5 Regional FNC distinguishes contrasts that are related to emotional processing, reward sensitivity and motor inhibition. a) For the EFT, one-sample t -tests were performed between FNC correlations with angry faces versus neutral faces. The distribution of t -statistic of all 1830 (61 \times 60/2) pair-wise FNC (i.e., connectivity between two ICA components) was

shown on the left. On the right shows the 20 task-specific FNC with the highest EFT t -statistics, which do not distinguish the MID and SST contrasts; similarly, **b)** FNC with the highest t -statistics of large-win versus no win condition in the MID do not distinguish the EFT and SST conditions; and **c)** FNC with the highest t -statistics of stop-success versus stop-failure condition in the SST do not distinguish the EFT and MID conditions. **d)** The connectogram plot shows anatomical locations of the 20 task-specific FNC of EFT, MID and SST. **Their functional domain names were labelled according to Figure S2 and Table S3.**

6. *Further, it isn't clear which states any of the regional FNCs are from? Or are they not from specific states but averaged along the whole task?*

Response: To clarify, in Figure 5, the regional FNC are derived from time-varying FNC matrices, and are calculated parallel to FNC states. As shown in the revised Figure 1(b), for each pair of brain regions (ICA components), we computed a time-varying connectivity strength, $FNC_{(i,j,t)}$, where i and j are two ICA components, and t represents a sliding window during the scanning session. We then correlated the connectivity strength and occurrences of task stimuli to determine task modulation on regional FNC. To make this clear, we revised Figure 1 to clarify the methodology.

Figure 1 Summary of analysis strategy. a) Resting-state (REST) and three reinforcement-related task-fMRI of the IMAGEN cohort were pre-processed, quality controlled, and parcellated into 61 components using the group independent component analysis (ICA). Time-varying functional network connectivity (FNC) between each pair of the 61 components was estimated using the sliding window analysis. The window length is 17.6s for the REST, emotional faces task (EFT), and stop-signal task (SST) sessions and is 8.8s for the monetary incentive delay task (MID) to accommodate the block duration of MID (~10 s). **b) On the whole-brain level, *k*-means clustering analysis was applied to windowed FNC matrices to derive reoccurring FNC states from each scanning session. FNC state occurrences (i.e., the proportion of participants classified into a given state) were correlated with ongoing task stimuli to investigate how task conditions modulate whole-brain FNC. On the regional level,**

connectivity strength between each pair of ICA components was assessed. Correlation between regional connectivity strength and task stimuli was calculated to investigate task modulation on regional connectivity. c) To determine the behavioural relevance of whole-brain FNC state and regional FNC, we performed sparse partial least squares (sPLS) between FNC states dwell time (the average length of a subject having a certain state) and mental health symptoms and cognitions. Additionally, we compared behavioural variance explained by regional time-varying FNC versus static FNC using linear regression. d) The results were validated in an independent clinical dataset, including the reproducibility of FNC states and their associations with task modulation and behavioural outcomes.

7. *I found the section describing the time-varying vs. static FNC prediction a little confusing, it's possible 'state' is being used to mean both unique FNC pattern and task in this section as in "The 20 task specific FNC was averaged to derive four state-based time-varying FNC and one static FNC for every participant. "Are these 4 different FNC's one for each task? I am also confused by what metric is being used to predict reinforcement related behaviors , is it the average FC of that FNC? How does this capture the time-varying aspect?*

Response: We apologize for the unclear description. In the revised Figure 1c, we illustrated how regional time-varying FNC and static FNC were derived and used in behavioural fitting shown in Figure 6.

Briefly, in Figure 5, we found there are regional FNC correlated with ongoing task stimuli and distinguish task conditions (angry versus neutral faces in EFT; large-win versus no-win in MID and stop-success versus stop-failure in SST, Figure 5a-c). We then took the top 20 task-specific regional FNC (Figure 5d) from the four time-varying FNC patterns of each participant (Figure 2a shows the cluster centroid of each FNC pattern, or FNC state). In parallel, we derived the same 20 task-specific regional FNC from static connectivity analysis (i.e., to calculate a connectivity strength using the time course of an entire scanning session, rather than from a segment of time window). To reduce the number of behavioural fitting parameters, we averaged the connectivity strength of the 20 task-specific regional FNC. Therefore, we obtained four state-based time-varying FNC and one static FNC for every participant, and used these FNC to regress every behavioural item shown in Table S9.

We revised Figure 1c. and provided a detailed description in Supplementary Materials.

Figure 1 Summary of analysis strategy. a) Resting-state (REST) and three reinforcement-related task-fMRI of the IMAGEN cohort were pre-processed, quality controlled, and parcellated into 61 components using the group independent component analysis (ICA). Time-varying functional network connectivity (FNC) between each pair of the 61 components was estimated using the sliding window analysis. The window length is 17.6s for the REST, emotional faces task (EFT), and stop-signal task (SST) sessions and is 8.8s for the monetary incentive delay task (MID) to accommodate the block duration of MID (~10 s). b) **On the whole-brain level, *k*-means clustering analysis was applied to windowed FNC matrices to derive reoccurring FNC states from each scanning session. FNC state occurrences (i.e., the**

proportion of participants classified into a given state) were correlated with ongoing task stimuli to investigate how task conditions modulate whole-brain FNC. On the regional level, connectivity strength between each pair of ICA components was assessed. Correlation between regional connectivity strength and task stimuli was calculated to investigate task modulation on regional connectivity. c) To determine the behavioural relevance of whole-brain FNC state and regional FNC, we performed sparse partial least squares (sPLS) between FNC states dwell time (the average length of a subject having a certain state) and mental health symptoms and cognitions. Additionally, we compared behavioural variance explained by regional time-varying FNC versus static FNC using linear regression. d) The results were validated in an independent clinical dataset, including the reproducibility of FNC states and their associations with task modulation and behavioural outcomes.

Supplementary Materials and Methods, page 6, Behavioural variance explained by time-varying FNC and static FNC, Supplementary Materials

To examine the variance explained for reinforcement-related behaviours (Table S9) by regional time-varying FNC and static FNC, we extracted the 20 task-specific FNC (Fig. 5d) from the four FNC states of each participant (Figure 2a shows the cluster centroid of each FNC state). In parallel, we derived the same 20 task-specific regional FNC from static connectivity analysis (i.e., to calculate a connectivity strength using the time course of an entire scanning session, rather than from a segment of time window). To reduce the number of behavioural fitting parameters, we averaged the connectivity strength of the 20 task-specific regional FNC. Therefore, we obtain four state-based time-varying FNC, and one static FNC for every participant.

8. *The discussion felt a little long and repetitive. It would help the reader if the authors slimmed down the results summarizing.*

Response: We appreciated the Reviewer's suggestion. We have shortened the first paragraph of the Discussion to make it concise and clear. The length of this paragraph has changed from 180 words to 145 words.

Manuscript page 12, Discussion

By applying time-varying connectivity analyses to two large, behaviourally well-characterized datasets, we found two sets of complementary brain network features. The first set, a whole-brain network, is characterized by the transition between different FNC states according to ongoing task demands, which potentially reflect fluctuation of state of mind such as attention, engagement, motivation, and emotion processing. The second set of network features are regional and task-specific. By combining information from these two sets of networks, we found that time-varying FNC, compared to the commonly used static FNC, explains a higher amount of variance of reinforcement-related behaviours in the general population and is much more sensitive in detecting disease status in patients with depression and alcohol use disorder. These findings offer a more detailed understanding of how dynamic changes in brain connectivity are linked to behaviour and psychopathology, beyond what is captured by static neuroimaging analyses.

9. *In the discussion when the authors state - “These discoveries represent a major refinement in mechanistic characterizations of behaviour and psychopathology compared to standard neuroimaging analyses.” I’m not sure what the mechanism is they are referring to or how it has been refined, I would suggest either laying out the mechanism or avoiding such language.*

Response: We thank the Reviewer for this suggestion. We removed the last sentence in the first paragraph of Discussion, and changed it to:

Manuscript page 12, Discussion

These findings offer a more detailed understanding of how dynamic changes in brain connectivity are linked to behaviour and psychopathology, beyond what is captured by static neuroimaging analyses.

10. *Some of the figures such as Figure 3 and 4 and in many of the supplementary figures, the individual plots are very tiny and not legible. Please increase font and plot sizes or consider breaking them into several figures.*

Response: We thank the Reviewer for the suggestion. We have revised Figure 3, Figure 4, and

Fig. S4 to increase font and plot size.

Figure 3 Correlation between FNC state occurrence and task conditions for the EFT, MID and SST sessions. Correlation coefficients (r value) were shown on the left and compared between conditions. Occurrence of FNC states (grey line) and task conditions (coloured lines) as a function of scanning time were plotted on the right. For the **a)** EFT and **b)** MID, stimuli onset and duration were the same for all participants, thus we calculated the Pearson correlation between state occurrence and task stimuli at the group level. **The significance of correlation coefficients was compared with a null model by randomly shuffling the order of the FNC state label.** **c)** For the SST, as stimuli onsets differ across participants, we calculated the partial correlation between state occurrence and task stimuli (controlling for other types of stimuli) on the individual level. A random subject's state occurrence and stimuli onset were shown. Correlation coefficients were compared using one-sample t -test to determine their significances (Table S7). Asterisks indicate FDR-corrected statistical significance when comparing correlations between task stimuli and state occurrence with the null model (EFT and MID) or with zero (SST).

Figure 4 Sparse partial least squares (sPLS) analysis on each FNC state dwell time and behaviour questionnaires for resting-state and task-fMRI sessions. a) A multiple hold-out framework was applied to the sPLS analysis using the CCA/PLS Toolkit¹⁷. The statistical significance of association in the test set was determined by 5,000 iterative permutation procedures ($p < 0.05$). An example correlation between latent variables in the training and test set was plotted on the right. **b)** For significant sPLS models ($p < 0.05$), behavioural weights were plotted with a heatmap. Behavioural items were grouped into categories: internalizing symptoms, externalizing symptoms, substance use, and cognitive performances. The correlation value and significance of all sPLS models are listed in Table S10.

Behavioural questionnaire abbreviations: Development and Well-Being Assessment (DAWBA); Strengths and Difficulties Questionnaire (SDQ); Adolescent Depression Rating Scale (ADRS); European School Survey Project on Alcohol and Other Drugs (ESPAD); Alcohol Use Disorders Identification Test (AUDIT); Substance Use Risk Profile Scale (SURPS); Monetary-Choice Questionnaire (MCQ); Affective Go-Nogo task (AGN) (CANTAB, www.cambridgecognition.com); Cambridge Gambling Task (CGT) (CANTAB).

Fig. S4 Connectogram plot of the FNC states for the resting-state (REST) and three task-fMRI sessions (EFT, MID, SST). The strongest 100 connectivity of each state were shown.

Reviewer #2 (Remarks on code availability):

I reviewed the github with the code, I did not attempt to use any of the code myself. It appears as though the code is well commented. There are several changes that could help with code usability The code does not have a README nor a guide to how to use the scripts. The scripts have all paths hardcoded and it's not clear what the structure of the author's data and scripts are so including that in the README would be helpful too.

Response: We thank the Reviewer for the valuable suggestion. We have added a comprehensive README file that includes step-by-step instructions for using the code and data. Additionally, we have provided example data to demonstrate the main analyses performed in this project.

The revised code and data are available at the following repository:
https://github.com/xchang007/tvFNC_IMAGEN.git

A screenshot of the updated README file is attached below for reference.

tvFNC_IMAGEN

This repository contains code and data for the project "Neurocognitive characterization of behaviour and mental illness through time-varying brain network analysis"

code

- rest & task ICA**
GIFT toolbox (<https://trendscenter.org/software/gift/>)
Calhoun, V. D., Adalı, T., Pearlson, G. D. & Pekar, J. J. Hum. Brain Mapp. 14, 140-151 (2001)
- time-varying FNC**
GIFT toolbox – Temporal dFNC
- time-varying FNC corr and FNC network metrics**
code\S_FNC_states_network.m
data\R_FNC_states_network.mat
- FNC states corr task stimuli**
code\S_FNC_states_corr_task_EFT.m
data\R_FNC_states_corr_task_EFT.mat
example data EFT
- sPLS between state dwell time and behaviours**
sPLS analysis between FNC states' dwell time and 29 behavioural items using CCA/PLS Toolkit
Mihalik, A. et al. Biol. Psychiatry Cogn. Neurosci. Neuroimaging 7, 1055-1067 (2022).
code\S_run_sPLS.m
data\R_run_sPLS.mat
- FNC task specificity**
code\S_corr_SPM_regional_FNC.m
data\S_corr_SPM_regional_FNC.mat
- compare_state_based_Nfc_mdI**
code\S_compare_state_based_Nfc_mdI.m
data\R_compare_state_based_Nfc_mdI.mat
example data EFT

© 2025 GitHub, Inc. Terms Privacy Security Status Docs Contact Manage cookies Do not share my personal information

Reviewer #3 (Remarks to the Author):

The manuscript by Chang and colleagues characterizes the time-varying functional connectivity in three different cognitive tasks and investigates the relationship between the dynamic configurations during the task with behavioral and cognitive measures that indicate neuropsychiatric profile of the individuals. The study is exploratory in nature, which is fine given that the study uses a large dataset and they use multiple reinforcement-related tasks with clinically-relevant behavioral measures. The main methodological approaches regarding the derivation of time-varying functional connectivity is sound and consistent. However, there several major issues in the manuscript that should be addressed. In brief, a central point of the manuscript is the discrete FNC states, given the exploratory nature of their study, it is essential to characterize those states, which is not sufficiently done in the manuscript. Another general issue about the manuscript is that the relationship between multiple analysis approaches was not very established. Finally, there are some concerns regarding the some of the results that could be addressed by more rigorous approaches.

Response: We thank the Reviewer for the thoughtful evaluation of our manuscript and for providing constructive feedback to strengthen our work. In this revised version of the Manuscript, we have:

- 1) added description and additional analysis to show characteristics and behavioural relevance of the identified FNC states
- 2) revised the overview Figure 1 and strengthened the link between sections
- 3) provided additional analysis to test the robustness of our results

We provide a point-by-point response to each question below. Corresponding changes to the revised version of the Manuscript and Supplementary Materials are **marked in red**.

Major comment 1: The manuscript puts little emphasis on characterizing and defining what each state might indicate. For example, how each state is related to the global fluctuations? Do linearly increasing trend in state 3 is also exhibited in the resting-state? What are the proportions of the networks (and how they are similar/different to resting-state networks) at each state? What is the level of anti-correlations for each state and across which networks? Does high number states lead to very frequent jumps between states rather than more

continuous dwell times? Do some FNCs coincide with responses, or epochs in which head movement is more probable? The elbow criterion is a straight forward approach to decide the number of states. However, a systematic investigation of these states would affect the decision as well. So far, I see the dwell times of each state do not differ much between resting-state and across task scans, which raises doubts on their implication on task. Furthermore, figure 3 suggest that there are very subtle differences between states 2, 3 and 4, which is very difficult to figure out. Although there are a lot of material in supplementary text, they help understanding different number of states or sliding window lengths, but they do not help understanding what each FNC state might indicate.

Response: We provide a response to each question below:

1) “*how each state is related to the global fluctuations?*”

We thank the Reviewer for raising the question. To investigate the relationship between each state and global fluctuations, we calculated Pearson’s correlation between state occurrence and the global signal for each individual across the three task-fMRI sessions (**Author Response Figure 5**). Statistical significance was determined by one-sample t-test compared with 0. The results showed that state 3 in the EFT and SST exhibited positive correlations with the global signal (EFT: $\text{mean} \pm \text{std} = 0.05 \pm 0.20$, $t = 6.29$, $p = 6.51 \times 10^{-10}$, SST: $\text{mean} \pm \text{std} = 0.06 \pm 0.18$, $t = 7.37$, $p = 6.42 \times 10^{-13}$). Additionally, MID state 1 has a negative correlation with global signal ($\text{mean} \pm \text{std} = -0.01 \pm 0.13$, $t = -3.48$, $p = 0.0005$). No significant correlations were observed for the other states.

Author Response Figure 5. Correlation between subjects' global signal and state occurrences. Statistical significance was determined by one-sample t-test compared with 0. Asterisks indicate significant differences after FDR correction for multiple comparisons.

2) “Do linearly increasing trend in state 3 is also exhibited in the resting-state?”

The resting-state FNC state occurrence is shown below (**Author Response Figure 6**). The x-axis represents scanning time, and the y-axis is the state occurrence, calculated as the proportion of participants classified into a certain state. We fitted the state occurrence as a function of time (1,2,3...). Our analysis revealed that both FNC state 3 and state 4 showed a significant positive slope ($\beta = 0.01, 0.05, p = 6.01 \times 10^{-4}, 1.10 \times 10^{-66}$, respectively), suggesting a statistically significant linear increase over time. In resting-state, FNC state 3 showed an initial increase from 10.80% at the 1st time window to a peak of 18.77% at the 86th time window, but did not continue to increase until the end of the scanning session. In Fig S11, we found a continuous increase of state 3 in the EFT (increase from 0.87% to 12.51%) and SST (increase from 0.57% to 11.25%).

As we respond to **Major comment 5** below, during SST scanning, state 3 positively correlated with SST go too-late errors (mean $r = 0.14$, one-sample $t = 15.86, p = 2.88 \times 10^{-43}$, Table S7), which is an indicator of attention lapses or low vigilance in previous SST studies¹⁸. In general, state 3 has a higher occurrence in REST (15.86%) than EFT (6.31%) and SST (6.28%) across the whole scanning session (Figure 2a), and reaches a plateau in REST session.

Author Response Figure 6. FNC state occurrence in resting-state scanning session using a window length of 17.6s.

Fig. S11 FNC state occurrence using window length of 8.8s and 17.6s. The almost identical state occurrence fluctuation between different window lengths suggests that FNC state occurrence was not driven by the choice of window length.

3) *What are the proportions of the networks (and how they are similar/different to resting-state networks) at each state?*

We compared the proportion of each state within a scanning session in **Author Response Figure 7** below. Comparing the REST and task-fMRI sessions, we found that EFT and SST have a lower proportion (or fraction) of state 3 than REST, whereas other states showed a higher proportion than REST sessions. For MID, the fractions of states 1 and 4 are lower and higher than the counterparts in the REST session, respectively. The connectivity pattern of MID state 3 is dis-similar to other scanning sessions (Figure 2a), thus its fraction time may not be

comparable to other sessions.

Author Response Figure 7. The proportions (or fractions) of FNC states within a scanning session are shown and compared between resting-state and task-fMRI sessions using one-sample *t*-tests. FDR-corrected significant differences are marked with red lines.

The comparisons of connectivity strength of ICA components within each FNC state were calculated between REST and task-fMRI sessions, shown in **Author Response Figure 8** below. For each connectivity, we further divided the connectivity strength by the global mean of the participant in the particular scanning session to control for global differences, and compared the normalized connectivity strength between REST and task-fMRI sessions using one-sample *t*-tests. Multiple comparison was set as $p < 0.05/1830$ connectivity/12 comparisons. The significant *t*-statistics are shown in colors, with red representing higher connectivity strength in task versus resting-state, and blue representing the opposite. We can see that there is a significant variation in the proportion of connections in each state between REST and task-fMRI sessions (state 1: 48.58~57.16%; state 2: 39.78~54.81%; state 3: 22.46~79.56%; state 4: 43.17~52.79%), similar to previous findings^{2,19,20}. However, considering the length of the current manuscript, and the focus on the differences between task-fMRI sessions, we didn't highlight these comparisons with REST in the manuscript.

Author Response Figure 8. Comparison of connectivity strength between REST and task-fMRI sessions using one-sample t -tests. Multiple comparison was set as $p < 0.05/1830$ connectivity/12 comparisons. The significant t -statistics are shown in colors, with red representing higher connectivity strength in task versus resting-state, and blue representing the opposite.

4) *What is the level of anti-correlations for each state and across which networks?*

We showed the global mean negative connectivity (anti-correlation) of each FNC state of each scanning session in **Author Response Figure 5**. Mean negative connectivity strength and one-way ANOVA comparison across scanning sessions were shown in the **Author Response Table 1**.

Author Response Figure 5. Global mean positive and negative connectivity of each FNC state across scanning sessions

Author Response Table 1. Global mean negative connectivity and group comparisons across scanning sessions

	REST	EFT	MID	SST	ANOVA F	ANOVA p
State 1	0.10±0.02	0.08±0.02	0.07±0.01	0.08±0.02	445.55	0.000
State 2	0.15±0.03	0.12±0.02	0.12±0.02	0.12±0.02	597.40	0.000
State 3	0.19±0.03	0.18±0.03	0.12±0.02	0.17±0.03	1455.55	0.000
State 4	0.16±0.03	0.13±0.03	0.12±0.03	0.12±0.02	409.95	0.000

5) *Does high number states lead to very frequent jumps between states rather than more continuous dwell times?*

As the FNC states exhibit different frequencies of occurrence (e.g., REST FNC state 1 accounts for 48.56% of all FNC matrices, while state 3 accounts for only 15.86% in resting-state, Figure 2a), it naturally results in more transitions involving state 1 compared to other states.

To address this, we calculated the normalized number of transitions by dividing the number of times each state transitions to or from other states by its total occurrences during a scanning session (**Author Response Figure 9**). The results showed that, states 2 and 4 showed higher probabilities of transitioning to other states (out), as well as higher probabilities of transiting from other states (in) compared to states 1 and 3. This is consistent between resting-state and task-fMRI sessions.

When comparing states 2 and 4, state 4 showed a higher rate of transition out and transition in for the SST session, while other sessions did not show significant differences. For states 1 and 3, state 1 exhibited a higher rate of transition out (EFT and SST) and transition in (EFT) than state 3. The MID state 3 is dis-similar to other states (Figure 2a), so its transition rate is not comparable to other sessions.

Author Response Figure 9. Comparison of transitions across FNC states for resting-state and task-fMRI sessions. Red lines indicate significant differences between FNC states after FDR correction.

6) *Do some FNCs coincide with responses, or epochs in which head movement is more probable?*

Unfortunately, it is impractical to differentiate signals between stimuli and response time

courses in the present fMRI experiment design, because the time interval between target and response in MID and SST is about 1-2 seconds, shorter than a scanning TR = 2.2s and a sliding window length of 8.8-17.6s. Nevertheless, the participants' responses are unlikely to be the main drive of FNC states' occurrences. For example, in the EFT task, a passive viewing task with no response required, we still observed that the occurrences of FNC states strongly coincide with task stimuli.

Regarding head motion, we calculated the correlation between subjects' head motion (framewise displacement, FD) and state occurrences for each state. Overall, the correlations are minimal (**Author Response Figure 10**), except for the resting-state, of which the highest mean correlation coefficient is between head motion and state 4 (mean±std: 0.10±0.15).

We have carefully taken several steps to mitigate the influence of head motion on our results: 1) we first excluded subjects with excessive head motion (mean framewise displacement (FD) > 0.2 mm), to exclude the influence of excessive motion on our results (Table S1); 2) when calculating time-varying FNC, we regressed the six realign parameters as covariates in the sliding window analysis to minimize micro head movements on FNC estimation (Material and Methods); 3) further, in between-subject analyses, we regressed covariates including age, sex, recruitment sites, and head motion (mean FD). These procedures minimized the confounding effects of head motion.

Author Response Figure 10. Correlation between subjects' head motion (framewise displacement, FD) and state occurrences.

7) *The elbow criterion is a straight forward approach to decide the number of states. However,*

a systematic investigation of these states would affect the decision as well. So far I see the dwell times of each state do not differ much between resting-state and across task scans, which raises doubts on their implication on task.

We appreciate the reviewer's comments. We showed the reproducibility of FNC states and dwell time of each state using different window lengths (from 8.8s~70.4s, Fig S5-6, Table S5) and different clustering numbers (from 2 ~ 5 clusters, Fig S7-8, Table S6).

Regarding the dwell time of each state, we compared the dwell times of FNC states between the resting-state and task sessions using one-sample *t*-tests in overlapped participants (**Author Response Figure 11**). Similar to the comparison of the FNC fraction in **Author Response Figure 6** (fraction and dwell time are highly correlated measurements), EFT and SST have a lower dwell time of state 3 as compared to REST, whereas other states showed a longer dwell time as compared to REST session. For MID, dwell times are all lower than those in the REST session. This result can be partly replicated in the STRATIFY cohort.

We would also like to clarify that we do not claim the whole-brain FNC states are task-specific. Instead, the whole-brain FNC states may reflect mental states such as attention, engagement, motivation, and emotion processing, which fluctuate with task demands^{2,3}. While the regional FNC between two ICA networks is more sensitive to fluctuate with particular task conditions and showing task specificity (Figure 5).

Author Response Figure 11. Comparison of dwell times of FNC states between resting-state and task sessions using one-sample *t*-tests in overlapped participants. FDR corrected significant

differences between resting-state and task sessions were marked by red lines.

8) *Furthermore, figure 3 suggest that there are very subtle differences between states 2, 3 and 4, which is very difficult to figure out. Although there are a lot of material in supplementary text, they help understanding different number of states or sliding window lengths, but they do not help understanding what each FNC state might indicate.*

We thank the Reviewer for the suggestion. From Figure 2b, differences in modularity and participation coefficient between states 2, 3, and 4 are not significant. Yet the connectogram plot, which shows the strongest 100 connectivity of each state, may better illustrate differences between states (Fig. S4). From Fig. S4, FNC state 1 is the least modular state, with mainly weakly positive connectivity within functional domains. Both state 2 and state 3 exhibited segregation (strong negative connectivity) between primary regions (visual and sensorimotor networks) and higher-order regions (default mode and cognitive control networks), with state 3 exhibiting higher modularity and segregation between primary regions and subcortical and cerebellum. The connectivity profile of state 4 is similar to state 1 but with stronger connectivity within functional domains and a lower participation level. We agree with the Reviewer that the description of the connectivity profile helps us understand what each FNC state might indicate, and therefore, and have now added this to the revised Manuscript. Also, we increased the font size of Fig S4 for better visualization

Fig. S4 Connectogram plot of the FNC states for the resting-state (REST) and three task-fMRI sessions (EFT, MID, SST). The strongest 100 connectivity of each state were shown.

Manuscript page 6, Resting-state and task-fMRI sessions contain reoccurring FNC states, page

The connectogram plot showed the strongest 100 connectivity of each state (Fig. S4). FNC state 1 is the least modular state, with mainly weakly positive connectivity within functional domains. Both state 2 and state 3 exhibited segregation (strong negative connectivity) between primary regions (visual and sensorimotor networks) and higher-order regions (default mode, cognitive control networks), with state 3 exhibiting higher modularity and segregation between primary regions and subcortical and cerebellum. The connectivity profile of state 4 is similar to state 1 but with stronger connectivity within functional domains and a lower participation level.

Major comment 2: In Figure 2, the values in radial plot at panel b does not resemble those in the bar plots at panel c. For example, the average participation coefficients of state 2 and state 4 is very similar in panel c, but in the radial plot state 2 and 3 exhibit similar participation neg, whereas state 4 has a much higher values close to state 1. Similarly, modularity of state 4

appears to be around 3 in panel c, but it is around 2.5 in panel b. Please double check or explain why these differences appear in the figure.

Response: We apologize for the unclear description of Figures 2b and 2c. Original Figure 2b represents the network modularity and participation coefficients of the four FNC states **from the resting-state session** (as noted in the figure legend), while Figure 2c shows the modularity and participation coefficients of the FNC states **across both resting-state and task-fMRI sessions**. Therefore, the participation coefficients in Figure 2b (resting-state only) do not correspond directly to those in Figure 2c (combined sessions). In the revised Figure 2, we changed the Figure 2b with averaged metrics from both resting-state and task-fMRI sessions, to better match with values plotted in the Figure 2c.

For your reference, we have provided the modularity and participation coefficients of the FNC states in **Author Response Table 2** below.

Figure 2 Resting-state and task-fMRI sessions contain reoccurring FNC states. a) Four representative FNC states (cluster centroids) were derived from the *k-means* clustering applied to sliding-window FNC matrices from the resting-state (upper row) and three task-fMRI sessions (lower row). Each FNC state represents a reoccurring functional connectivity pattern between 61 ICA components, which are categorized into seven functional domains (Figure S2). The percentage of each state within a scanning session is shown in parentheses. Connectivity

profiles of the same state across resting-state and task-fMRI sessions exhibit a higher correlation (average $r=0.86\pm 0.17$) compared to correlations between different states (average $r=0.67\pm 0.10$, two-sample t -test: $t_{118}=7.17$, $p=6.93\times 10^{-11}$, 95% CI=[0.13, 0.24], Cohen's $d=1.63$).

b) The four FNC states differ in network modularity and participation coefficients, which quantify the level of segregation and integration between functional modules. **The modularity and participation coefficients are averaged across resting-state and task-fMRI sessions.** **c)** One-way ANOVA and post-hoc t -test comparisons between FNC states on modularity and participation coefficients. Red lines indicate significant differences between FNC states after FDR correction (Table S4).

Abbreviations for the seven functional domains: subcortical (SCN), temporal (TEP), sensorimotor (SMN), visual (VSN), cognitive control (CON), default mode (DMN) and cerebellar (CEB) networks.

Author Response Table 2. Network modularity and participation coefficients of FNC states from resting-state and task-fMRI sessions

Modularity	State 1	State 2	State 3	State 4
REST	2.14	3.20	3.53	2.72
EFT	2.14	2.94	3.32	3.08
MID	2.18	3.11	3.15	3.25
SST	2.30	3.04	3.46	3.27
Mean	2.19	3.07	3.37	3.08
Participation coef, positive FNC	State 1	State 2	State 3	State 4
REST	0.56	0.19	0.19	0.44
EFT	0.60	0.33	0.21	0.21
MID	0.65	0.32	0.24	0.19
SST	0.57	0.30	0.18	0.18
Mean	0.59	0.28	0.21	0.25
Participation coef, negative FNC	State 1	State 2	State 3	State 4
REST	0.76	0.19	0.16	0.54

EFT	0.81	0.31	0.17	0.23
MID	0.81	0.28	0.26	0.20
SST	0.75	0.29	0.15	0.19
Mean	0.78	0.27	0.19	0.29

Major comment 3: I am confused due to the continuous time courses of the state occurrences for EFT and SST tasks in Figure 3. As they used k-means clustering to identify time-varying FNC states, it is expected that for each centroid the occurrences would be in discrete times. I could not find any explanation to understand how that occurrences were calculated. I am guessing that the numbers indicate total occurrences across subjects (or probability of occurrences, which is not clear neither in the text nor in the figure).

Response: We thank the Reviewer for highlighting this issue. As the Reviewer noted, the time courses of state occurrences for the EFT and MID tasks are continuous because they represent total occurrences across subjects (i.e., probability of occurrences), which is feasible because the stimuli onset and duration for the EFT and MID tasks were identical for all participants, and thus we could conveniently calculate the correlations between task stimuli occurrences and state occurrences at the group level.

For the SST, however, as stimuli onsets varied across participants, we calculated the correlation between state occurrences and task stimuli occurrences at the individual level. Here, state occurrences were discrete (0 or 1). An example of a random subject's state occurrences and stimuli onsets is shown in Figure 3c.

The detailed description was provided in the **Supplementary Materials, page 5, Whole-brain FNC state occurrences correlated with task events and performances:**

“For EFT and MID tasks, experimental designs (stimuli sequence and onset time) are the same across all participants, and thus we calculate Pearson's correlation between occurrences of task events and FNC state at the group level. The occurrence of a task event was generated using stimuli onsets and duration convolved with canonical hemodynamic response function (HRF). The occurrence of a state at the group level was calculated as the proportion of subjects having the given state at a particular time point. Correlation coefficients were compared between conditions using Pearson and Filon's z-statistic from the R package ‘cocor’²⁸ (Fig. 3

and Table S8). For the SST, as stimuli onsets differ across subjects, we calculated the partial correlations between state occurrences and a given task stimulus, while controlling for other types of stimuli at the individual level. Correlation coefficients were compared to 0 using a one-sample t-test to determine their significances.”

In the revised Manuscript, we included a brief description on page 7:

FNC states are differentially modulated by ongoing task events

Having identified four stable FNC states, we examined how task events modulate their occurrences by calculating the Pearson correlation between FNC state occurrence and task conditions for each task session. For the EFT and MID, where stimuli onset and duration are the same for all participants, the state occurrences were calculated as the proportion of subjects having a given state at a particular time point at the group level. Pearson’s correlation was performed between state occurrences and task conditions at the group level, and compared with a null model to determine their significances (Supplementary Materials). We also compared the correlation coefficients between conditions (a contrast of interest) using Pearson and Filon's z-statistic ²¹. For the SST, as stimuli onsets varied across participants, we calculated the partial correlations between state occurrences and task stimuli (controlling for other types of stimuli) at the individual level. Correlation coefficients were compared to 0 using a one-sample t-test to determine their significances.

Major comment 4: The authors claim differences in correlations between FNC states and blocks that indicates certain features of the task. However, they do not show whether these correlations actually mean something significant at first place. Although for EFT task it is pretty clear that there is an association between detected states and task blocks, for the other two task is the correlation is very low and the evidence for a significant effect is not convincing. For example, although the authors claim a significant difference between large-win and small-win correlations of state 1 and 2, the correlations for MID task are very low $r < 0.2$. Indeed, a visual inspection of the task block suggest that MID task blocks appears to have a temporal structure, i.e. “no-win” trials are concentrated in the beginning of the session, “small-win” trials are concentrated at the middle and “large-win” trials are concentrated at the end of the session.

Whether or not this is imposed by chance, it might be imposing a low correlation, also given that the authors seem to report Pearson correlation in this analysis that is sensitive to outliers. The authors should have strictly control for any redundant correlations before claiming any association between FNC states and task blocks. This is even more clear in SST task. The all distributions presented in Figure 3c suggest that the average correlation between FNC states and task blocks might not be different than 0, maybe with an exception of state 3 vs. “go-too-late”. Before comparing the correlations across task blocks, the authors should test whether these distributions are significantly different than 0, preferably against a null model in which the correlations are calculated by random states with matching frequencies but time information is scrambled.

Response: We thank the Reviewer for the valuable suggestion. As suggested, we rigorously compared the empirical correlation between task stimuli and state occurrence with a null distribution generated from 5,000 permutations for the EFT and MID tasks. In each permutation, we randomly shuffled the order of the FNC state label (for example, the empirical state label is: 2, 1, 1, 3, 4,... for windowed FNC matrices of a subject, and the shuffled state label could be: 4, 2, 1, 3, 1,...). We then compared the empirical correlation with pseudo correlations between task stimuli and shuffled state occurrences. Permutation p-value is determined by the number of times that the absolute value of empirical correlation is greater than the absolute value of pseudo correlations, divided by 5,000 permutation times.

For the SST, as stimuli onsets differ across participants, we calculated the correlation between state occurrence and task stimuli at the individual level. Correlation coefficients were compared to 0 using a one-sample *t*-test to determine their significances (Table S7).

In the revised Figure 2, we indicate whether correlations between task stimuli and state occurrence are significantly different from the null model (EFT and MID) or significantly different from 0 (SST). Asterisks indicate FDR-corrected statistical significance corrected for multiple testing for 4 states \times 11 task stimuli across three tasks.

The three types of MID trials were randomized and were not intended to have any temporal structure. Indeed, if cutting the MID scanning time into halves, the first and second halves have no major difference in the construction of trials: the first half of the session (0~210s) contains 6, 7, 8 large-win, small-win and no-win trials, respectively, and the second half (210~420s)

contains 8, 7, 6 large-win, small-win and no-win trials, respectively. We, therefore, would not consider meaningful differences in the temporal order of trial types.

Manuscript page 7, FNC states are differentially modulated by ongoing task events

Having identified four stable FNC states, we examined how task events modulate their occurrences by calculating the Pearson correlation between FNC state occurrence and task conditions for each task session. **For the EFT and MID, stimuli onset and duration are the same for all participants, so state occurrence was calculated as the proportion of subjects having this state at a given time point at the group level. Pearson's correlation was performed between state occurrences and task conditions at the group level, and compared with a null model to determine their significances (Supplementary Materials).** We also compared the correlation coefficients between conditions (a contrast of interest) using Pearson and Filon's z -statistic²¹. **For the SST, as stimuli onsets varied across participants, we calculated the partial correlation between state occurrence and task stimuli (controlling for other types of stimuli) at the individual subject level. Correlation coefficients were compared to 0 using a one-sample t -test to determine their significances.**

In the EFT task (Figure 3a), state 1 and state 4 showed positive correlations with angry faces (all $r \geq 0.27$, $p < 0.0002$, Table S7) and negative correlations with non-face stimuli ($r \leq -0.55$, $p < 0.0001$). Comparing between conditions, state 1 and state 4 exhibited significantly higher correlations for faces stimuli (angry, happy and neutral faces) than non-face stimuli (all Pearson and Filon's $z \geq 6.79$, $p < 0.0001$, Table S8). In contrast, FNC state 2 showed strong positive correlation with non-faces stimuli ($r = 0.85$, $p < 0.0001$), and negative correlations with all types of faces stimuli ($r \leq -0.26$, $p < 0.0001$). Occurrence of state 3 linearly increased with scanning time and exhibited positive correlations with non-face stimuli ($r = 0.20$, $p = 0.007$).

In the MID task (Figure 3b), state 1 had a higher correlation with large-win than small-win trials ($z = 2.35$, $p = 0.02$, **Table S8**), whereas state 2 showed the inverse pattern again ($z = -2.79$, $p = 0.005$), **and negative correlations with large-win trials ($r = -0.20$, $p = 0.008$, Table S7).**

In the SST task (Figure 3c), stimuli onsets differ across subjects. We therefore calculated partial correlations between state occurrence and task stimuli at the individual subject level and compared correlations between conditions using one-sample t -test. We analysed stop signals

(stop-success and stop-failure) and go errors (go too-late and go-wrong). FNC state 1 showed a positive correlation with stop success ($t=2.60$, $p=0.01$, Table S7), whereas FNC state 2 exhibited a negative correlation ($t=-4.60$, $p=4.63 \times 10^{-6}$). For go-related errors, FNC state 3 showed a strong positive correlation with go too-late error ($t=15.86$, $p=2.88 \times 10^{-43}$), while other states exhibited negative correlations ($t < -4.01$, $p < 6.72 \times 10^{-5}$). State 1 and 4 showed positive correlations with go wrong errors ($t > 4.32$, $p < 1.72 \times 10^{-5}$) and state 2 showed a negative correlation ($t = -10.33$, $p = 5.45 \times 10^{-24}$).

Figure 3 Correlation between FNC state occurrence and task conditions for the EFT, MID and SST sessions. Correlation coefficients (r value) were shown on the left and compared between conditions. Occurrence of FNC states (grey line) and task conditions (coloured lines) as a function of scanning time were plotted on the right. For the **a)** EFT and **b)** MID, stimuli onset and duration were the same for all participants, thus we calculated the Pearson correlation between state occurrence and task stimuli at the group level. **The significance of correlation coefficients was compared with a null model by randomly shuffling the order of the FNC state label.** **c)** For the SST, as stimuli onsets differ across participants, we calculated the partial correlation between state occurrence and task stimuli (controlling for other types of stimuli) on the individual level. A random subject's state occurrence and stimuli onset were shown. Correlation coefficients were compared using one-sample t -test to determine their significances

(Table S7). Asterisks indicate FDR-corrected statistical significance when comparing correlations between task stimuli and state occurrence with the null model (EFT and MID) or with zero (SST).

Table S7 Correlation coefficients between FNC state occurrences with ongoing task events

	State 1	State 2	State 3	State 4
EFT (group-level)*				
Angry faces	$r = 0.27, p = 2.00e-4$	$r = -0.43, p < 1.00e-4$	$r = -0.13, p = 0.09$	$r = 0.41, p < 1.00e-4$
Neutral faces	$r = 0.39, p < 1.00e-4$	$r = -0.44, p < 1.00e-4$	$r = -0.13, p = 0.08$	$r = 0.16, p = 0.03$
Happy faces	$r = 0.15, p = 0.05$	$r = -0.26, p < 1.00e-4$	$r = -0.004, p = 0.96$	$r = 0.16, p = 0.03$
Non-face	$r = -0.61, p < 1.00e-4$	$r = 0.85, p < 1.00e-4$	$r = 0.20, p = 0.007$	$r = -0.55, p < 1.00e-4$
MID (group-level)				
Large win	$r = 0.17, p = 0.05$	$r = -0.20, p = 0.008$	$r = 0.11, p = 0.15$	$r = -0.001, p = 0.99$
Small win	$r = -0.12, p = 0.12$	$r = 0.14, p = 0.07$	$r = -0.01, p = 0.84$	$r = -0.04, p = 0.56$
No win	$r = -0.06, p = 0.43$	$r = 0.06, p = 0.43$	$r = -0.06, p = 0.42$	$r = 0.03, p = 0.68$
SST (individual-level, partial correlation)				
Stop success	$t = 2.60, p = 0.01$	$t = -4.60, p = 4.63e-6$	$t = 1.92, p = 0.06$	$t = 0.32, p = 0.75$
Stop failure	$t = -0.47, p = 0.64$	$t = -1.12, p = 0.26$	$t = -0.46, p = 0.64$	$t = 2.10, p = 0.04$

Go too-late	$t = -7.04, p = 5.32e-12$	$t = -8.50, p = 1.51e-16$	$t = 15.86, p = 2.88e-43$	$t = -4.01, p = 6.72e-5$
Go wrong	$t = 4.32, p = 1.72e-5$	$t = -10.33, p = 5.45e-24$	$t = -0.59, p = 0.56$	$t = 4.75, p = 2.27e-6$

*For EFT and MID tasks, experimental designs are the same across all participants, thus we calculate Pearson's correlation between occurrence of task events and group-averaged FNC state occurrence. For SST, stimuli onsets differ across subjects. We therefore calculated partial correlations between state occurrence and task stimuli at the individual subject level. Correlation coefficients were compared to 0 using a one-sample *t*-test to determine their significances.

FDR significant ($q < 0.05$) correlations are shown in **BOLD**.

Major comment 5: The authors claim that state 3 might be related to "attention lapses". Related to the previous comment that the authors put insufficient efforts to characterize each of these states, the authors do not present any evidence that might suggest such speculation plausible. In resting state studies similar states can be observed and they are typically seem to be related to changes in arousal or sleep. State 3 appears to be related to activation or deactivation of sensory networks. As I suggested before please check whether this state exhibit similar temporal structure in resting-state as well and carefully support any claims with evidence and/or reference to previous literature.

Response: We thank the Reviewer for the critical feedback regarding the interpretation of State 3. We hypothesized that state 3 is related to attention lapses (shifts of attention away from ongoing tasks) based on the finding that state 3 was positively correlated with SST go too-late errors (mean $r = 0.14$, one-sample $t = 15.86$, $p = 2.88 \times 10^{-43}$, Table S7). Previous SST studies have suggested that go too-late errors (or omission errors) are thought to reflect attention lapses or low vigilance¹⁸. Another study also found that SST go too-late errors were related to subjective reports of attention lapses (i.e., mind blanking)²². Other results of our study showed that in EFT and SST, the occurrence of state 3 linearly increases with scanning time (Fig S11), and across subjects, more SST go too-late errors are related to longer state 3 dwell time (the average length of a subject having a certain state) (Spearman's $\rho = 0.42$, $p = 9.03 \times 10^{-24}$, Fig. S10), consistent with our hypothesis. In our above reply to Major comment 1(2), in the resting state, FNC state 3 showed an initial increase from 10.80% at the 1st time window to a peak of

18.77% at the 86th time window but did not continue to increase until the end of the scanning session. This temporal structure is similar to those of FNC state 3 in the EFT and SST. However, as there are no behavioural measures in the resting state to assess participants' level of attention, we did not interpret the potential behavioural relevance of state 3 in the resting state.

Also, the connectivity profile of state 3 showed strong negative connectivity between primary regions (visual and sensorimotor networks) and higher-order regions (the default mode and cognitive control networks), subcortical and cerebellum. We have added the description in the revised manuscript and shown the connectogram plot in Figure S4.

Manuscript page 6, Resting-state and task-fMRI sessions contain reoccurring FNC states

The connectogram plot showed the strongest 100 connectivity of each state (Fig. S4). FNC state 1 is the least modular state, with mainly weakly positive connectivity within functional domains. Both state 2 and state 3 exhibited segregation (strong negative connectivity) between primary regions (visual and sensorimotor networks) and higher-order regions (default mode, cognitive control networks), while state 3 exhibited higher modularity and segregation between primary regions and subcortical and cerebellum. The connectivity profile of state 4 is similar to state 1 but with stronger connectivity within functional domains and a lower participation level.

Fig. S4 Connectogram plot of the FNC states for the resting-state (REST) and three task-fMRI sessions (EFT, MID, SST). The strongest 100 connectivity of each state was shown.

Major comment 6: The results that are presented in Figure 4 is not penetrable and it is not easy to match them to those described in the text. Possibly, there are some latent variables representing externalizing and internalizing symptoms that are correlated with each other. On the righthand side the regression plots suggest that there is a weak but significant effects. However, it is not possible to understand what is being plotted and that what the latent variable indicate. Also the weights for those behavioral abbreviations are difficult to make sense of, it is not possible to spot externalizing and internalizing symptoms. Moreover, the authors do not elaborate how these results make sense in light of other results. As far as I see there is no good correspondence between the results presented in Figure 2 and PLS results. For example despite strong correlations with task blocks with EFT, this task does not appear in PLS analysis. Also, this analysis shows that FNC state 3 also appears to be relevant in resting-state. This also supports the suspicion that the component what is thought to be attention lapse might be sleep/arousal-level related.

Response: We thank the Reviewer’s detailed critique, which highlights critical gaps in the clarity and interpretation of Figure 4. The motivation of this section is to find behavioural

correlates of each FNC state. We leveraged the extensive behavioural questionnaire data from the IMAGEN cohort to extract latent behavioural variables associated with FNC state dwell time (i.e., the average duration a subject remains in a given state).

Our findings indicate that FNC states 1 and 4 were primarily negatively associated with psychopathology and substance use behaviours, whereas states 2 and 3 showed mostly positive associations (Figure 4b). FNC state 2 was predominantly positively correlated with both internalizing and externalizing behaviours, while state 3 was mainly associated with substance use and related personality traits.

Direct comparisons between these behavioural associations and the temporal correlations with task stimuli in Figure 3 are not straightforward, as task-related neural responses do not directly correspond to the behavioural measures tested. However, we observed that FNC states 2 and 3 were negatively correlated with key task contrasts (e.g., emotional faces in EFT, large-win conditions in MID, and successful stops in SST), representing mental states that disengaged from the ongoing task requirement. These two states are mainly positively associated with psychopathology and substance use behaviours. This suggests a degree of consistency between behavioural associations and task-related processing.

To improve clarity, we have revised Figure 4 to better illustrate the analysis workflow and have grouped the behavioural measures into categories: internalizing symptoms, externalizing symptoms, substance use, and cognitive performances. We hope this revision enhances the interpretability of our results.

Figure 4 Sparse partial least squares (sPLS) analysis on each FNC state dwell time and behaviour questionnaires for resting-state and task-fMRI sessions. a) A multiple hold-out framework was applied to the sPLS analysis using the CCA/PLS Toolkit¹⁷. The statistical significance of association in the test set was determined by 5,000 iterative permutation procedures ($p < 0.05$). **An example correlation between latent variables in the training and test set was plotted on the right. b)** For significant sPLS models ($p < 0.05$), behavioural weights were plotted with heatmap. Behavioural items were grouped into categories: internalizing symptoms, externalizing symptoms, substance use, and cognitive performances. The correlation value and significance of all sPLS models are listed in Table S10.

Behavioural questionnaire abbreviations: Development and Well-Being Assessment (DAWBA); Strengths and Difficulties Questionnaire (SDQ); Adolescent Depression Rating Scale (ADRS); European School Survey Project on Alcohol and Other Drugs (ESPAD); Alcohol Use Disorders Identification Test (AUDIT); Substance Use Risk Profile Scale (SURPS); Monetary-Choice Questionnaire (MCQ); Affective Go-Nogo task (AGN) (CANTAB, www.cambridgecognition.com); Cambridge Gambling Task (CGT) (CANTAB).

Major comment 7: I am not sure if I understood very well the analysis regarding task contrasts presented in figure 5. As far as I understood, this analysis shows that other features of FNCs explain task related activity better than discrete states that are extracted by k-means clustering. Indeed, all these results points to a that trying to detect discrete whole-brain states might not be possible and instead there might be overlapping configurations of networks. So far I know, this is a common problem in many time-varying functional connectivity studies. Therefore it is not possible to expect the authors to address this issues completely. However, it is important to consider its implications, which might change the interpretation of the results. Moreover, from the distributions in figure 5, one would expect there would be negative connectivity patterns that are as strong as the positive ones. Why the authors did not include negative connections?

Response: As illustrated in the revised Figure 1b), our analysis not only examined the correlations between task stimuli and whole-brain FNC state occurrences (Figure 3) but also assessed the correlation between task stimuli and regional FNC connectivity strength (Figure 5).

Our results indicate that while FNC states are broadly modulated by task stimuli (Figure 3), they do not exhibit fine-grained, condition-specific modulations. For example, FNC states 1, 2, and 4 were correlated with face stimuli, but their correlations did not significantly differ between specific contrasts such as Angry vs. Neutral faces or Happy vs. Neutral faces (Table S8). In contrast, regional FNC connectivity strength demonstrated greater task specificity. For instance, in EFT, the correlation between regional connectivity strength and angry faces significantly differed from that of neutral faces (as determined by one-sample t-tests). Importantly, the connections showing high t-statistics in EFT did not exhibit similar differentiation in MID or SST conditions, suggesting that regional connectivity is more sensitive to specific task-related processes.

These findings suggest that whole-brain FNC states may reflect general mental states such as attention, engagement, motivation, and emotion processing, which fluctuate with task demands^{2,3}. In contrast, regional FNC connectivity appears to be more closely tied to specific task conditions, exhibiting finer-grained task sensitivity.

In Figure 5, we deliberately focused on positive contrasts: angry versus neutral faces in

EFT, large-win versus no-win in MID, and stop-success versus stop-failure in SST because they highlight connectivity patterns potentially related to emotional processing, reward sensitivity, and motor inhibition. While it is possible to examine negative contrasts (e.g., stronger connectivity for neutral faces compared to angry faces), such contrasts may not directly capture the cognitive or affective processes of interest.

We have revised Figure 1 to better illustrate the workflow of our analysis. We hope these revisions make the methodology and findings more comprehensible to readers.

a) Time-varying functional network connectivity (FNC) from resting-state and task-based fMRI

b) Task modulation on whole-brain FNC state occurrence and regional connectivity

c) Associations between time-varying or static FNC and behaviours

d) Validation in external STRATIFY cohort

Figure 1 Summary of analysis strategy. a) Resting-state (REST) and three reinforcement-related task-fMRI of the IMAGEN cohort were pre-processed, quality controlled, and

parcellated into 61 components using the group independent component analysis (ICA). Time-varying functional network connectivity (FNC) between each pair of the 61 components was estimated using the sliding window analysis. The window length is 17.6s for the REST, emotional faces task (EFT) and stop-signal task (SST) sessions, and is 8.8s for the monetary incentive delay task (MID) to accommodate the block duration of MID (~10 s). **b)** On the whole-brain level, *k*-means clustering analysis was applied to windowed FNC matrices to derive reoccurring FNC states from each scanning session. FNC state occurrences (i.e., the proportion of participants classified into a given state) were correlated with ongoing task stimuli to investigate how task conditions modulate whole-brain FNC. On the regional level, connectivity strength between each pair of ICA components was assessed. Correlation between regional connectivity strength and task stimuli was calculated to investigate task modulation on regional connectivity. **c)** To determine the behavioural relevance of whole-brain FNC state and regional FNC, we performed sparse partial least squares (sPLS) between FNC states dwell time (the average length of a subject having a certain state) and mental health symptoms and cognitions. Additionally, we compared behavioural variance explained by regional time-varying FNC versus static FNC using linear regression. **d)** The results were validated in an independent clinical dataset, including the reproducibility of FNC states and their associations with task modulation and behavioural outcomes.

Minor comment: Please add a legend to Figure 2 indicating the abbreviations used for each network as in supplementary figure 4.

Response: We have added the abbreviations of the seven domains of network to Figure 2.

Figure 2 Resting-state and task-fMRI sessions contain recurring FNC states. a) Four representative FNC states (cluster centroids) were derived from the *k-means* clustering applied to sliding-window FNC matrices from the resting-state (upper row) and three task-fMRI sessions (lower row). Each FNC state represents a recurring functional connectivity pattern between 61 ICA components, which are categorized into seven functional domains (Figure S2). The percentage of each state within a scanning session is shown in parentheses. Connectivity profiles of the same state across resting-state and task-fMRI sessions exhibit a higher correlation (average $r=0.86\pm0.17$) compared to correlations between different states (average $r=0.67\pm0.10$, two-sample *t*-test: $t_{118}=7.17$, $p=6.93\times 10^{-11}$, 95% CI=[0.13, 0.24], Cohen's $d=1.63$). b) The four FNC states differ in network modularity and participation coefficients, which quantify the level of segregation and integration between functional modules. The modularity and participation coefficients are averaged across resting-state and task-fMRI sessions. c) One-way ANOVA and post-hoc *t*-test comparisons between FNC states on modularity and participation coefficients. Red lines indicate significant differences between FNC states after FDR correction (Table S4).

Abbreviations for the seven functional domains: subcortical (SCN), temporal (TEP), sensorimotor (SMN), visual (VSN), cognitive control (CON), default mode (DMN) and cerebellar (CEB) networks.

Minor comment: In figure 3, the time-courses of the first state suggest that indeed this state is almost perfectly in line with presentation of faces regardless of the emotional content. It is clear that if they merge all face trials, the correlations with state 1 would drastically increase.

Response: As the Reviewer correctly pointed out, the EFT state occurrences are highly correlated with face stimuli, regardless of their emotional content. We have provided the Pearson's correlation coefficients and p-values between state occurrences and face stimuli (aggregated across angry, neutral, and happy faces) as well as control stimuli in the **Author Response Table 3** below.

In the manuscript, we have retained the original results, including the more detailed correlations with each type of face stimulus.

Author Response Table 3 Pearson's correlation between state occurrences and face stimuli (aggregated across angry, neutral, and happy faces) as well as control stimuli for the emotional faces task (EFT).

	r_faces	p_faces	r_control	p_control
State 1	0.6061	7.82E-20	-0.6060	7.88E-20
State 2	-0.8500	1.49E-52	0.8500	1.51E-52
State 3	-0.1993	6.67E-03	0.1989	6.79E-03
State 4	0.5491	6.98E-16	-0.5484	7.70E-16

Minor comment: In the field of time-varying fMRI, multiple terminologies appears, which is inconvenient. This issue is not addressed well in the paper. One important distinction is that many papers that the authors discussed in the introduction deals with the functional connectivity that defines a cortical area through contiguous parcels based on anatomical and/or functional architecture. As far as I understood, the authors especially chose the term functional network connectivity (FNC) to emphasize the fact that they measure the interactions between functional networks from ICA instead of cortical areas. It is important to clearly distinguish these differences throughout the manuscript.

Response: We thank the Reviewer for highlighting this issue. Indeed, we use the term “functional network connectivity (FNC)” to emphasize the connectivity was calculated between ICA functional networks, consistent with previous studies using ICA as a brain parcellation method¹⁵. We have added this clarification in the Introduction and the first paragraph of Results section in the revised manuscript.

Manuscript page 4, Introduction

Reinforcement-related cognitive processes **engage interactions between brain networks, often measured as functional network connectivity (FNC)** derived from functional magnetic resonance imaging (fMRI) during execution of cognitive tasks.

Manuscript page 5, Summary of analysis strategy

We analyzed resting-state and three reinforcement-related task-fMRI of 1,417 participants of the IMAGEN cohort¹ at age 19 years (Table S1-S2). (i) Time-varying functional network connectivity (FNC) of each scanning session was derived using sliding window analysis¹³ (Figure 1a) on time-series of 61 components from group independent component analysis (ICA)¹⁴ (Figure S2 and Table S3). **In this study, we used the term FNC to refer to the correlation between ICA component time-courses¹⁵;**

Reviewer #4 (Remarks to the Author):

Response: We thank the Reviewer for the time and effort in reviewing our manuscript. We have provided a point-by-point response to each question above and hope that the Reviewer finds our revisions satisfactory.

References

1. Schumann, G. *et al.* The IMAGEN study: reinforcement-related behaviour in normal brain function and psychopathology. *Mol. Psychiatry* **15**, 1128–1139 (2010).
2. Gonzalez-Castillo, J. & Bandettini, P. A. Task-based dynamic functional connectivity: Recent findings and open questions. *Neuroimage* **180**, 526–533 (2018).
3. Shine, J. M. *et al.* The Dynamics of Functional Brain Networks: Integrated Network States during Cognitive Task Performance. *Neuron* **92**, 544–554 (2016).
4. Quinlan, E. B. *et al.* Identifying biological markers for improved precision medicine in psychiatry. *Mol. Psychiatry* **25**, 243–253 (2020).
5. Xie, C. *et al.* A shared neural basis underlying psychiatric comorbidity. *Nat. Med.* **29**, 1232–1242 (2023).
6. Wise, R. A. Dopamine, learning and motivation. *Nat. Rev. Neurosci.* **5**, 483–494 (2004).
7. Collins, A. G. E. Reinforcement learning: bringing together computation and cognition. *Curr. Opin. Behav. Sci.* **29**, 63–68 (2019).
8. Etkin, A., Büchel, C. & Gross, J. J. The neural bases of emotion regulation. *Nat. Rev. Neurosci.* **16**, 693–700 (2015).
9. CARDINAL, R. N., WINSTANLEY, C. A., ROBBINS, T. W. & EVERITT, B. J. Limbic Corticostriatal Systems and Delayed Reinforcement. *Ann. N. Y. Acad. Sci.* **1021**, 33–50 (2004).
10. Mikl, M. *et al.* Effects of spatial smoothing on fMRI group inferences. *Magn. Reson. Imaging* **26**, 490–503 (2008).
11. Rubinov, M. & Sporns, O. Complex network measures of brain connectivity: Uses and interpretations. *Neuroimage* **52**, 1059–1069 (2010).
12. Newman, M. E. J. Fast algorithm for detecting community structure in networks. *Phys. Rev. E* **69**, 1–5 (2004).
13. Allen, E. A. *et al.* Tracking Whole-Brain Connectivity Dynamics in the Resting State. *Cereb. Cortex* **24**, 663–676 (2014).
14. Calhoun, V. D., Adali, T., Pearlson, G. D. & Pekar, J. J. A method for making group inferences from functional MRI data using independent component analysis. *Hum.*

-
- Brain Mapp.* **14**, 140–151 (2001).
15. Calhoun, V. D., Miller, R., Pearlson, G. & Adalı, T. The Chronnectome: Time-Varying Connectivity Networks as the Next Frontier in fMRI Data Discovery. *Neuron* **84**, 262–274 (2014).
 16. Sporns, O. & Betzel, R. F. Modular Brain Networks. *Annu. Rev. Psychol.* **67**, 613–640 (2016).
 17. Mihalik, A. *et al.* Canonical Correlation Analysis and Partial Least Squares for Identifying Brain–Behavior Associations: A Tutorial and a Comparative Study. *Biol. Psychiatry Cogn. Neurosci. Neuroimaging* **7**, 1055–1067 (2022).
 18. Wright, L., Lipszyc, J., Dupuis, A., Thayapararajah, S. W. & Schachar, R. Response inhibition and psychopathology: A meta-analysis of go/no-go task performance. *J. Abnorm. Psychol.* **123**, 429–439 (2014).
 19. Kaufmann, T. *et al.* Task modulations and clinical manifestations in the brain functional connectome in 1615 fMRI datasets. *Neuroimage* **147**, 243–252 (2017).
 20. Cole, M. W., Bassett, D. S., Power, J. D., Braver, T. S. & Petersen, S. E. Intrinsic and Task-Evoked Network Architectures of the Human Brain. *Neuron* **83**, 238–251 (2014).
 21. Diedenhofen, B. & Musch, J. cocor: A Comprehensive Solution for the Statistical Comparison of Correlations. *PLoS One* **10**, e0121945 (2015).
 22. Andrillon, T., Burns, A., Mackay, T., Windt, J. & Tsuchiya, N. Predicting lapses of attention with sleep-like slow waves. *Nat. Commun.* **12**, 1–12 (2021).

Point-by-point response

Reviewer #1 (Remarks to the Author):

The authors have addressed several of the previous comments and revised the manuscript accordingly, resulting in improved clarity and presentation. However, a number of issues remain that would benefit from further revision to enhance the overall rigor and coherence of the study:

Response:

We thank the reviewer for the thorough assessment and the constructive suggestions to improve our manuscript. In this revised manuscript, we have:

- 1) reperformed main results in each recruitment sites and provided quantitative measures to show similarity to main results;
- 2) reported the validation results using STRATIFY patients with M.I.N.I. confirmed diagnosis and to perform analysis in each patient and control groups separately;
- 3) assessed potential influence of head motion on our results;

We provide a point-by-point response to each question below. Corresponding changes to the revised version of the Manuscript and Supplementary Materials are **marked in red**.

1. To address potential site effects, the authors calculated spatial correlations between FNC states derived from each site and those from all other sites combined, reporting similar patterns that suggest general consistency across sites. While this provides some supportive evidence, it does not directly address whether the subsequent statistical analyses were affected by site effects. A more robust and widely accepted approach would be to apply harmonization techniques, such as ComBat, to explicitly account for site-related variability in the FNC data. It would also be valuable to test whether the main findings are reproducible across individual sites. Additionally, although the authors plotted state occurrence per site and noted a high similarity with the overall results, including quantitative similarity measures would strengthen this conclusion.

Response: To strengthen our assessment of site effects, we have revised the Fig. S19 to include

quantitative correlation values for both the FNC states and their occurrences, derived separately from each of the eight recruitment sites. The results demonstrate that the FNC states from individual sites were highly similar to those derived from the other sites, and their occurrences showed moderate to large consistency across sites (Fig. S19).

To further address the potential influence of site effects, we replicated the main analyses within each recruitment site individually, including: (i) the correlation between FNC state occurrence and task conditions (Fig. S20); (ii) the correlation between FNC state dwell time and task performance (Fig. S21); and (iii) the behavioral variance explained by time-varying and static FNC (Fig. S22). The main findings were reproducible, supporting the robustness of our results across recruitment sites.

Manuscript page 12, Sensitivity analysis

To assess whether the results were influenced by potential site effects (Table S2), we first extracted the four FNC states separately for each of the eight recruitment sites (Fig. S19). We then calculated the spatial correlation of the FNC states and their occurrences between each individual site and the combined data from all other sites. The results showed that the FNC states derived from individual sites were highly similar to those from the other sites, and their occurrences exhibited moderate to large consistency across sites. Furthermore, we replicated the main analyses within each recruitment site, including: (1) the correlation between FNC state occurrence and task conditions (Fig. S20); (2) the correlation between FNC state dwell time and task performance (Fig. S21); and (3) the behavioral variance explained by time-varying and static FNC (Fig. S22). The main findings were reproducible, supporting the robustness of our results across recruitment sites.

Supplementary Materials

Fig. S19 Influence of recruitment site effect on FNC states. a) Four FNC states were identified separately for each of the eight recruitment sites. As an example, the cluster centroids of FNC states from recruitment site 1 are shown. b) To quantitatively assess the similarity of FNC states across sites, we computed the spatial correlation between the FNC states derived from each site and those derived from all sites combined. The bar plot shows the mean correlation for each

site, represented in different colors. Mean correlation values are indicated above each bar, and error bars represent the standard deviation across the four states. The results demonstrate that FNC states derived from individual sites were highly similar to those from all sites combined.

c) Correlation of FNC state occurrences between each individual site and those from all sites combined. For each site, the mean and standard deviation of the correlations across the four states are shown. Although there was greater variability in state occurrence correlations across sites, the mean correlations ranged from 0.34 to 0.82, indicating moderate to large consistency.

a) Example of four FNC states from recruitment site 1

b) Correlation of FNC states between sites

c) Correlation of FNC occurrences between sites

Fig. S20 Replication of the correlation between FNC state occurrence and task conditions across recruitment sites. Correlation coefficients (r values) were computed using the same method as in the main results shown in Figure 3. The bar plot shows the mean and standard deviation of the correlation coefficients for each site, with individual site-level values indicated by dots. A fixed-effect meta-analysis was conducted to assess the overall significance across sites. Asterisks indicate FDR-corrected statistical significance across all three tasks and all conditions.

The main results from Figure 3 are shown below for references:

Figure 3 Correlation between FNC state occurrence and task conditions for the EFT, MID and SST sessions. Correlation coefficients (r value) were shown on the left and compared

between conditions. Occurrence of FNC states (grey line) and task conditions (coloured lines) as a function of scanning time were plotted on the right. For the **a)** EFT and **b)** MID, stimuli onset and duration were the same for all participants, thus we calculated the Pearson correlation between state occurrences and task stimuli at the group level. The significance of correlation coefficients was compared with a null model by randomly shuffling the order of the FNC state label. **c)** For the SST, as stimuli onsets differ across participants, we calculated the partial correlation between state occurrence and task stimuli (controlling for other types of stimuli) on the individual level. A random subject's state occurrence and stimuli onset were shown. Correlation coefficients were compared using one-sample *t*-test to determine their significances (Table S7). Asterisks indicate FDR-corrected statistical significance when comparing correlations between task stimuli and state occurrence with the null model (EFT and MID) or with zero (SST).

REDACTED

Fig. S21 Replication of the correlation between FNC state dwell time and task performance across recruitment sites. The main results (Fig. S10) showed that go-too-late errors in the SST task were positively associated with dwell time in state 3 and negatively associated with dwell time in state 1. Correlation coefficients were computed separately for each recruitment site. The bar plot displays the mean and standard deviation of the correlation coefficients across sites, with individual site-level values represented as dots. A fixed-effect meta-analysis was conducted to assess the overall significance across sites. Asterisks indicate FDR-corrected statistical significance.

The main results from Fig.S10 are shown below for references:

Fig. S10 Spearman's correlation between FNC state dwell time and task performances in MID and SST. After FDR, SST go too-late error showed a positive association with state 3 (Spearman's $\rho=0.42$, $p=9.03 \times 10^{-24}$) and a negative association with state 1 ($\rho=-0.10$, $p=9.29 \times 10^{-4}$).

Fig. S22 Replication of behavioural variance explained by time-varying and static FNC across recruitment sites. Using linear regression models, we assessed the variance explained (R^2) by time-varying FNC (blue) and static FNC (green) using linear regression model across the 23 behavioral items, as reported in the main results (Figure 6a). The bar plot shows the mean and standard deviation of the R^2 values across the recruitment sites.

The main results from Figure 6a are shown below for references:

Figure 6 Behavioural variance explained by time-varying FNC and static FNC from the EFT, MID and SST sessions. The same set of 20 task-specific connectivity identified from the previous section was taken from time-varying FNC states and static FNC. **a)** In the IMAGEN cohort, linear regression of 29 behaviours (Table S9) were regressed as a function of time-varying FNC (blue), static FNC (green), or both (gray). Model significance is obtained by comparing the full model with null models (only constant term). Among the 23 significantly

fitted behaviours (model significance uncorrected $p_{\text{one-tail}} < 0.05$, Table S12), 22 behaviours showed a higher R^2 using time-varying FNC than static FNC regression model.

a) Behavioural variance explained in IMAGEN

2. *The STRATIFY sample includes participants with clinical diagnoses; however, the current analyses were performed on the full sample. To assess whether the findings are consistent across different diagnostic groups, it is recommended that the authors report FNC state results and their associations with task stimuli and behavioral performance separately for each group.*

Response: In the revised manuscript, we have reported all relevant results separately for the STRATIFY control (CON), major depressive disorder (MDD), and alcohol use disorder (AUD) groups. Specifically: (1) the correlations between FNC states from the IMAGEN resting-state and STRATIFY groups were shown in Fig. S14; (2) the associations between FNC state occurrences and task conditions within STRATIFY CON, MDD and AUD groups were presented in Fig. S15; (3) the correlations between FNC state dwell time and task performances

in each groups were provided in Table S14; (4) behavioral regression analyses using both time-varying and static FNC in the STRATIFY CON group are shown in Figure 6c, and logistic regression analyses distinguishing CON from MDD, AUD, and all patients (PAT) are presented in Figure 6d.

Supplementary Materials

Fig. S14 FNC states derived from the STRATIFY cohort and their spatial correlations with FNC states derived from the IMAGEN resting-state (upper row). Pearson's correlation between FNC matrices of the same state ranges from 0.80 ~ 0.96, showing a high-level of correspondence. Similar to the IMAGEN dataset, we couldn't extract the FNC state 3 from the MID session as other sessions.

Fig. S15 Correlation between FNC state occurrence and task conditions in STRATIFY CON, MDD and AUD groups. For the a) EFT and b) MID, Pearson correlation between state occurrences and task stimuli were calculated at the group level similar to the main analysis (Figure 3). c) For the SST, we calculated the partial correlation between state occurrence and task stimuli (controlling for other types of stimuli) on the individual level. Asterisks indicate FDR-corrected statistical significance across the three tasks.

a) EFT (group level)

b) MID (group level)

c) SST (individual level)

Table S14 Replication of correlation between FNC states dwell time with SST go too-late error in the STRATIFY cohort. Spearman's rank correlation was performed as task performance do not follow normal distribution.

	State 1	State 3
Healthy controls (CON)	$\rho = -0.26, p = 0.01$	$\rho = 0.31, p = 0.003$
Major depressive disorder (MDD)	$\rho = -0.14, p = 0.25$	$\rho = 0.28, p = 0.02$
Alcohol use disorder (AUD)	$\rho = -0.05, p = 0.70$	$\rho = 0.34, p = 0.004$

FDR significant ($q < 0.05$) correlations are shown in **BOLD**. Note that the p -values are larger than correlations in the IMAGEN cohort probably due to differences in sample size (IMAGEN $n=1218$; STRATIFY CON, MDD and AUD $n=183, 131, 125$). Yet Spearman's ρ was in the same direction as the IMAGEN cohort.

Figure 6 Behavioural variance explained by time-varying FNC and static FNC from the EFT, MID and SST sessions. The same set of 20 task-specific connectivity identified from the previous section was taken from time-varying FNC states and static FNC. **a)** In the IMAGEN cohort, linear regression of 29 behaviours (Table S9) were regressed as a function of time-varying FNC (blue), static FNC (green), or both (gray). Model significance is obtained by comparing the full model with null models (only constant term). Among the 23 significantly fitted behaviours (model significance uncorrected $p_{\text{one-tail}} < 0.05$, Table S12), 22 behaviours showed a higher R^2 using time-varying FNC than static FNC regression model. **b)** As an example, we showed one behavioural fitting using time-varying FNC and static FNC from each task. **c)** The same model fitting and comparison procedure were applied in the STRATIFY healthy controls. For the behavioural items that are available in STRATIFY (Table S9), time-

varying FNC has a higher R^2 using time-varying FNC in 14 out of 19 behavioural items. **d)** Further, the disease status of STRATIFY patients and controls was regressed as a function of time-varying FNC and static FNC using logistic regression. All models showed a numeric higher R^2 using time-varying FNC than static FNC regression model.

Abbreviations: CON: healthy controls; AUD: alcohol use disorder; MDD: major depressive disorder; PAT: all patients. Covariates of age, sex, recruitment sites and head motion were regressed before analysis.

3. *The diagnostic criteria for clinical group inclusion in the STRATIFY cohort should be described more precisely. For instance, it is unclear whether MDD was diagnosed using DSM-IV criteria or another standardized approach. The current description—"patients with MDD were included if they exhibited current and acute depressive symptoms, with a PHQ-9 score of 15 or above"—does not constitute a formal diagnosis and would benefit from additional clarification.*

Response: We agree that the Patient Health Questionnaire-9 (PHQ-9) and Alcohol Use Disorders Identification Test (AUDIT) alone do not constitute formal diagnostic tools. To ensure diagnostic validity, we included Mini International Neuropsychiatric Interview (M.I.N.I.), version 5.0.0 as an addition criterion to confirm their diagnoses. M.I.N.I was administered as a screening questionnaire at the time of recruitment.

In the revised manuscript, we now clarify that MDD patients were included only if they met both of the following criteria: (i) had a PHQ-9 score ≥ 15 and (ii) met diagnostic criteria for a current major depressive episode, confirmed by scoring "1" (i.e., "yes") on at least one of the following M.I.N.I. modules: MINI_A_MDEC (Major Depressive Episode – Current), MINI_A_MDEMFC (Major Depressive Episode with Melancholic Features – Current), or MINI_A_MDER (Major Depressive Episode – Recurrent). Similarly, participants with AUD were included if they (i) had an AUDIT score ≥ 15 and (ii) met the diagnostic criteria for alcohol dependence or harmful alcohol use, confirmed by a score of "1" on one of the relevant M.I.N.I. modules: MINI_J_AAC (Alcohol Abuse - Current) or MINI_J_ADC (Alcohol Dependence -

Current). Healthy controls were excluded if they met criteria for a major depressive episode, alcohol dependence, or harmful alcohol use on the M.I.N.I.

As a result of this additional diagnostic verification step, the final sample sizes were reduced to: AUD ($n = 125$), MDD ($n = 131$), and CON ($n = 183$), from initial totals of AUD ($n = 150$), MDD ($n = 168$), and CON ($n = 210$). All relevant results, including those referenced in Point 2, have been updated accordingly. The findings remain unchanged. We have now clarified this diagnostic procedure in the revised manuscript and included a detailed description in the Supplementary Materials.

Manuscript page 10,

Validation in an independent dataset

We validated our results in the independent STRATIFY dataset which includes patients with alcohol use disorder (AUD, $n=125$), major depressive disorder (MDD, $n=131$) and healthy controls (CON, $n=183$, Table S13)^{34,35}. Patients with MDD were included if they (i) had a Patient Health Questionnaire-9 (PHQ-9) score ≥ 15 , and (ii) met the diagnostic criteria for a current major depressive episode. AUD patients were included if they (i) scored ≥ 15 on the Alcohol Use Disorders Identification Test (AUDIT), and (ii) met the diagnostic criteria for alcohol dependence or harmful alcohol use (Supplementary Materials).

Supplementary Materials, page 6,

Replication of findings in the independent STRTIFY cohort STRATIFY cohort

The STRATIFY cohort^{31,32} recruited patients (ages 19-25) with eating disorder, alcohol use disorder, major depression and healthy controls from three recruitment sites in Berlin, London and Southampton, of which Berlin and London are also recruitment sites for IMAGEN. For the current study, we included participants with alcohol use disorder (AUD, $n = 125$), major depressive disorder (MDD, $n = 131$), and healthy controls (CON, $n = 183$).

Patients with MDD were included if they (i) exhibited current and acute depressive symptoms, with a Patient Health Questionnaire-9 (PHQ-9) score ≥ 15 , and (ii) met the diagnostic criteria for a current major depressive episode, confirmed by scoring “1” on at least

one of the following M.I.N.I. modules: MINI_A_MDEC, MINI_A_MDEMFC, or MINI_A_MDER.

Patients with AUD were included if they (i) scored ≥ 15 on the Alcohol Use Disorders Identification Test (AUDIT), and (ii) met the diagnostic criteria for alcohol dependence or harmful alcohol use, confirmed by scoring “1” on the MINI_J_AAC or MINI_J_ADC modules.

Healthy controls were excluded if they met criteria for any of the above diagnoses. Demographic information is provided in Table S13. MRI acquisition parameters and task designs in STRATIFY were harmonized with the IMAGEN cohort to maximize comparability between datasets. Additional details on study protocols are available at: <https://stratify-project.org/documentation/>.

Table S13 Demographic information of STRATIFY participants

	Alcohol use disorder (AUD, $n = 125$)	Major depressive disorder (MDD, $n = 131$)	Healthy controls (CON, $n = 183$)	p	F or chi^2
Age	22.19±2.06	22.25±2.26	22.11±1.27	0.80	0.23
Sex (M/F)	49/76	31/100	68/115	0.01	8.58
Recruitment Sites	5/82/38	10/71/50	41/93/49	5.42e-06	29.78

Fig. S16 MID and SST task performances of STRTIFY CON, MDD and AUD patients. Red lines indicate significant group differences after FDR correction.

Fig. S18 Post-hoc group comparison of connectivity strength of time-varying FNC and static FNC between CON, MDD, and AUD patients. Two-sample t -tests were performed between groups to compare the connectivity strength of each state in time-varying FNC and static FNC across groups. Red lines indicate significant differences after FDR correction for multiple comparisons.

4. On page 10, the authors compare task performance across MDD, AUD, and control groups, stating, for example, “For MID task performances, MDD patients exhibited lower accuracy and longer reaction time (RT) compared to CON, ...”. While informative, this content appears in a section focused on correlations between FNC state occurrence and ongoing task stimuli, which disrupts the logical flow. It is recommended that this paragraph be relocated to a section more appropriate for reporting behavioral performance results.

Response: As recommended, we have separated the section related to task performances correlation with FNC state dwell time into a distinct paragraph for improved clarity and logical

flow.

Manuscript page 11,

Our replication of the correlation between occurrences of FNC state and ongoing task stimuli (Fig. S15) yielded results similar to the IMAGEN dataset (Figure 3 and Table S8). Specifically, in the EFT, states 1 and 4 positively correlated with faces stimuli, while state 2 positively correlated with non-faces stimuli. MID state 2 negatively correlated with large-win condition, but this was observed only in MDD patients. SST state 3 positively correlated, and state 2 negatively correlated with go too-late errors across all three groups.

In replicating the correlation between FNC state dwell time and task performances (Fig. S10), we confirmed significant associations between SST go too-late errors with dwell time in state 1 ($\rho=-0.26, p=0.01$) and state 3 ($\rho=0.31, p=0.003$) within the STRATIFY CON group (Table S14). Among individuals with MDD and AUD, SST go too-late errors were positively correlated with dwell time in state 3 ($\rho=0.28, 0.34; p=0.02, 0.004$ respectively). Group comparisons of task performance showed that MDD patients exhibited lower accuracy and longer reaction time (RT) during the MID task compared to CON, while AUD patients showed only longer RT relative to CON (Fig. S16). For SST task performances, AUD patients had longer RT in response to stop signals, and MDD patients exhibited longer RT for go signals.

5. *There appears to be an inconsistency in the description of preprocessing procedures. The response letter states that both resting-state and task-based fMRI data underwent the same standard preprocessing, whereas the Supplementary Materials describe different pipelines for each. This discrepancy should be resolved to ensure clarity and consistency. Additionally, a brief note on whether any preprocessing differences could have influenced the results would be helpful.*

Response: We apologise for the confusing statement in our previous description. As stated in the Supplementary Materials, the resting-state fMRI were preprocessed using FSL and ANTs, whereas the task-based fMRI were preprocessed with SPM. Our earlier response to Point 8:

*“Both resting-state and task-based fMRI data underwent **standard** preprocessing steps, including brain extraction, motion correction, normalisation, spatial smoothing, and resliced to 3mm isotropic voxels. The same customised EPI template was used for normalisation across all sessions to ensure a consistent transformation from individual space to MNI standard space.”*

—was intended to emphasize that both datasets followed conventional preprocessing practices. However, we acknowledge that the pipelines themselves were distinct for resting-state and task-based data. We have addressed this point in the revised Discussion.

Manuscript page 14,

Fourth, resting-state fMRI were preprocessed using the FSL and ANTs, while task-based fMRI were preprocessed with the SPM (Supplementary Materials). All preprocessing scripts and documentation are openly available at: https://github.com/imagen2/imagen_processing/. Despite the use of different pipelines, we found that the four FNC states were highly similar between resting-state and task-based fMRI sessions (Figure 2a), suggesting that the identified FNC states are robust to preprocessing variations.

6. *The authors are encouraged to consider implementing a scrubbing procedure to further reduce motion-related artifacts. This method preserves the number of time points while mitigating motion effects. At least, evaluating the correlation between head motion and FNC measures would help confirm that motion has been sufficiently controlled for.*

Response: In the revised manuscript, we have added analyses to examine the potential influence of head motion on our results. First, we assessed the correlation between subjects’ head motion (framewise displacement, FD) and FNC state occurrences (Fig. S23). These correlations were generally low, with median values ranging from -0.046 (REST state 3 and FD) to 0.078 (REST state 4 and FD). Second, we incorporated head motion as a covariate in the correlation analyses between FNC state occurrences and task events (Fig. S24). These results remained consistent, further supporting the robustness of our findings. We have added these analyses to the **Sensitivity analysis** section of the Manuscript and the Supplementary Materials.

For analyses involving correlations between FNC state dwell time and task performance (Fig. S10) as well as behavioral measures (Figure 4), we regressed out age, sex, site, and head motion prior to the correlation analysis. As shown in Author Response Figure 1, the residuals of dwell time no longer correlate with FD. This procedure is clarified in the **Summary of Analysis Strategy** (Manuscript page 6), where we state: “*Throughout the manuscript, age, sex, recruitment sites, and head motion were regressed in between-subject analyses.*”

Similarly, for the analysis of behavioral variance explained by time-varying and static FNC (Figure 6), we regressed the FNC connectivity strength on age, sex, site, and head motion prior to the regression analysis. As shown in Author Response Figure 2, the residuals of connectivity strength also no longer correlate with FD.

Manuscript page 13,

To minimize the influence of head motion on our results, we excluded subjects with excessive head motion (mean framewise displacement (FD) > 0.2 mm; see Table S1). Additionally, in the estimation of time-varying FNC, we included the six realignment parameters as covariates in the sliding window analysis to reduce the impact of micro-movements on FNC estimation (Materials and Methods). We evaluated the correlation between FD and state occurrences during each scanning session (Fig. S23), which showed consistently low correlations (median range: -0.046 to 0.078). Furthermore, we included head motion as a covariate in the correlation analyses between state occurrences and task events (Fig. S24), and found that the results from the EFT and SST sessions remained robust. In all between-subject analyses, covariates including age, sex, recruitment site, and head motion (mean FD) were regressed out.

Fig. S23 Correlation between subjects' head motion (frame-wise displacement, FD) and state occurrences during each scanning session.

Fig. S24 Replication of the correlation between FNC state occurrence and task conditions with head motion (FD) added as an additional regressor using partial correlation. Correlation coefficients (r values) were computed using the same method as in the main results shown in Figure 3. Asterisks indicate FDR-corrected statistical significance across the three tasks. For EFT and SST, we replicated all significant results in the main analysis, while for MID, negative correlation between state 2 and large-win trials was not replicated with FD as a regressor.

Author Response Figure 1. Correlation between head motion (FD) and FNC state dwell time (i.e., the average duration a subject remains in a given state) across subjects. Dwell time was regressed on age, sex, site, and head motion prior to its correlation with task performance (Fig. S10). The residuals of dwell time no longer show a correlation with FD.

Author Response Figure 2. Correlation between head motion (FD) and both time-varying and static FNC across subjects. Task-specific connectivity strength was derived from each FNC state and from static connectivity estimates, as described in the Results section *Time-varying FNC explains more behavioral variance than static FNC*. Connectivity strength was regressed on age, sex, site, and head motion before correlating with behavioral items (Figure 6). The residuals of connectivity strength no longer show a correlation with FD.

7. *In Figure 2, the percentages for the four FNC states in the EFT and MID conditions still do not sum to 100%. The authors are encouraged to carefully review the manuscript and all supplementary figures to ensure accuracy and consistency throughout.*

Response: We appreciate the Reviewer’s attention to detail. We have carefully checked the percentages reported for the four FNC states in Figure 2. The numbers presented are correct; however, due to decimal rounding, the percentages for the EFT and MID tasks do not sum exactly to 100%. For the EFT session, the number of sliding windows assigned to the four states were: 121,533; 49,912; 14,661; and 46,286, respectively—summing to a total of 232,392 windows, which corresponds to 1,263 subjects \times 184 sliding windows. The corresponding percentages are 52.30%, 21.48%, 6.31%, and 19.92%, which sum to 100.01%. For the MID

session, the number of sliding windows were 100,690; 42,319; 37,737; and 41,476, respectively—summing to 222,222 windows, equivalent to 1,221 subjects × 182 sliding windows. The respective percentages are 45.31%, 19.04%, 16.98%, and 18.66%, totaling 99.99%.

To avoid confusion, we have added a clarifying note in the legend of Figure 2. We have also carefully reviewed the manuscript and Supplementary Materials to ensure consistency and accuracy throughout.

Manuscript page 25, Legend of Figure 2:

Figure 2 Resting-state and task-fMRI sessions contain reoccurring FNC states. a) Four representative FNC states (cluster centroids) were derived from the *k-means* clustering applied to sliding-window FNC matrices from the resting-state (upper row) and three task-fMRI sessions (lower row). Each FNC state represents a reoccurring functional connectivity pattern between 61 ICA components, which are categorised into seven functional domains (Figure S2). The percentage of each state within a scanning session is shown in parentheses. (Note: Due to decimal approximation, the percentages for the four FNC states in the EFT and MID sessions sum to 100.01% and 99.99%, respectively).

Reviewer #1 (Remarks on code availability):

I reviewed some of the code without executing it. The code appears clear and well-organized, and the accompanying README file is well-structured and easy to follow.

Response: We thank the Reviewer for the positive feedback. We appreciate the time taken to review the materials and code to improve clarity of this manuscript.

Reviewer #2 (Remarks to the Author):

The authors have thoroughly addressed all my comments and concerns. The changes have greatly improved the clarity of the manuscript. I believe this paper is now suited for publication in Nature Communications. Great job - I look forward to seeing it out.

Response: We sincerely thank Reviewer #2 for the thoughtful and encouraging comments. We are grateful for your careful review and constructive feedback, which greatly helped us improve the clarity and quality of the manuscript.

Reviewer #2 (Remarks on code availability):

The code has been improved with the revision. There is now a clear README with step by step instructions and information about external tools that were used. The authors also seemed to have made the code more flexible and easy to use with the data they provided.

Response: We appreciate the Reviewer's positive remarks on the improved code and documentation.

Reviewer #3 (Remarks to the Author):

I thank the authors for addressing all issues raised in the previous review thoroughly and to the point. All the responses were detailed and clear. I have no other comments.

Response: We thank the Reviewer for the thoughtful evaluation and constructive suggestion to our manuscript. We great appreciate the time and efforts for the Reviewer to help us improve our work.

Reviewer #3 (Remarks on code availability):

I did not attempt to run the codes. As far as I could see the codes reproduce the statistical analyses provided in the results given the main metrics pre-calculated using another library and stored in data folder. The codes look well organised. It would be useful to specify which Matlab toolboxes are required to run the code in the readme document (e.g. statistics..etc).

Response: We thank the Reviewer for the suggestion. We have added the Matlab toolboxes required to run the code in each script. Please see an example of the modification below:

```
xchang007 Update S_FNC_states_corr_task_EFT.m d81d201 · 2 weeks ago History
Code Blame 88 lines (65 loc) · 3.52 KB Raw Copy Download Edit History
1 % FNC state occurrences corr with task stimuli, group level, EFT
2 addpath(genpath('..\EffectSizeToolbox_v1.61'))
3 % https://github.com/hhentschke/measures-of-effect-size-toolbox
4 addpath(genpath('..\GroupICATv4.0b'))
5 % https://github.com/trendscenter/gift
```

Reviewer #4 (Remarks to the Author):

Response: We thank the Reviewer for the time and effort in reviewing our manuscript. We have provided a point-by-point response to each question above and hope that the Reviewer finds our revisions satisfactory.